EMBO
Molecular Medicine

# Extracellular vesicle-based targeted protein degradation platform for multiple extracellular proteins

Bide Tong[1,2,4], Xiaoguang Zhang [iD][1,4], Dingchao Zhu[1,4], Yulei Wang[3,4], Junyu Wei[1], Zixuan Ou[1], Huaizhen Liang [iD][1], Hanpeng Xu[1], Zhengdong Zhang[1], Jie Lei[1], Xingyu Zhou[1], Di Wu[1], Yu Song [iD][1], Kun Wang[1], Xiaobo Feng[1], Lei Tan [iD][1✉], Zhiwei Liao [iD][1,2✉] & Cao Yang [iD][1,2✉]

## Abstract

**Targeted protein degradation (TPD) is an emerging therapeutic approach that enables the degradation of undruggable targets via intracellular degradation systems. Extracellular vesicles (EVs) have shown potential to act as next-generation TPD platforms. However, the molecular mechanism underlying their degradation remains unknown, which restricts their application in TPD. In this study, we found that the autophagy-mediated lysosomal pathway was the major route by which EVs were degraded. MAP1LC3B recognized the LIR motifs of SQSTM1 and induced the degradation of EVs in the autophagy pathway. Based on the EV degradation mode, we developed an EV-based targeted protein degradation platform (EVTPD) using EVs loaded with the LIR motif of SQSTM1 as a degradation signal. Additionally, target protein-binding domains were integrated into the EVTPD to capture target proteins. EVTPD selectively degraded extracellular proteins without requiring receptors on target cells. Furthermore, dual-targeting EVTPD effectively degraded both TNF-α and IL-1β and exhibited potent anti-inflammatory effects in rat and goat models of intervertebral disc degeneration. This study has established a modular EV-based TPD strategy with multi-targeting potential.**

**Keywords** Targeted Protein Degradation; Extracellular Vesicles; Autophagy; Inflammation; Intervertebral Disc Degeneration
**Subject Category** Biotechnology & Synthetic Biology

## Introduction

Targeted protein degradation (TPD) is an emerging treatment that degrades undruggable targets via intracellular degradation systems. TPD has attracted wide attention over the past two decades (Békés

et al, 2022). The ubiquitin-proteasome system was first used by proteolysis targeting Chimeras (PROTACs), which is the earliest TPD system that degraded intracellular targets and showed excellent degradation activity against methionine aminopeptidase-2 (Sakamoto et al, 2001). After approximately two decades of development and improvement, PROTAC entered clinical trials in 2019 and showed promising activity in cancer therapies (Chirnomas et al, 2023). However, the high cost of developing an effective combination of protein-targeting ligand and E3 ligase severely hindered the discovery of PROTACs with new targets (Li and Crews, 2022). In contrast, the ubiquitin–proteasome system-based degradation mode enabled PROTACs to degrade intracellular proteins without affecting the cell membrane or extracellular proteins (Zhao et al, 2022). The antibody-based PROTACs (AbTACs) is a modified PRPTAC-based technique that simplified PROTAC development for new intracellular targets by harnessing the antibody-binding characteristics and extending targeted degradation to cell membrane targets by using transmembrane E3 ligases such as RNF43 (Cotton et al, 2021) and ZNRF3 (Marei et al, 2022). However, the AbTAC is an antibody-based method that faces inherent limitations of biocompatibility, self-degradation in vivo, and non-targeting of extracellular proteins. The lysosomal degradation pathway is another important intracellular degradation system that has not been used for target protein degradation until the development of the lysosome-targeting chimeras (LYTACs) in 2020 (Banik et al, 2020). LYTACs use glycopeptide ligands to recognize CI-M6PR and degrade the cell surface and extracellular molecules via lysosomes. This approach mitigates the limitation of the TPD targeting scope. Subsequently, cytokine receptor-targeting chimeras (KineTACs) were developed to degrade cell-surface and extracellular proteins by inducing lysosomes in a CXCR7-dependent manner (Pance et al, 2023). However, both lysosome-based degradation protocols require high expression of specific cell membrane proteins (CI-M6PR and CXCR7) on the cell surface, which is difficult to achieve in several organs and tissues, thereby limiting their applications. Several novel degradation systems such

[1]Department of Orthopaedics, Union Hospital, Tongji Medical College, Huazhong University of Science and Technology, Wuhan, China. [2]Shenzhen Huazhong University of Science and Technology Research institute, Shenzhen, China. [3]Department of Histology and Embryology, Tongji Medical College, Huazhong University of Science and Technology, Wuhan, China. [4]These authors contributed equally to this work as first authors: Bide Tong, Xiaoguang Zhang, Dingchao Zhu, Yulei Wang. ✉E-mail: tanlei_union@hust.edu.cn; lzwhust@hust.edu.cn; caoyangunion@hust.edu.cn

as Apatamer-LYTAC, DNA framework-engineered chimeras, and autophagy-targeting chimeras (AUTACs) have been introduced in recent years (Takahashi et al, 2019; Wu et al, 2023; Zhou et al, 2023). However, the challenges in developing target-binding molecular ligands and the unpredictable biocompatibility and in vivo clearance remain urgent problems in the field of TPD. Furthermore, its 3-element complex nature restricts the targeting of the TPD technology primarily to single targets, whereas pathological processes such as cancers, autoimmune diseases, and degenerative diseases usually occur as the result of synergistic actions of multiple targets (Koeberle and Werz, 2014; Luo et al, 2020; Tao et al, 2015). Thus, a TPD technology that simultaneously targets multiple targets is needed to treat these diseases.

Extracellular vesicles (EVs) are cell-derived nanovesicles that contain RNA, proteins, and DNA from donor cells (El Andaloussi et al, 2013). EVs are widely distributed in the body and perform intercellular communication to regulate various biological processes such as tumorigenesis and immune responses (Buzas, 2023). EVs are enriched in a variety of bioactive molecules and immunomodulatory functions; hence, they have been widely applied in tumors, autoimmune diseases, and neurodegenerative diseases and have shown promising prospects for clinical transformation (Herrmann et al, 2021). Moreover, EVs are nanosized with good biocompatibility and CD47-mediated anti-phagocytosis mechanisms; hence, they may be engineered to effectively load drugs by donor cell gene editing or in vitro loading, which makes EVs a highly promising next-generation drug delivery platform (Herrmann et al, 2021; O'Brien et al, 2020). Furthermore, EVs are biological carriers and may be transported into the lesion tissue via the circulatory system without being engulfed or causing an immune response. Thus, they may be easily loaded with biological ligands to target receptors to compensate for the limitations of the existing TPD system, including uncertain biocompatibility, phagocytosis, and challenges in designing molecular ligands (de Abreu et al, 2020; Roefs et al, 2020). However, the intracellular transport mechanism and degradation pathway associated with EVs remain poorly understood because of the lack of efficient labeling and imaging technologies to monitor their intracellular transport and degradation(van Niel et al, 2018). Although some studies loaded GFP and tetracycline transactivator (tTA) into EVs to discover their intracellular transport process after endocytosis (Joshi et al, 2020; Somiya and Kuroda, 2021), these studies were conducted primarily to identify the EV cargo delivery process rather than their degradation. The lack of knowledge of the EV degradation mechanisms significantly hinders the engineering of EVs for application in TPD.

Low back pain (LBP) is a leading contributor to disability, and it imposes a huge global disease burden (Knezevic et al, 2021; Maher et al, 2017). The major cause of LBP is intervertebral disc degeneration (IVDD), which is characterized by an increase in the levels of multiple inflammatory cytokines such as TNF-α, IL-1β, IL-6, and IL-17 in the lesion discs (Risbud and Shapiro, 2014). These cytokines—especially TNF-α and IL-1β—disturb the disc cells, such as nucleus pulposus cells (NPC) and annulus fibrosus cells, and cause an imbalance in the synthesis and degradation of the IVD extracellular matrix, which leads to herniation. Furthermore, they stimulate local nerves and cause radicular pain. As cytokines play an initial promoting role in IVDD, anti-cytokine therapy is a promising treatment option for early prevention of IVDD. Additionally, antibody- and inhibitor-based therapies targeting inflammatory cytokines have benefited patients with LBP in terms of both short-term pain relief and long-term reduction in the number of required surgeries (Genevay et al, 2012; Genevay et al, 2010). However, an inflammatory cytokine-targeting TPD system has not been applied to IVDD because of the lack of suitable TPD techniques to target inflammatory cytokines in IVDD. Thus, the development of TPD technology adapted to the IVD environment would effectively remove the local inflammatory environment and alleviate IVDD.

In this study, we investigated the degradation properties of EVs and engineered EVs based on these degradation mechanisms to generate an efficient modular degradation platform. We termed this EV-based degradation platform as EVTPD and applied it to degrade TNF-α and IL-1β to alleviate IVDD. In addition, we applied EVTPD to a goat model of IVDD to validate its clinical translational potential.

# Results

## Degradation of extracellular vesicles occurs through an autophagy-dependent lysosomal pathway

We monitored EV degradation by transfecting CD63-EGFP plasmids into the HEK293T cell line to generate EGFP-labeled EVs (Fig. 1A). EGFP-EVs exhibited a typical cup shape with an average diameter of 131.7 nm (Fig. EV1A,B). Western blotting results showed that, in addition to the typical EV markers such as ALIX, CD9, TSG101, and CD63, EGFP was expressed in the extracted EVs (Fig. EV1C,D). Next, we monitored the degradation of these EGFP-expressing HEK293T-derived EVs in the NPCs by determining the EGFP protein level in the NPCs. Three independent EGFP-EV batches ($5 \times 10^3$ cells) were incubated for 12 h, followed by washing and degradation for 12 h before EV degradation was determined. We found that the natural half-life of EVs in these NPCs was ~12 h, which was selected as the observation period (Fig. EV1E). Eukaryotic cells possess two primary protein degradation pathways: the lysosomal pathway, which is specifically inhibited by BafA1, and the proteasomal pathway, which is specifically inhibited by MG132 (Fig. 1G). To determine which degradation pathway plays the primary role in EV degradation, we pretreated NPCs with BafA1 or MG132 and examined the EGFP-EV degradation level in the NPCs. The results showed that BafA1 significantly inhibited the degradation of the HEK293-derived EVs, whereas MG132 did not exert a significant effect, which indicates that the HEK293T-EVs were primarily degraded through the lysosomal pathway rather than the proteasomal pathway (Figs. 1B and EV1F). Moreover, we confirmed that HeLa- and mesenchymal stem cell (MSC)-derived EVs predominantly underwent degradation via the lysosomal pathway (Figs. 1B and EV1G,H). Similar results were obtained when HeLa cells were used under the same experimental conditions (Fig. EV1I–L), which further confirmed the predominant role of the lysosomal pathway in EV degradation.

After EV uptake, a portion of EVs are exocytosed back to the extracellular space through the endosomal recycling process (Mares et al, 2011; Morad et al, 2019). Therefore, the attenuation of intracellular EV signals may not only be attributed to EV degradation but also to their exocytosis. To detect the level of EV exocytosis and whether it was affected by the degradation system,

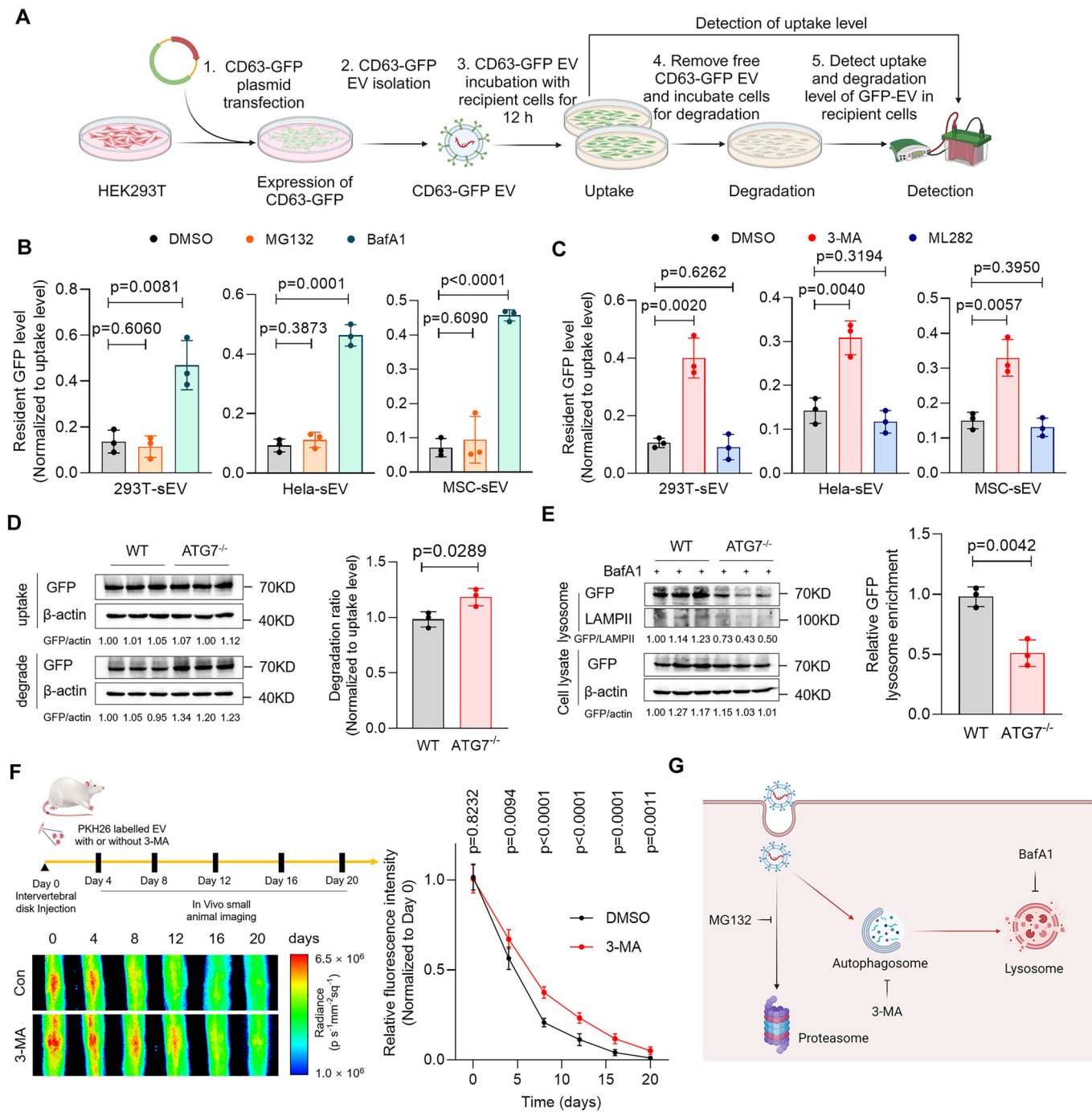

**Figure 1. The degradation of extracellular vesicles occurs through an autophagy-dependent lysosomal pathway.**

(A) Schematic diagram of GFP-EV generation and detection. HEK293T-derived EGFP-EVs were added to recipient cells at a concentration of $5 \times 10^3$ per cell and incubated for 12 h for uptake. Then, the supernatants were removed and washed twice with PBS. Cells were collected to measure cellular uptake levels at this time, while the degradation groups were incubated in fresh complete medium for another 12 h. After 12 h of degradation time, cells were collected. GFP levels were examined by western blot assay. (B) Quantitative analysis of the resident GFP levels of HEK293T, HeLa, and MSC-derived EVs in the DMSO, MG132, and BafA1 groups NPCs ($n = 3$). (C) Quantitative analysis of the resident GFP level of HEK293T, HeLa, and MSC-derived EVs in the DMSO, 3-MA, and ML282 groups NPCs ($n = 3$). (D) Representative immunoblot images and quantitative analysis of the resident GFP level of HEK293T-derived EVs in the wild-type and $ATG7^{-/-}$ HeLa cell lines ($n = 3$). (E) Representative immunoblot images and quantitative analysis of the GFP lysosomal enrichment level of HEK293T-derived EVs in the wild-type and $ATG7^{-/-}$ HeLa cell lines ($n = 3$). (F) SD rat intervertebral disc injection model and in vivo small animal imaging of SD rat intervertebral discs at 0, 4, 8, 12, 16, and 20 days ($n = 6$). (G) Schematic diagram of the main protein degradation pathways in cells. Data were analyzed by unpaired two-tailed $t$ tests (B–F). Data are shown as mean ± SD. Each $n$ in (F) is an individual rat, $n$ in (B–E) is a biological independent sample. The $P$ values are labeled in the figure. Source data are available online for this figure.

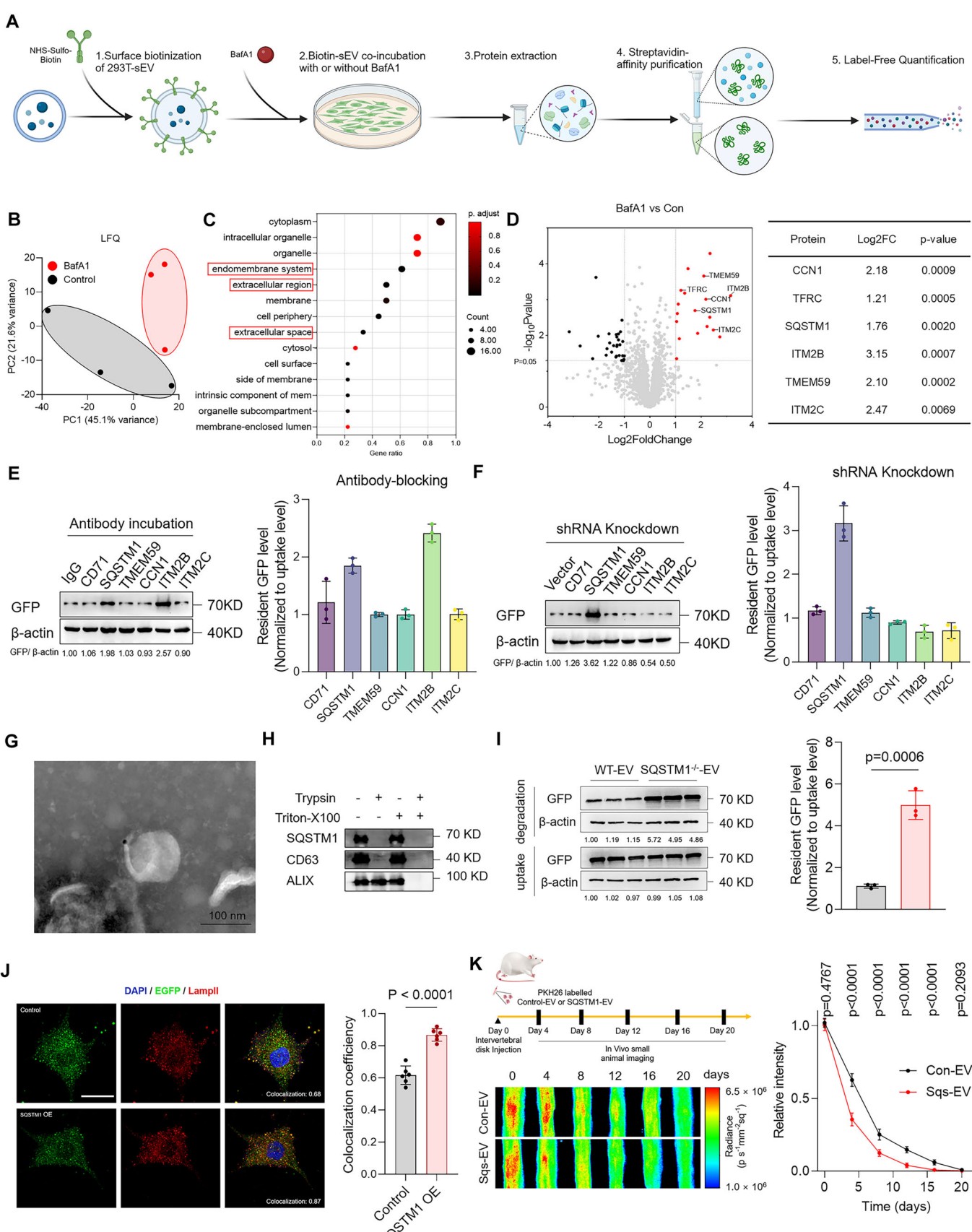

**Figure 2. SQSTM1 mediates the entry of extracellular vesicles into lysosome-dependent degradation pathways.**

(A) Schematic diagram of the analysis of EV-proteomic changes before and after degradation based on biotin-labeling technology. (B) PCA analysis of EV proteins in control and BafA1 groups ($n = 3$). (C) GO enrichment analysis of upregulated genes in the BafA1 group compared with the control group ($n = 3$). (D) Volcano plot and table of the top six upregulated EV proteins in the BafA1 group compared with the control group ($n = 3$). (E) EVs were blocked with specific antibodies before being co-incubated with cells. Their degradation efficiency was examined by analysis of the resident GFP level of NPCs in different groups ($n = 3$). (F) The effect of specific gene knockdown on EV degradation efficiency was examined by analysis of the resident GFP level of NPCs in different groups ($n = 3$). (G) Representative immunoelectron microscopy (IEM) images of SQSTM1 expression on the surface of HEK293T-derived EVs (Scale bar: 100 nm). (H) 0.25% Trypsin was added at a one-quarter volume ratio to digest HEK293T-derived EVs with or without 1% Triton-X-100 at 37 °C for 2 h. EV without trypsin degradation was used as a negative control. The expression of SQSTM1, CD63, and ALIX of the resulting EVs were examined by western blot assays. (I) Representative immunoblot images and quantitative analysis of the resident GFP levels of HeLa-derived EVs in the wild-type and $SQSTM1^{-/-}$ HeLa cell lines ($n = 3$). (J) Immunofluorescence images and co-localization analysis of EV-EGFP and intracellular Lamp II (scale bar: 10 μm) ($n = 6$). (K) Schematic illustration of PKH26-labeled control and SQSTM1-overexpression EVs injected into intervertebral discs of SD rats. In vivo small animal imaging system was used to monitor and analyze the degradation rate of control and SQSTM1-overexpression EVs ($n = 6$). Data were analyzed by unpaired two-tailed $t$ tests (D, I–K). Data were shown as mean ± SD. Each $n$ in (K) is an individual rat, $n$ in (B–F, I, J) is a biological independent sample. The $P$ values are labeled in the figure. Source data are available online for this figure.

we labeled the EVs with CFSE and incubated them with cells for 12 h to allow EV uptake. Then, we washed the remaining uninternalized EVs from the extracellular fluid and detected the intracellular EV signal as the initial overall level. After an additional 12 h of incubation, we collected the supernatant and detected the CFSE signal to calculate the level of exocytosis of CFSE-EVs in 12 h. By calculating the proportion of the CFSE signal attributed to exocytosis to that of the initial total CFSE signal, we inferred the proportion of EVs that underwent exocytosis to that of the total internalized EVs. The results showed that the signal from exocytosed EVs accounted for 3–5% of the initial signal, and this proportion was not affected by the inhibition of lysosomal degradation by BafA1 (Appendix Fig. S1f). This result may be attributed to the endocytic sorting process of EVs before they enter the lysosomal degradation system.

As the autophagy pathway is an important source of lysosomal degradation, we used 3-MA to observe its contribution to EV degradation (Fig. 1G). The results showed that 3-MA significantly inhibited the degradation of HEK293-derived EVs in NPCs, which suggests that autophagy is essential for EVs to enter the lysosomal pathway for degradation (Figs. 1C and EV1M). Similar results were observed for HeLa- and MSC-derived EVs, which further supports the dominant role of the autophagy pathway in EV degradation (Figs. 1C and EV1N,O). Rab7 is a coordinator of intracellular vesicular traffic and is likely to play a role in the sorting of EVs into lysosomes (Stenmark, 2009). Furthermore, ML282 is a known Rab7GTPase inhibitor that efficiently inhibits Rab7 activity (Appendix Fig. S1d). Therefore, we used ML282 to determine the role of Rab7 in EV degradation (Hong et al, 2010). The results showed that ML282 did not exert a significant effect on EV degradation, which suggests a limited role for Rab7 in sorting EVs into lysosomes (Figs. 1C and EV1M–O). In addition, 3-MA significantly inhibited EV degradation in HeLa cells (Fig. EV1P–S). Furthermore, we knocked down ATG7 in HeLa cells and compared the EV degradation rates with those in wild-type HeLa cells. The results showed that ATG7 knockout (KO) significantly inhibited EV degradation, which indicates the important role of autophagy in EV degradation (Fig. 1D). Furthermore, we conducted an ATG7 rescue experiment to rule out the influence of clonal variation in the ATG7 KO cells. The results showed that ATG7 expression in the ATG7 KO cells effectively restored the EV degradation efficiency (Appendix Fig. S1e) and confirmed the role of the autophagy pathway in EV degradation.

As both normal lysosomal function and natural autophagy process are essential for EV degradation, we hypothesized that EV sorting into lysosomes occurs via the autophagy pathway. To validate this hypothesis, we observed the contribution of the autophagy pathway to intracellular EV sorting by using BafA1 to disrupt the lysosomal function and 3-MA to interfere with autophagy. Fluorescence co-localization experiments showed that EV co-localization and LAMPII (lysosomal marker) expression decreased significantly after blocking the autophagy pathway, whereas inhibiting Rab7 with ML282 did not elicit a similar effect (Fig. EV1T; Appendix Fig. S2a). In addition, we isolated the lysosomes from the NPCs and found that 3-MA pretreatment significantly reduced the EGFP level in the NPC lysosomes (Fig. EV1U). Moreover, the $ATG7^{-/-}$ cell line and ATG7 knockdown NPCs showed a significantly lower lysosomal EV sorting efficiency compared with that of the wild-type cells. This further confirmed the role of the autophagy pathway in sorting EVs into lysosomes (Figs. 1E and EV1V). To further verify the role of autophagy in EV degradation in vivo, we injected PKH26-labeled EVs with or without 3-MA into the IVDs of SD rats and observed EV degradation (Fig. 1F). Compared with that in the control group, EV degradation rate in the IVDs was significantly lower in the 3-MA group. This confirmed the primary role of autophagy in EV degradation in vivo (Fig. 1F). Overall, these results confirm that EVs are primarily sorted into lysosomes for degradation in an autophagy-dependent manner.

## SQSTM1 mediates extracellular vesicle sorting into the lysosome-dependent degradation pathway

To identify the proteins within the EVs that play major roles in mediating EV sorting into lysosomes, we used a biotin-labeled method to monitor the proteomic change associated with EVs during their degradation (Fig. 2A). The EV-associated proteins were labeled with NHS-Sulfo-biotin in vitro. The biotinylated EVs were incubated with cells for 12 h. Then, BafA1 was added to block EV degradation. BafA1 was not added to the control group to enable normal EV degradation. We compared the biotin-labeled proteomes of the BafA1 and control groups to identify the EV-associated proteins that showed the highest degradation rates, as these were most likely to be the proteins that mediate EV degradation. Principal component analysis (PCA) showed different protein expression patterns between the two groups (Fig. 2B). Gene ontology (GO) analysis of the differentially expressed proteins

showed that most of these proteins were categorized into the GO classes of extracellular space, cytoplasm, and endomembrane system, which are closely related to EV composition (Fig. 2C). This confirms that our biotin-labeled method specifically reflects changes in proteins during EV degradation. We hypothesized that the proteins with the highest degradation rates would mediate EV degradation. Therefore, we selected the first six proteins with the largest fold changes between the control and BafA1 groups (CD71, SQSTM1, TMEM59, CCN1, ITM2B, and ITM2C) for further experiments (Fig. 2D). We investigated their contribution to EV degradation by blocking them with antibodies and observing the effects on the EV degradation efficacy (Fig. 2E). IgG was used as the negative control. We found that antibody application did not affect EV uptake by the cells (Fig. EV2A). The degradation assay showed that blocking ITM2B and SQSTM1 elicited the most pronounced effect on EV degradation, which suggests that these proteins play important roles in EV degradation. Additionally, we used shRNA to knock down the corresponding genes in HEK293T cells and isolated the knockdown EVs. pLKO.1-Scrambled encodes a non-targeting random sequence, and this was used as the negative control. The knockdown efficacy was verified by western blotting (Appendix Fig. S8a–f). We found that the knockdown of these genes did not affect EV uptake (Fig. EV2B). In addition, the EV degradation assay showed that SQSTM1 depletion significantly reduced the EV degradation efficacy in the NPCs and HeLa cells (Figs. 2F and EV2C). This indicates that SQSTM1 plays an important role in EV degradation.

SQSTM1 is a cytosolic autophagy receptor that is known for its role in recruiting ubiquitinated substrates to form droplets and delivering these cargoes to autophagosomes (Lamark et al, 2017). However, whether SQSTM1 is sorted into the EVs especially on the EV surface remains unknown. Therefore, we observed SQSTM1 expression on EVs using immunoelectron microscopy (IEM). The results showed that at least part of SQSTM1 was expressed on the EV membrane (Fig. 2G). EVs produced by SQSTM1$^{-/-}$ cells were used as negative controls to confirm the specificity of SQSTM1 staining (Fig. EV2D). In addition, we performed a digestion assay and confirmed that SQSTM1 was expressed on the surface of the EV membrane, and these were digested with trypsin without permeabilization (Fig. 2H). ALIX is a well-established intraluminal EV marker that has been used as a representative indicator of changes in intraluminal protein levels (Baietti et al, 2012). Treatment with trypsin alone did not alter the ALIX levels in EVs, which confirmed that trypsin alone did not target intraluminal proteins within EVs (Fig. 2H). Co-immunoprecipitation assay showed that the SQSTM1 antibody efficiently bound and enriched the EVs, which confirmed SQSTM1 expression on the EV surface (Fig. EV2E). The loading mechanism of SQSTM1 on EVs may be based on the post-translational modification of SQSTM1 itself (S-acylation) or the interaction between SQSTM1 and membrane proteins such as LC3 (Huang et al, 2024; Kraft et al, 2016).

The acidification and maturing of the endosomes disrupts the antibody–EV binding; therefore, the effect of EV blocking by the antibodies is limited to the period before endosomal maturation (Igawa et al, 2010). As SQSTM1 blocking significantly inhibits the EV degradation efficiency, we hypothesized that SQSTM1-mediated EV sorting occurs before the full acidification of the endosomes. To avoid the effect of the SQSTM1 antibody on intracellular SQSTM1, we observed EV degradation in SQSTM1$^{-/-}$

cells. We found that the EV degradation rates in the SQSTM1$^{-/-}$ cells were not significantly different from those of the wild-type cells (Appendix Fig. S1l). In addition, we used HeLa-derived GFP EVs to minimize potential bias from the difference between the cell lines. The results showed that SQSTM1 knockout in recipient HeLa cells did not influence the degradation of HeLa-derived EVs, which is consistent with the results obtained for HEK293T-derived EVs (Appendix Fig. S1m). Moreover, we treated NPCs, HeLa cells, and HEK293T cells with GFP-EVs generated from WT and SQSTM1$^{-/-}$ cells. These results showed that regardless of the type of cells used, SQSTM1 KO in the EVs effectively inhibited their degradation efficiency (Fig. 2I; Appendix Fig. S1j,k). These results confirm the important role of EV-SQSTM1 rather than that of intracellular SQSTM1 in EV degradation.

As both antibody-blocking and knockdown experiments confirmed the important role of SQSTM1 in EV degradation, we examined the effect of SQSTM1 on lysosomal sorting of EVs. We isolated the lysosomes from the recipient cells and found that the SQSTM1-knockdown EVs were less enriched in lysosomes of NPCs compared with that observed for the control EVs (Fig. EV2F). Immunofluorescence co-localization analysis showed that SQSTM1 knockdown in EVs deregulated EV co-localization and LAMPII expression (Fig. EV2G). Moreover, SQSTM1-knockdown EVs showed a lower sorting efficiency to lysosomes in HeLa cells (Fig. EV2H), which indicates that SQSTM1 mediates EV entry into lysosomes.

We investigated the effect of SQSTM1 overexpression on EV degradation. EV degradation assay showed that SQSTM1 over-expression significantly upregulated EV degradation in NPCs (Fig. EV2I). Moreover, SQSTM1 overexpression in EVs elicited a higher sorting efficacy into lysosomes in NPCs (Figs. 2J and EV2J). Similar results were obtained in HeLa cells, which confirmed the lysosomal sorting ability of SQSTM1 (Fig. EV2K,L). We further investigated the effect of SQSTM1 on EV degradation in vivo by injecting PKH26-labeled control EVs and SQSTM1-overexpressing EVs into the IVDs of SD rats and observed its degradation. The results showed that SQSTM1 overexpression significantly accelerated EV degradation in the IVDs of SD rats (Fig. 2K). Overall, these results show that EV-associated SQSTM1 mediates EV lysosomal sorting and subsequent degradation.

## EVs engineered with SQSTM1 degradation signal were efficiently sorted into lysosomes and degraded via the interaction with MAP1LC3B

Next, we explored the molecular mechanism underlying SQSTM1-mediated entry of EVs into lysosomes. EV membrane surface proteins interact with intracellular receptors that determine the intracellular EV transport routes (Kooijmans et al, 2021). We identified the intracellular receptors that interact with EV-associated SQSTM1 by tagging SQSTM1 with Flag and analyzing the interacting proteins using IP-MS (Fig. 3A). EVs lacking the Flag-SQSTM1 were used as the controls. GO analysis showed that the interacting proteins were highly enriched in the autophagosome pathway (Fig. 3B). Among them, MAP1LC3B was the most abundant after SQSTM1 (Fig. 3C). As MAP1LC3B and MAP1LC3B2 could not be distinguished using mass spectrometry, we examined the MAP1LC3B and MAP1LC3B2 expression in NPCs and HeLa cells using qRT-PCR. The results showed that

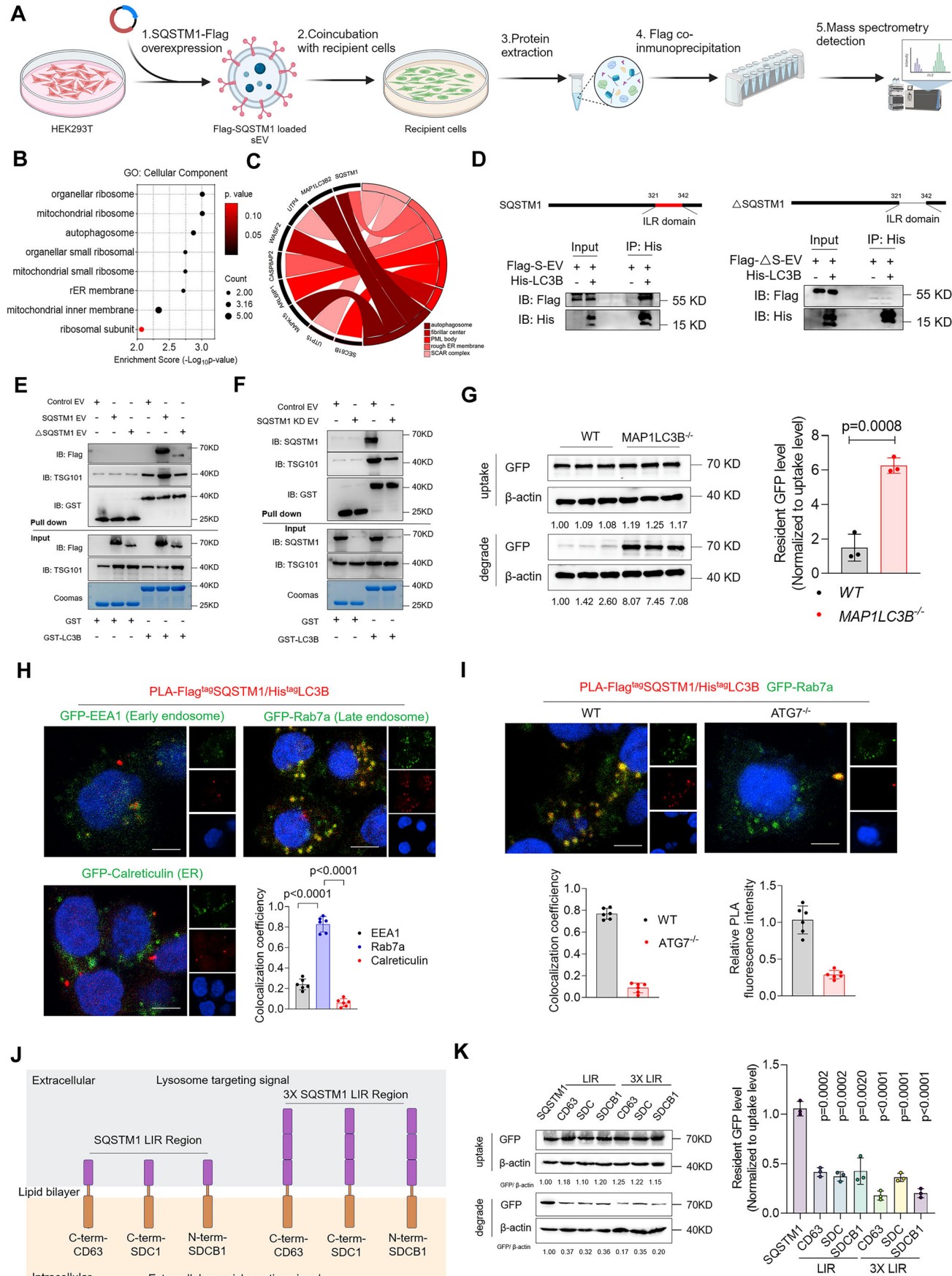

**Figure 3. EVs engineered with SQSTM1 degradation signal can be effectively sorted into lysosomes and degraded via interactions with MAP1LC3B.**

(A) Schematic diagram of IP-MS identification of EV-SQSTM1-interacting proteins in NPCs; blank-EVs were used as negative controls. (B) GO analysis of enriched EV-SQSTM1-interacting proteins. (C) Chord diagram of EV-SQSTM1-interacting proteins and the GO pathways. (D) The interaction between Flag-SQSTM1 and His-MAP1LC3B was examined in His-MAP1LC3B-transfected HeLa cells after incubation with Flag-SQSTM1-EVs for 24 h with BafA1. The interaction between truncated Flag-SQSTM1 with MAP1LC3B was also examined in His-MAP1LC3B-transfected HeLa cells after incubation with truncated Flag-SQSTM1-EVs for 24 h with BafA1. (E) GST or GST-LC3B were used to pull down control EVs, SQSTM1 EVs or truncated SQSTM1 EVs. (F) GST or GST-LC3B was used to pull down control EVs or SQSTM1 knockdown EVs. (G) The uptake and degradation level of the SQSTM1-EVs were detected in wild-type and MAP1LC3B-knockout HeLa cell lines ($n = 3$). (H) PLA assay was used to detect the interaction sites (Red) of EV-Flag-SQSTM1 and intracellular His-MAP1LC3B in HeLa cells. Meanwhile, HeLa cells were transfected with GFP-EEA1, GFP-Rab7a, or GFP-Calreticulin to visualize the subcellular location of interaction sites. Co-localization of PLA signals and GFP was analyzed by ImageJ ($n = 6$). Scale bar: 10 μm. (I) The interaction sites (Red) of EV-Flag-SQSTM1 and intracellular His-MAP1LC3B in wild-type and ATG7 KO HeLa cells were detected by PLA assay. Co-localization coefficient of PLA/GFP-Rab7a and relative fluorescence intensity of PLA signals were analyzed by ImageJ ($n = 6$). Scale bar: 10 μm. (J) Schematic diagram of constructing EV degradation signal peptides. (K) The uptake and degradation level of EVs loaded with SQSTM1 or degradation signal peptides were detected in HeLa cells ($n = 3$). Data were analyzed by unpaired two-tailed $t$ tests (G–I, K). Data are shown as mean ± SD. Each $n$ in (G–I, K) is a biological independent sample. The $P$ values are labeled in the figure. Source data are available online for this figure.

MAP1LC3B showed a predominant tenfold higher expression than that of MAP1LC3B2 in both NPCs and HeLa cells (Fig. EV3A). Therefore, we inferred that MAP1LC3B rather than MAP1LC3B2 plays a major role in EV-SQSTM1 recognition.

We verified the interaction between EV-associated SQSTM1 and intracellular MAP1LC3B by tagging MAP1LC3B with His to differentiate it from the EV-associated MAP1LC3B and performed Co-IP. The results verified that EV-SQSTM1 interacted with intracellular MAP1LC3B (Fig. 3D). As the LIR motif of SQSTM1 is essential for its binding to MAP1LC3B (Pankiv et al, 2007), we hypothesized that intracellular MAP1LC3B also binds to the same region of EV-SQSTM1. We constructed a truncated SQSTM1 protein without the LIR motif and found that the loading efficiency of the truncated protein was similar to that of full-length SQSTM1 (Fig. EV3B). However, the EVs loaded with truncated SQSTM1 lost their ability to bind to intracellular MAP1LC3B (Fig. 3D), which indicates that EV-SQSTM1 binds to MAP1LC3B via their LIR motifs. Furthermore, the truncated SQSTM1-loaded EVs showed a significantly lower degradation rate and lysosomal sorting efficacy than those of full-length SQSTM1 (Fig. EV3C,D). This indicates that the interaction between EVs-SQSTM1 and intracellular MAP1LC3B is essential for EV degradation. To further verify that EVs directly bind to MAP1LC3B via SQSTM1, we extracted SQSTM1-loaded EVs and GST-LC3B proteins and used GST-LC3B to pull down SQSTM1-EVs in vitro. Blank EVs were used in the control group. The results showed that EVs overexpressing Flag-SQSTM1 exhibited a significantly increased binding capacity for GST-LC3B, as evidenced by a pronounced increase in the TSG101 (EV marker) signal (Fig. 3E). In contrast, the pull-down levels of EVs overexpressing truncated SQSTM1 were comparable to those observed in the control group, with no significant difference in TSG101 signal intensity. The detection of the truncated SQSTM1 signal may be attributed to the low-level background expression of SQSTM1 on the EV surface, resulting in the passive pull-down of a limited number of truncated SQSTM1-loaded EVs by GST-LC3B. However, SQSTM1 knockdown in EVs reduced EV binding to GST-LC3B (Fig. 3F). These findings confirm that SQSTM1 overexpression promotes EV–LC3B interaction and that this effect is dependent on the LIR domain of SQSTM1.

We verified whether MAP1LC3B acts as an intracellular receptor of EV-SQSTM1 to mediate EV degradation by constructing *MAP1LC3B* knockdown NPCs and *MAP1LC3B*$^{-/-}$ HeLa cells to observe the effect of *MAP1LC3B* deletion on EV degradation. The results showed that the *MAP1LC3B* knockdown in NPCs significantly reduced the promotional effect of SQSTM1 overexpression on EV degradation and lysosomal sorting efficacy (Fig. EV3E,F). In addition, *MAP1LC3B*$^{-/-}$ cells showed significantly slower degradation rates and lower lysosomal sorting efficacy of SQSTM1-EVs than those of wild-type HeLa cells (Figs. 3G and EV3G). *MAP1LC3B2* knockout in HeLa cells elicited a relatively mild inhibitory effect on SQSTM1-EV degradation, which may be related to its low expression (Fig. EV3A,H,I). In contrast, MAP1LC3B overexpression in NPCs and HeLa cells enhanced their lysosomal sorting efficacy and degradation rates (Fig. EV3J–M), which confirmed the essential role of LC3B in EV degradation.

EVs do not enter the cytoplasm directly as free particles (Mulcahy et al, 2014). Therefore, the subcellular localization of the interaction between EV-SQSTM1 and intracellular MAP1LC3B remains unclear. To investigate the mechanism underlying the interaction of EV-SQSTM1 with intracellular MAP1LC3B, a proximity ligation assay (PLA) was performed to determine their subcellular localization. The EV-SQSTM1 and intracellular MAP1LC3B interaction sites were marked by PLA, whereas the subcellular markers (EEA1-early endosome (Mu et al, 1995), Rab7a-late endosome (Skjeldal et al, 2021), and Calreticulin-ER (Michalak, 2024)) fused with GFP and were simultaneously transfected, which enabled the visualization of subcellular localization. The early endosome marker GFP-EEA, late endosome marker GFP-Rab7a, and endoplasmic reticulum marker GFP-Calreticulin were found to have been accurately distributed to their respective organelles (Appendix Fig. S2c–e). We minimized the risk of false-positive signals in the PLA assay by establishing a negative control group by omitting the anti-Flag and anti-His primary antibodies, and followed the rest of the standard PLA protocol. The absence of detectable PLA signals in the negative control group confirmed the specificity of the PLA assay (Appendix Fig. S2b). The PLA results showed that the interaction sites overlapped with Rab7a with the highest co-efficiency (Fig. 3H). As the autophagosome marker LC3B and late endosomal marker Rab7a co-expressed, we hypothesized that the EV-SQSTM1–intracellular MAP1LC3B interaction occurs in the amphisomes, which are formed from the fusion of late endosomes with autophagosomes (Ganesan and

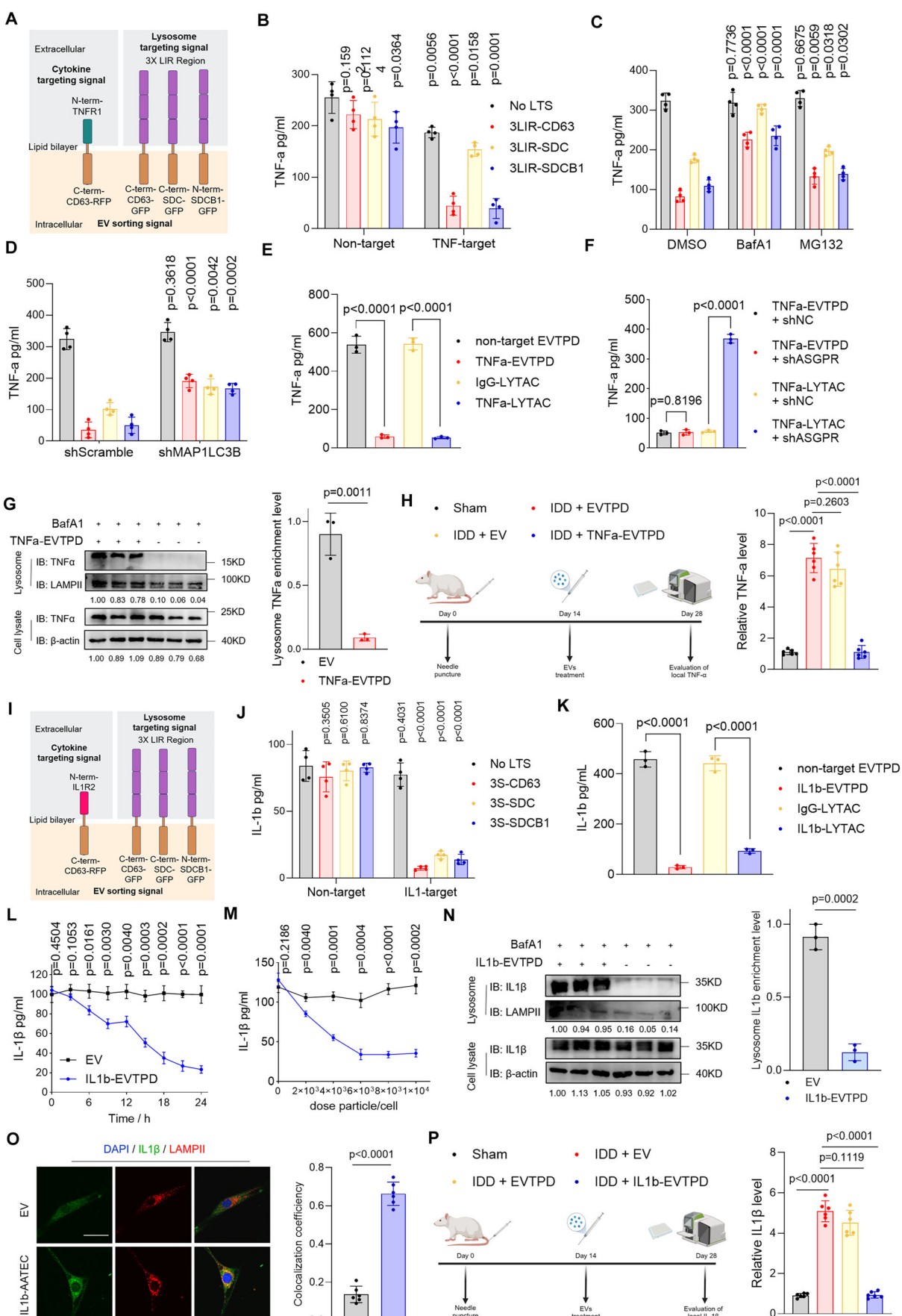

**Figure 4. EVs engineered with both degradation signals and cytokine-binding domains efficiently degrade TNF-α and IL-1β.**

(A) Schematic diagram of EVs loaded with LIR motifs and TNF-α binding domain. (B) ELISAs were used to detect the TNF-α degradation ability of EVs loaded with TNF-α binding domain and/or LIR motifs after co-incubation with NPCs for 24 h (Deg1: CD63 scaffold; Deg2: SDC scaffold; Deg3: SDCB1 scaffold) ($n = 4$). (C) ELISAs were used to detect the degradation rate of TNF-α by engineered EVs in the DMSO, BafA1, and MG132 pretreatment groups ($n = 4$). (D) TNF-α degradation level of engineered EVs was detected by ELISA in shScramble and shMAP1LC3B1-treated NPCs ($n = 4$). (E) ELISAs were used to detect the degradation rates of TNF-α in HepG2 cells by non-target EVTPD, TNFα-EVTPD, IgG-LYTAC, and TNFα-LYTAC ($n = 3$). (F) ELISAs were used to detect the degradation rates of TNF-α in HepG2 cells by TNFα-EVTPD and TNFα-LYTAC in shNC and shASGPR groups ($n = 3$). (G) A lysosome enrichment assay was used to detect the sorting efficiency of TNF-α into lysosomes after TNFα-EVTPD incubation ($n = 3$). (H) The ability of EVs, EVTPD, and rTNFα-EVTPD to degrade TNF-α in rat intervertebral discs was detected by ELISA ($n = 6$). (I) Schematic diagram of EVs loaded with LIR motifs and IL-1β binding domain. (J) ELISAs were used to detect IL-1β degradation ability of EVs loaded with IL-1β binding domain and/or LIR motifs after co-incubation with NPCs for 24 h (Deg1: CD63 scaffold; Deg2: SDC scaffold; Deg3: SDCB1 scaffold) ($n = 4$). (K) ELISAs were used to detect the degradation rates of IL-1β in HepG2 cells by non-target EVTPD, IL1β-EVTPD, IgG-LYTAC, and IL1β-LYTAC ($n = 3$). (L) ELISAs were used to detect the degradation rate of IL-1β by blank EVs and IL1β-EVTPD at 0, 3, 6, 9, 12, 15, 18, 21, 24 h ($n = 3$). (M) ELISAs were used to detect the degradation rates of IL-1β by blank EVs and IL1β-EVTPD at concentrations of 0, 2, 4, 6, 8, and 10 × $10^3$ per cell ($n = 3$). (N) Lysosome enrichment assays were used to detect the sorting efficiency of IL-1β into lysosomes after IL1β-EVTPD incubation ($n = 3$). (O) Fluorescence co-localization analysis of IL-1β and LAMPII in NPCs after EV or IL1β-EVTPD treatment for 24 h. (P) The ability of blank EVs, EVTPD, and rIL1β-EVTPD to degrade IL-1β in rat intervertebral discs was detected by ELISA ($n = 6$). Data were analyzed by unpaired two-tailed $t$ tests (B–H, J–P). Data are shown as mean ± SD. Each $n$ in (H, P) is an individual rat, $n$ in (B–G, J–O) is a biological independent sample. The $P$ values are labeled in the figure. Source data are available online for this figure.

Cai, 2021). Interfering with autophagy through ATG7 KO reduced the SQSTM1–LC3B interaction and its co-localization with Rab7 (Fig. 3I; Appendix Fig. S2f). This may be attributed to impaired amphisome formation. Overall, these findings indicate that the EVs-SQSTM1–intracellular MAP1LC3B interaction occurs in the amphisomes.

As SQSTM1/LC3B binding is essential for EV degradation, we loaded the LIR motifs onto EV scaffolds to enhance their degradation. The C-terminal of CD63, the C-terminal of Syndecan-1 (SDC1), and the N-terminal of syntenin-1 (SDCB1) are intracellular EV expression scaffolds (Gupta et al, 2021). We used nanoflow to test the loading efficacy of these scaffolds (Fig. EV4A–T). Next, we examined the degradation efficiency of the engineered EVs loaded with single or triple LIR motifs (Fig. 3J). Comparison of degradation rates showed that although the engineered EV degradation signal significantly enhance the EV degradation rate compared with that of SQSTM1 over-expression, the C-terminal CD63 fused with triple LIR motifs showed the highest degradation rates and lysosomal sorting levels (Figs. 3K and EV4U). Moreover, we generated a fusion protein comprising the L341V-mutated triple LIR sequence (previously reported LC3-binding-deficient mutation (Goode et al, 2016)) and the C-terminal of CD63 and evaluated its effect on EV degradation efficiency. Unmodified blank EVs were used as negative controls, whereas EVs carrying the wild-type triple LIR motifs were used as positive controls. The results indicated that the incorporation of the mutant triple LIR motif did not significantly enhance EV degradation efficiency compared with that of the negative control (Fig. EV4V). This finding confirmed that the triple LIR motifs promoted EV degradation by binding to LC3. We confirmed whether the degradation mechanism of LIR motif-loaded EV was dependent on ATG7 and MAP1LC3B by evaluating the EV degradation efficiency in wild-type, $ATG7^{-/-}$, and $MAP1LC3B^{-/-}$ cells. The results showed that both ATG7 KO and MAP1LC3B KO significantly impeded the degradation efficiency of the triple LIR motif-loaded EVs (Appendix Fig. S3b,c). This confirmed that the triple LIR motif-loaded EVs entered the degradation process in an autophagy-dependent and MAP1LC3B recognition-dependent manner. Thus, LIR motif-loaded EVs are efficiently sorted into lysosomes for degradation via the interaction with MAP1LC3B.

## Extracellular vesicle-based targeted degradation platform results in effective degradation of extracellular and cell surface proteins

Both TNF-α and IL-1β play crucial roles in promoting the progression of inflammation in the pathogenesis of numerous inflammatory diseases, such as IVDD (Kopf et al, 2010; Schett et al, 2013). As EVs loaded with triple LIR motifs exhibit high degradation rates, we co-loaded the cytokine receptor domains to generate an EV-based targeted protein degradation platform (EVTPD) to degrade cytokines. We fused the TNFR1-derived TNF-α binding domain with EV scaffolds. Then, triple LIR motifs fused with the EV scaffolds were co-loaded to generate TNFα-EVTPD (Fig. 4A). Nanoflow was used to assess the loading efficiency of both the signals at the single-EV level (Appendix Fig. S3e). To isolate dual-positive EVs, fluorescence-activated sorting was used based on the RFP signal fused to the C-terminal of the targeting signals and the GFP signal fused to the C-terminal of the degradation signals (as shown in Fig. 4A). The sorting efficiency was further validated using NanoFCM, which showed that the proportion of dual-positive EVs was >98% in all samples (Appendix Fig. S3e).

Next, ELISA was performed to assess the capacity of TNFα-EVTPD to degrade TNF-α. The results showed that neither the degradation signals alone (3x LIR-CD63, 3x LIR-SDC, or 3x LIR-SDCB1) nor targeting signals alone (TNFR1–CD63 fusion protein) degraded TNF-α (Fig. 4B). Only EVs with both signals significantly reduced the TNF-α level in the supernatant. In addition, among the three degradation signals, 3x LIR-CD63 and 3x LIR-SDCB1 showed better degradation ability against TNF-α (Fig. 4B). Therefore, the application of BafA1 rather than of MG132 may significantly inhibit TNF-α degradation by TNFα-EVTPD, which suggests that TNF-α degradation by EVTPD depends on normal lysosomal functioning (Fig. 4C). Furthermore, MAP1LC3B knockdown in NPCs significantly hindered the degradation ability of TNFα-EVTPD (Fig. 4D). These results confirm the essential role of MAP1LC3B in the EVTPD system.

In addition, we chose the widely used Tri-GalNAc-based LYTAC technology as a control to compare the characteristics of EVTPD in terms of targeted protein degradation ability and cell receptor dependence. To prepare LYTAC reagents for targeted

TNF-α degradation, we used Adalimumab (monoclonal antibody for TNF-α) as the basis to prepare TNFα-LYTAC. IgG-LYTAC was used as the negative control. Next, we compared the targeted protein degradation capacity of TNFα-EVTPD and TNFα-LYTAC for TNF-α in HepG2 cells. Non-target EVTPD without TNFR1 or IgG-LYTAC were used as negative controls. The results showed that both TNFα-EVTPD and TNFα-LYTAC effectively degraded extracellular TNF-α with comparable degradation capacities (Fig. 4E). Meanwhile, non-specific EVTPD and IgG-LYTAC did not exhibit the ability to target and degrade TNF-α. As the targeted degradation ability of LYTAC relies on the recognition of ASGPR, we used shRNA to knock down ASGPR in HepG2 cells and examined its influence on the targeted degradation capabilities of TNFα-EVTPD and TNFα-LYTAC towards TNFα. Western blotting analysis confirmed that ASGPR-targeted shRNA significantly decreased ASGPR expression in HepG2 cells (Appendix Fig. S1h). ASGPR knockdown led to a substantial reduction in the degradation ability of TNFα-EVTPD for TNFα compared with that in the non-specific shRNA group, whereas the degradation ability of TNFα-EVTPD remained unaffected (Fig. 4F). This finding verified the dependence of Tri-GalNAc-based LYTAC on ASGPR. However, the degradation ability of EVTPD was not affected by ASGPR expression.

As EVs loaded with triple LIR motifs fused with the C-terminal CD63 showed the highest degradation rate against TNF-α, we further examined their degradation characteristics. The time gradient assays showed that the degradation efficiency of TNFα-EVTPD was highest at ~18 h (Fig. EV5A). In addition, we investigated the concentration-dependent degradation characteristics of TNFα-EVTPD and found that the minimum concentration of $8 \times 10^3$ particles/cell was required to achieve peak degradation efficiency at 24 h (Fig. EV5B).

Given that EVTPD was efficiently sorted into lysosomes, TNF-α was also expected to be sorted into lysosomes through its binding to TNFα-EVTPD. To verify this hypothesis, we observed the intracellular distribution of TNF-α using immunofluorescence. TNFα-EVTPD group showed significantly higher sorting of TNFα into lysosome compared with that observed in blank EVs (Fig. EV5C). Moreover, TNFα-EVTPD enabled increased TNF-α lysosomal enrichment, which confirmed that TNFα-EVTPD promoted TNF-α transport to lysosomes (Fig. 4G).

We further verified the ability of TNFα-EVTPD to degrade TNF-α in vivo by constructing an rTNFα-EVTPD suitable for *Rattus norvegicus* and applied the rTNFα-EVTPD to degrade the local TNF-α in an IVDD model in SD rats. Unmodified blank EVs (EV group) and EVs modified with degradation signals but without the target signal (EVTPD group) were used as controls. After 14 days of needle puncture-induced local inflammation, we administered intra-disc injections of EV, EVTPD, and rTNFα-EVTPD. Disc samples collected at 28 days and subjected to TNF-α ELISA assays showed that TNF-α abundance was significantly increased in the IVD after injury (Freemont et al, 1997). In contrast, rTNFα-EVTPD significantly reduced the TNF-α level in the IVD compared with that observed in the EV and EVTPD groups. This confirmed the ability of rTNFα-EVTPD to degrade TNF-α in vivo (Fig. 4H). Moreover, rTNFα-EVTPD significantly improved Pfirrmann grades of IVD (Fig. EV5D), alleviated the loss of the nucleus pulposus matrix, and destroyed the annulus fibrosus and cartilaginous endplate (Fig. EV5E). Thus, rTNFα-EVTPD

showed potent TNF-α degradation and therapeutic potential for IVD in vivo.

Similar to TNF-α, IL-1β plays a crucial role in an inflammatory environment (Yang et al, 2015). Therefore, we manufactured EVs loaded with the IL-1β-binding domain and triple LIR motifs to develop IL-1β-EVTPD (Fig. 4I). The proportion of double-positive vesicles was assessed using NanoFCM (Appendix Fig. S3f). Next, we tested the targeted protein degradation capacity of IL1β-EVTPD. The ELISA assay showed that only EVs loaded with both the triple LIR motifs and IL1β-binding domains significantly reduced the IL-1β level in the cell supernatant (Fig. 4J). The CD63 scaffold exhibited the best targeted degradation performance among the three scaffolds. The application of BafA1 significantly blocked the ability of IL-1β-EVTPD to degrade IL-1β, which confirmed that IL-1β-EVTPD degraded IL-1β in a lysosome-dependent manner (Fig. EV5F). Moreover, MAP1LC3B knockdown decreased the IL1β-EVTPD degradation efficiency against IL-1β (Fig. EV5G). These results show that IL1β-EVTPD effectively mediates IL-1β degradation dependent on LC3B recognition and lysosomal function.

We compared the performance of IL1β-LYTAC and IL1β-EVTPD in targeting and degrading IL-1β in HepG2 cells. The results showed that the targeting and degradation capacity of IL1β-EVTPD was stronger than that of IL1β-LYTAC, which was different from the result observed for TNF-α (Fig. 4K). Investigating the time and dose-dependent ability of IL1β-EVTPD to degrade IL-1β showed that IL1β-EVTPD exhibited the highest degradation capacity at 21 h (Fig. 4L), whereas $6 \times 10^3$ particles/cell was the minimum concentration required to achieve the maximum IL-1β degradation rate (Fig. 4M). Both lysosome enrichment and fluorescence co-localization assays showed that IL-1β lysosomal sorting efficacy significantly increased after IL1β-EVTPD treatment (Fig. 4N,O). Notably, some IL-1β did not co-localize with lysosomes even after IL1β-EVTPD treatment. This may be because of two reasons: (1) the newly synthesized IL-1β in the cell is unaffected by EVTPD and should primarily exist in the Golgi apparatus, or (2) the IL-1β that has entered the endosome is yet to enter the lysosome, which should exist in the endosomal structure. Therefore, we observed the co-localization of IL-1β with the Golgi apparatus and endosomes, which verifies our hypothesis. These results showed that some IL-1β co-localized with the Golgi apparatus and that this co-localization was not affected by IL1β-EVTPD (Appendix Fig. S2g). This indicates that IL1β-EVTPD primarily affects IL-1β degradation rather than its generation. In contrast, IL-1β co-localization with endosomes was rarely observed without IL1β-EVTPD but significantly increased after IL1β-EVTPD treatment (Appendix Fig. S2h). This indicates that endosomes play a role in the intermediate degradation of IL-1β.

To verify the effect of IL1β-EVTPD on IL-1β degradation in vivo, we constructed an rIL1β-EVTPD and applied it to treat IVDD in SD rats. The results showed that rIL1b-EVTPD application significantly decreased IL-1β in degenerative IVDs (Fig. 4P), which suggests that rIL1b-EVTPD effectively degrades IL-1β in vivo. Moreover, rIL1b-EVTPD significantly improved the Pfirramann grades and alleviated IVDD histologically (Fig. EV5H,I).

Furthermore, we verified the ability of EVs to degrade cell surface proteins by loading the ILR1 binding domain (derived from

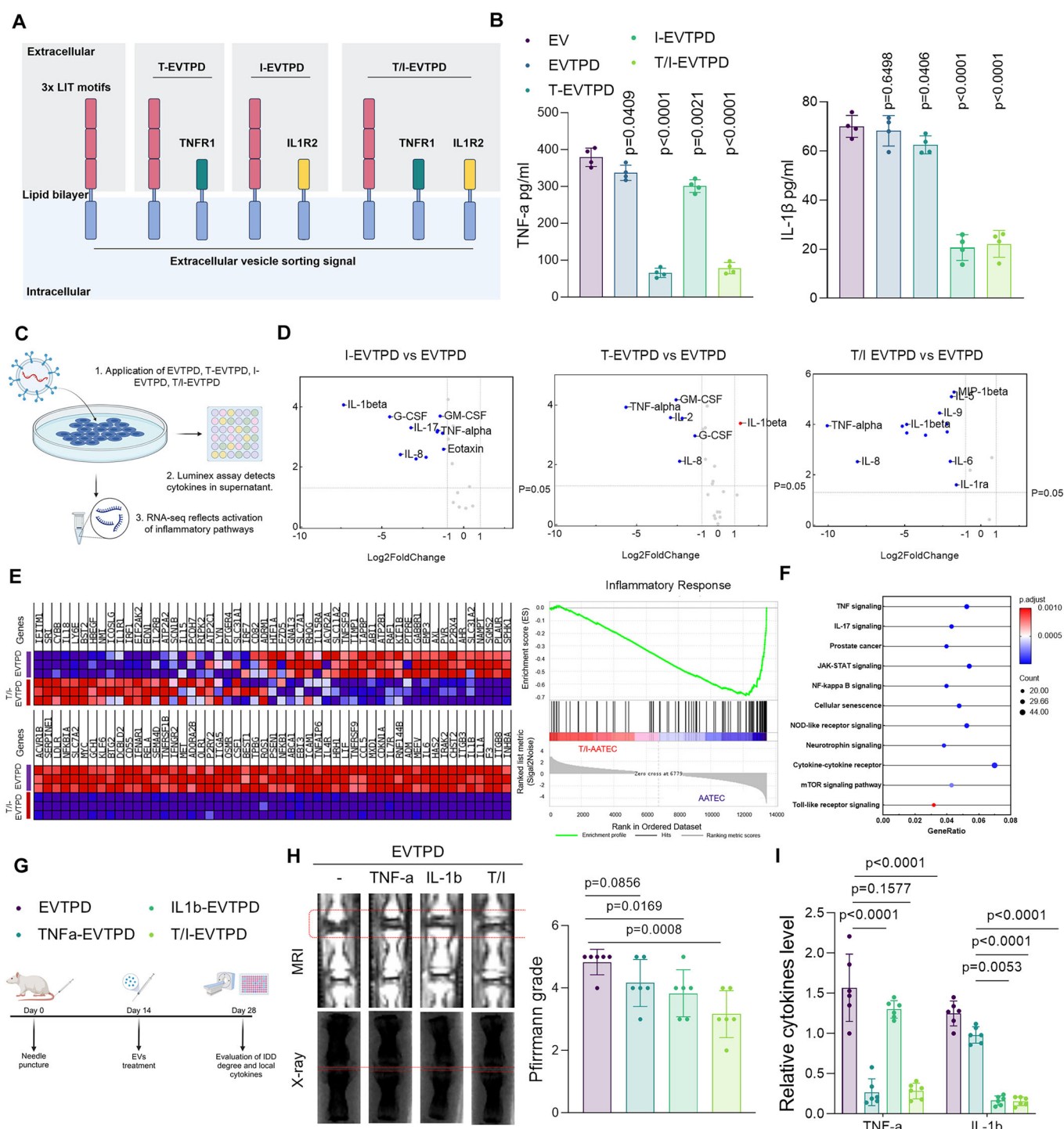

IL1Ra) and triple LIR motifs into IL1R1-EVTPD to degrade IL1R1 in human NPCs (Fig. EV5J). The results showed that IL1R1-EVTPD efficiently degraded IL1R1 (Fig. EV5K), which confirmed the ability of EVTPD to degrade cell-surface targets. Moreover, we created two control groups: one containing only IL1Ra (the recognition signal for IL1R1, i.e., the inhibitor) and the other containing only the triple LIR motifs in EVs. The results indicated

that neither of these two groups showed effective degradation of IL1R1 in NPCs (Appendix Fig. S3d). This validated the fact that the targeted degradation ability of EVTPD depended on the combined effect of both targeting and degradation signals. Overall, EVTPDs loaded with both degradation signals and binding domains effectively degrade extracellular and cell-surface proteins in a lysosome-dependent manner.

► **Figure 5. Dual-targeting EVTPD can effectively degrade both IL-1β and TNF-α and has a strong anti-inflammatory effect.**

(A) Schematic diagram of the construction of IL-1β and TNF-α dual-targeted degradation EVTPD (T/I-EVTPD). (B) ELISAs were used to detect the degradation of IL-1β and TNF-α by blank EVs, non-targeting EVTPD, IL1β-EVTPD, TNFα-EVTPD, and T/I-EVTPD after co-incubation with NPCs for 24 h ($n = 4$). (C) Schematic diagram of the investigation of the target-degradation ability of EVTPD and their effects on the transcriptional patterns of NPCs. (D) Luminex assay was used to monitor changes in cytokine levels in the supernatant of NPCs after the application of EVTPD, T-EVTPD, I-EVTPD, or T/I-EVTPD ($n = 3$). (E) GSEA of transcriptional patterns in the EV and T/I-EVTPD groups via an inflammatory response gene set ($n = 3$). (F) GO analysis of downregulated genes in the T/I-EVTPD group compared with the EVTPD group ($n = 3$). (G) Schematic illustration of EVTPD, rIL1β-EVTPD, rTNFα-EVTPD, and rT/I-EVTPD treatment in the rat IDD model. (H) MRI and X-ray imaging and quantitative analysis of intervertebral discs in the non-targeting EVTPD, rIL1β-EVTPD, rTNFα-EVTPD, and rT/I-EVTPD groups ($n = 6$). (I) ELISA assays of IL-1β and TNF-α in the intervertebral discs of the non-targeting EVTPD, rIL1β-EVTPD, rTNFα-EVTPD, and T/I-EVTPD groups ($n = 6$). Data were analyzed by unpaired two-tailed $t$ tests (B, D, H, I). Data are shown as mean ± SD. Each $n$ in (H, I) is an individual rat, $n$ in (B, D) is a biologically independent sample. The $P$ values are labeled in the figure. Source data are available online for this figure.

## EVs capable of dual-target degradation effectively degrade both IL-1β and TNF-α and show strong anti-inflammatory effects

As EVTPD effectively degrades TNF-α and IL-1β, we investigated whether EVTPD degrades both these cytokines simultaneously. EVs loaded with the triple LIR motif, TNF-α binding domain, and IL-1β binding domain (T/I-EVTPD) were constructed and examined for their ability to degrade both inflammatory cytokines (Fig. 5A; Appendix Fig. S3g). The results indicated that non-targeting EVTPD did not exhibit any notable degradative activity towards TNF-α and IL-1β. Both TNFα-EVTPD and IL1β-EVTPD significantly decreased the levels of the targeted cytokines in the supernatants. However, their capacity to degrade non-target cytokines was relatively low (Fig. 5B). Only T/I-EVTPD degraded both cytokines in the supernatant with high efficacy (Fig. 5B), which suggests that EVTPD may potentially act as a multiple-target degradation platform.

We hypothesized that as the simultaneous expression of TNFR1 and IL1R2 on the same EVs endows each EV with additional degrading capabilities such as the ability to degrade both TNF-α and IL-1β simultaneously, the targeted degradation ability of T/I-EVTPD is the sum of that of T-EVTPD and I-EVTPD at the same dose. We verified this hypothesis by performing an in vitro experiment. ELISA results showed that the targeted degradation effect of T/I-EVTPD was significantly better than that of the mixed pool of T-EVTPD and I-EVTPD at the same dose (Appendix Fig. S3h), which may be attributed to the difference in the loading quantities of TNFR1 and IL1R2 on each EV.

Next, we used Luminex technology to assess EVTPD specificity for its degradation targets. Additionally, we used RNA-seq to elucidate the remodeling of transcription patterns in NPCs following T/I-EVTPD administration (Fig. 5C). IL1β-EVTPD showed effective degradation of IL-1β compared with that observed for EVs. However, IL17, IL8, and TNF-α levels decreased along with that of IL1β. This may be attributed to the alleviation of the inflammatory environment by IL1β-EVTPD (Fig. 5D). TNFα-EVTPD effectively degraded TNF-α along with a synergistic reduction in IL2 and IL8 levels (Fig. 5D). Thus, T/I-EVTPD effectively degrades both TNF-α and IL-1β and causes a potent synergistic reduction of the levels of other inflammatory cytokines, including IL8, IL9, IL5, and IL6 (Fig. 5D). RNA-seq assays showed that various inflammation-related genes including *IL1A, IL1B, IL6, TNFRSD9*, and *TNFAIP6* were significantly downregulated after T/I-EVTPD treatment, whereas GSEA showed that the inflammatory response gene set was significantly enriched in the blank EV

treatment group. This suggests that T/I-EVTPD effectively reversed the inflammatory response pattern of NPCs (Fig. 5E). GO analysis showed that the downregulated genes in the T/I-EVTPD-treated group were enriched in various inflammatory pathways, including TNF signaling, IL-17 signaling, and NF-kB signaling (Fig. 5F), which indicates that T/I-EVTPD significantly inhibited multiple inflammation-related pathways that were activated in the NPCs. Real-time PCR assays showed that T/I-EVTPD additionally improved the production of extracellular matrix (ECM) such as collagen II and aggrecan and inhibited the expression of both inflammatory cytokines (IL1β and IL6) and matrix metalloproteinases (MMP3 and MMP14) (Appendix Fig. S4a). Immunoblotting assays showed reduced phosphorylation of p65, which confirmed that T/I-EVTPD significantly inhibited the activation of NF-kB signaling under inflammatory conditions (Appendix Fig. S4b).

We further investigated the effect of T/I-EVTPD on the inflammatory process in vivo by applying T/I-EVTPD for the treatment of IVDD in SD rats. After 2 weeks of needle puncture, blank EVs, IL1β-EVTPD, TNFα-EVTPD, and T/I-EVTPD were used to treat IVDD for 2 weeks. Subsequent imaging and histological analyses (Fig. 5G) showed that the single-target EVs mitigated IVDD and that T/I-EVTPD exhibited more potent anti-inflammatory and regenerative effects than those of single-target EVs (Fig. 5H; Appendix Fig. S4c). Micro-CT showed that the vertebral bone surfaces were severely damaged and worn in the blank EV group, whereas treatment with IL1β-EVTPD and TNFα-EVTPD exerted significant therapeutic effects on the vertebral bone surfaces (Appendix Fig. S4d). Moreover, T/I-EVTPD showed a more robust reparative effect than those of single-target degradation-EVs. The upregulation of collagen II and downregulation of MMP14 in the T/I-EVTPD group additionally validated their regenerative effect (Appendix Fig. S4e). ELISA assays showed that TNFα-EVTPD effectively degraded TNF-α, whereas IL1β-EVTPD significantly reduced the IL-1β level in the IVDs (Fig. 5I). Moreover, T/I-EVTPD showed dual degradation of both TNF-α and IL-1β. Histological staining showed that TNFα-EVTPD or IL1β-EVTPD application significantly alleviated the loss of the nucleus pulposus matrix, tears of the annulus fibrosus, and destruction of the cartilage endplate, whereas the regenerative ability of T/I-EVTPD was higher than that of single-target EVTPD (Appendix Fig. S4f–h).

As the decoy EVs that are only equipped with the targeting domain elicit anti-inflammatory effects (Gupta et al, 2021; Keller et al, 2020), we compared the therapeutic efficacy of cytokine-neutralizing agents against those of EV-based targeted degradation strategies. We performed comparative experiments using both

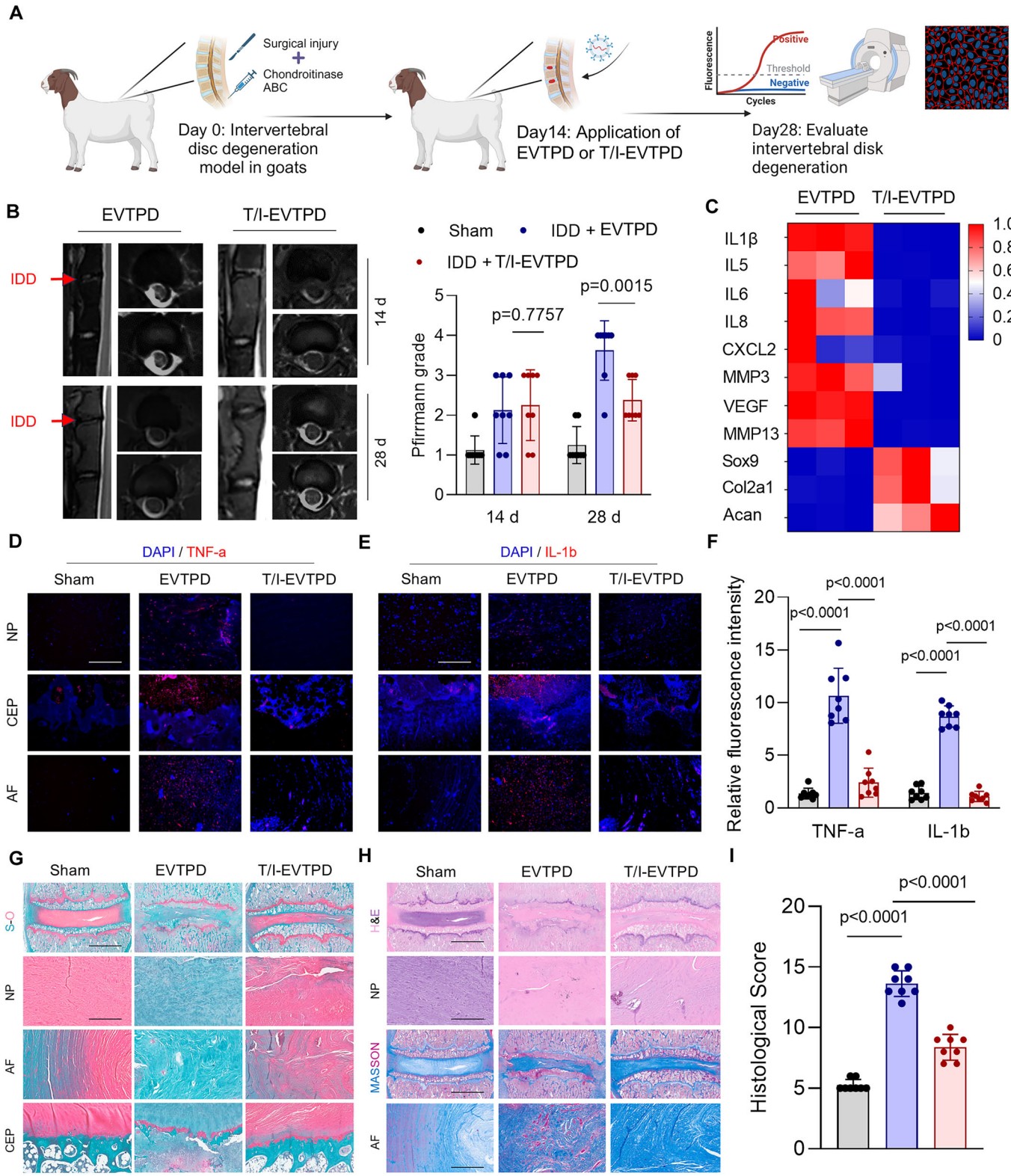

**Figure 6.   T/I-EVTPD displayed robust anti-inflammatory and regenerative effects in a goat intervertebral disc degeneration model.**

(A) Schematic diagram of goat intervertebral disc degeneration modeling and treatment with non-targeting EVTPD and gT/I-EVTPD groups. (B) MRI images and quantitative analysis of sham and surgery-performing intervertebral discs on days 14 and 28 ($n = 8$). (C) RT–qPCR was used to detect the transcription of inflammatory genes in nucleus pulposus tissues in the non-targeting EVTPD and gT/I-EVTPD groups after treatment for 14 days ($n = 3$). (D–F) Immunofluorescence was used to detect the expression of TNF-α and IL-1β in the intervertebral discs of the sham, non-targeting EVTPD, and gT/I-EVTPD groups ($n = 8$) (scale bar 50 μm). (G–I) Histological images and quantitative analysis of intervertebral discs stained with safranin and fast green (G) and eosin and hematoxylin (H) in the sham, non-targeting EVTPD and gT/I-EVTPD groups after treatment for 14 days ($n = 8$) (scale bar: 500 μm or 50 μm). Data are analyzed by unpaired two-tailed $t$ tests (B, F, I). Data are shown as mean ± SD. Each $n$ in (B, C, F, I) is an individual goat. The $P$ values are labeled in the figure. Source data are available online for this figure.

in vitro and in vivo IVDD models. EVs co-loaded with TNFR1 and IL1R2 were used as the neutralizing agent control group (T/I-EV), whereas EVs co-loaded with both TNFR1/IL1R2 targeting signals and triple LIR motifs constituted the targeted degradation group (T/I-EVTPD). In addition, we mixed EVs co-loaded with TNFR1 and IL1R2 with EVs carrying only the triple LIR motifs in a 1:1 ratio (T/I-EV + EVTPD) to assess whether targeting and degradation signals function independently. The results showed that T/I-EVTPD exerted a potent anti-inflammatory effect, whereas T/I-EV exhibited a relatively mild regenerative effect by promoting ECM production and inhibiting MMP and inflammatory cytokine production (Appendix Fig. S5a). Furthermore, combining T/I-EV and EVTPD did not significantly enhance the regenerative capacity of T/I-EVs (Appendix Fig. S5a), which shows that T/I-EVTPD plays a more effective anti-inflammatory role than that of decoy EVs by simultaneously trapping and degrading targeted cytokines. Additionally, the T/I-EVTPD group showed the lowest NF-kB pathway activation level (Appendix Fig. S5b). Next, we applied rT/I-EV and a mixture of rT/I-EVs and EVTPD to treat IVDD in vivo. The MRI images showed that both rT/I-EVs and the rT/I-EV-EVTPD mixture exerted a mild alleviating effect on IVD water loss, whereas T/I-EVTPD showed a relatively stronger therapeutic effect (Appendix Fig. S5c). Histological assays including haematoxylin–eosin (H&E), Safranin O-Fast Green (S-O), and Masson staining of the IVD showed that T/I-EVTPD exhibited a higher regenerative capacity than those of both T/I-EVs and T/I-EVs-EVTPD mixture(Appendix Fig. S5d–f). This confirms that T/I-EVTPD exert a more potent anti-inflammatory effect than that of the decoy EVs. Overall, T/I-EVTPD effectively degraded both TNF-α and IL-1β, which resulted in potent anti-inflammatory effects.

## T/I-EVTPD showed robust anti-inflammatory and regenerative effects in the goat intervertebral disc degeneration model

We validated the clinical translational potential of T/I-EVTPD in a large animal model by using T/I-EVTPD to treat IVDD in goats. First, we constructed gT/I-EVTPD to target both TNF-α and IL-1β in *Capra hircus* and verified their targeted degradation ability in vitro (Appendix Fig. S6a). ELISA showed that EVTPD effectively degraded target cytokines, whereas gT/I-EVTPD showed dual targeting and degradation abilities against TNF-α and IL-1β (Appendix Fig. S6b). gT/I-EVTPD reduced the expression of inflammatory genes and promoted ECM synthesis in goat NPCs (Appendix Fig. S6c). In addition, gT/I-EVTPD promoted NPC proliferation and viability (Appendix Fig. S6d,e), which indicated that gT/I- EVTPD effectively degrades both TNF-α and IL-1β to regenerate NPCs in vitro.

We investigated the therapeutic effects of gT/I-EVTPD on IVDD in goats in vivo. An IVDD goat model was established by inflicting surgical injury combined with local chondroitinase ABC injection (Gullbrand et al, 2017). Non-targeted EVTPD or gT/I-EVTPD was locally injected to treat IVDD 2 weeks later, and the therapeutic effects were observed after 2 weeks of treatment (Fig. 6A). MRI showed almost no water signal in the IVDs 14 days after surgery compared with that observed in the sham-treated IVDs, which showed severe IVDD induced by surgery and chondroitinase (Fig. 6B). gTI-EVTPD application significantly alleviated the degenerative process in surgically treated IVDs compared with that in the non-targeted EVTPD group after 14 days of treatment (Fig. 6B). RT-qPCR analysis of the nucleus pulposus tissues showed that inflammatory cytokines and MMPs were downregulated in the gT/I-EVTPD group, whereas ECM synthesis was enhanced. This suggests that gT/I-EVTPD inhibits the progression of inflammation and restores the function of NPCs by degrading TNF-α and IL-β (Fig. 6C). Immunofluorescence imaging showed significant accumulation of TNF-α and IL-β in the IVDs (especially in the cartilage endplate) in the non-targeted EVTPD group (Fig. 6D,E). In contrast, gT/I-EVTPD effectively degraded both TNF-α and IL-β in the IVDs (Fig. 6F). This confirmed the robust ability of gT/I-EVTPD to degrade TNF-α and IL-β in the goat IVDD model. S-O staining showed that gT/I-EVTPD significantly delayed the loss of collagen in the IVDs, whereas HE and Masson staining showed that gT/I-EVTPD alleviated ECM loss in the nucleus pulposus and destruction of the annulus fibrosus and cartilage endplate (Fig. 6G–I). Thus, gT/I-EVTPD showed robust anti-inflammatory and regenerative effects in the goat IVDD model.

## Discussion

Protein degradation in eukaryotic cells primarily relies on the ubiquitin–proteasome system (UPS) and lysosomal system (Riching et al, 2022). As an emerging treatment method, TPD mediates the degradation of undruggable targets via the degradation system already present in the cell (Békés et al, 2022; Schapira et al, 2019). The UPS was first used for TPD via the PROTAC technology (Sakamoto et al, 2001), and this system exhibited excellent degradation ability for intracellular undruggable proteins (Liu et al, 2020). However, because the PROTAC-mediated degradation process was based on the UPS, their targeting range was limited to intracellular proteins (Neklesa et al, 2017). Moreover, significant issues such as biocompatibility and route of administration of this system as a small-molecule drug hindered its clinical administration (Nieto-Jiménez et al, 2022). To overcome these limitations,

AbTACs, LYTACs, and KineTACs were developed to effectively degrade both membrane and extracellular proteins using lysosome-targeted plasma membrane receptors (Wells and Kumru, 2024). However, these extracellular targeting protein degradation (eTPD) techniques required the expression of specific cell membrane receptors such as CI-M6PR, RNF43, and CXCR7, which limited their application in several organs of the body. Moreover, a single target is ineffective in addressing the occurrence of multiple pathological processes, and interventions targeting multiple targets are often more effective in these cases (Ciardiello et al, 2006). However, these technologies involved small-molecule drugs that couple ligands with degradation signals; hence, degrading two or more extracellular targets was difficult. In the present study, we developed a modular EVTPD to overcome these technological drawbacks. Mechanistically, EVTPD identified intracellular MAP1LC3B through SQSTM1-derived LIR motifs that mediate the lysosome-dependent degradation of extracellular targets. Moreover, EVTPD simultaneously degraded multiple targets through the various display scaffolds engineered on the surface of the EVs (Richter et al, 2021). By co-loading TNFR and ILR2 recognition domains, this EVTPD system simultaneously degraded both extracellular TNF-α and IL-β, which are two of the most important proinflammatory cytokines, and attenuated IVDD in rats and goats. Moreover, EVTPD extended the scope of application of eTPD to multiple organs without limiting the expression of cell surface receptors.

EVs are emerging nanovesicles derived from the cell supernatant and have been widely used to treat inflammatory, autoimmune, tumor, and neurodegenerative diseases (Clancy and D'Souza-Schorey, 2023; Kim et al, 2022; Turpin et al, 2016; Yang et al, 2024). EVs regulate various biological processes in target cells by delivering a variety of bioactive molecules such as miRNAs, lncRNAs, and proteins to their parent cells (Garcia-Martin et al, 2022; Isaac et al, 2021; O'Brien et al, 2020). Moreover, EVs are cell-derived carriers with excellent biocompatibility. Hence, EVs have been widely used to deliver various active molecules in vivo (de Jong et al, 2019; Maas et al, 2017). As EVs are primarily used to deliver functional molecules, most studies have focused on the process of loading and delivering EV cargo. Hence, few studies have discussed the EV degradation process (Dixson et al, 2023; van Niel et al, 2018). Two problems are associated with studying EV degradation in cells. The first is the lack of a labeling system to accurately monitor the localization and degradation of EVs (Chuo et al, 2018). Second, the high homology of EV proteins to their source cells causes difficulty in distinguishing the EVs from intracellular components to identify their interactions after entering the cell (Bao et al, 2023). In this study, we constructed GFP-loaded EVs and monitored their intracellular localization and degradation by analyzing the GFP signals. Thus, we found that EVs were degraded via an autophagy-mediated lysosomal pathway. To investigate the proteomic changes in EVs before and after degradation, we used biotin-labeling technology to biotinylate the EV surface proteins and analyzed the differences in EV protein composition before and after degradation using streptavidin enrichment. We hypothesized that the interaction between EVs and intracellular proteins mediates the intracellular EV sorting process. To validate this, we chose to use EVs derived from the same species (for example, HEK293T cells) to preserve their interaction with intracellular proteins in human NPCs. A non-

biotin-labeled EV group was included to eliminate false-positive signals. The mass spectrometry results showed that the degradation of several EV proteins (such as SQSTM1) was inhibited by BafA1, which concurs with our hypothesis and indicates that the lysosome-dependent degradation pathway could degrade these specific EV subgroups. However, BafA1 did not inhibit the degradation of certain EV proteins, which may be attributed to the fact that the degradation of these specific EV subgroups does not rely on the BafA1-dependent pathway or that these EVs are resistant to the lysosomal degradation system. Notably, the attenuation of a small number of EV proteins was enhanced after BafA1 inhibition. We speculate that this unusual activation may be caused by the activation of the endosomal recycling system after the inhibition of lysosomal degradation, leading to the exocytosis of these EVs (Risbud and Shapiro, 2014). Future studies are warranted to confirm these inferences.

In addition, the in vivo experiments helped confirm that the duration for the natural degradation of EVs in the local IVD is 3 weeks, which is much longer than the 24 h observed for degradation in vitro. The reason may be attributed to the number of EVs used in the in vivo experiment ($1 \times 10^{10}$ particles/mL), which is much higher than that used in the in vitro experiment ($1 \times 10^8$ particles/mL). In addition, the internal microenvironment of the IVD is relatively independent and avascular with low metabolic activity and a small number of NPCs (Risbud and Shapiro, 2014b). These factors may have contributed to the slower in vivo EV degradation rate in the IVDs. The in vivo experiments show that SQSTM1 overexpression significantly promotes EV degradation in vivo. This effect was evident as early as day 5 and persisted until day 15 when most of EVs in both groups were degraded. This indicates that SQSTM1 plays an important role in EV degradation in vivo. However, SQSTM1 overexpression did not significantly alter the EV degradation trend. This may be related to the limited metabolic activity of cells in the IVD and the limitation of autophagic flux (Risbud and Shapiro, 2014b).

Furthermore, we found that SQSTM1 mediated EV sorting into lysosomes by recognizing MAP1LC3B. We chose MAP1LC3 from the potential SQSTM1-interacting protein spectrum (identified via IP-MS) primarily because of two key considerations: (1) MAP1LC3 exhibited the highest level of enrichment among proteins in the IP-MS analysis only next to that of SQSTM1, which was used as the bait protein. This suggests that MAP1LC3 is a major interacting partner of SQSTM1; (2) as autophagy plays a critical role in EV degradation (Fig. 1) and as MAP1LC3 functions as an adapter protein in autophagy by facilitating the delivery of cargo into the autophagic pathway (Li et al, 2020), MAP1LC3 was considered a plausible candidate for mediating EV entry into the autophagy-dependent degradation pathway. However, we did not perform further experimental validation to determine whether other identified proteins are involved in EV degradation, and their potential roles need to be elucidated in future studies. As the SQSTM1–MAP1LC3B interaction plays an important role in mediating the degradation of EVs, we engineered SQSTM1-derived LIR motifs on the EV surface to act as a degradation signal to increase degradation efficacy. We found that EVs loaded with degradation signals and binding domains for protein targets entered the lysosomal pathway efficiently and degraded the target proteins with high selectivity. Moreover, EVs loaded with both TNF-α and IL-β binding domains degraded both inflammatory

factors simultaneously, which indicates that EVTPD may be used as a dual-target protein degradation platform, which is a novel breakthrough in the field of TPD.

As the leading cause of disability, lower back pain imposes a substantial disease burden worldwide. IVDD is a major cause of low back pain. Proinflammatory cytokines—especially TNF-α and IL-1β—play important roles in IVDD development (Risbud and Shapiro, 2014; Wang et al, 2020). These proinflammatory factors not only recruit immune cells such as macrophages and trigger local angiogenesis and inflammation but also stimulate peripheral nerves to cause discogenic pain (Wang et al, 2020). Moreover, these factors disrupt normal metabolic processes in NPCs and annulus fibrosus cells, leading to ECM loss and IVD degeneration. Antibodies and inhibitors against TNF-α and IL-β have shown therapeutic effects on IVDD in clinical trials. However, a TPD technique targeting proinflammatory cytokines has yet to be applied to IVDD. Hence, we developed an EVTPD based on the degradation properties of EVs to degrade TNF-α and IL-β for the treatment of IVDD. In this study, EVTPD exhibited highly efficient and selective degradation of TNF-α and IL-β in vitro. EVTPD application in SD rats and goats with IVDD confirmed the anti-inflammatory and regenerative effects with promising clinical translation prospects. Furthermore, the good biocompatibility and circulatory characteristics of EVs could enable EVTPD to accelerate the clinical translation of TPD technology and expand its application to organs in systemic diseases.

# Methods

### Reagents and tools table

| Reagent/resource | Reference or source | Identifier or catalog number |
|---|---|---|
| **Experimental models** | | |
| Sprague-Dawley rats | Charles River Laboratories | CD®(SD) IGS |
| Boer goat | Chengde Zhaonong Laboratory Animal Breeding Co., Ltd. | 130083240100000371 |
| **Recombinant DNA** | | |
| lentiCRISPRv2 | Addgene | 52961 |
| scramble shRNA | Addgene | 1864 |
| **Antibodies** | | |
| GFP tag Polyclonal antibody | Proteintech | 50430-2-AP |
| Alix Monoclonal antibody | Proteintech | 67715-1-Ig |
| CD9 Monoclonal antibody | Proteintech | 60232-1-Ig |
| TSG101 Polyclonal antibody | Proteintech | 28283-1-AP |
| CD63 Polyclonal antibody | Proteintech | 25682-1-AP |
| LAMP2 Monoclonal antibody | Abcam | ab25631 |

| Reagent/resource | Reference or source | Identifier or catalog number |
|---|---|---|
| CD71 Polyclonal antibody | Proteintech | 10084-2-AP |
| SQSTM1 Polyclonal antibody | Proteintech | 18420-1-AP |
| TMEM59 Polyclonal antibody | Proteintech | 24134-1-AP |
| CCN1 Polyclonal antibody | Proteintech | 26689-1-AP |
| ITM2B Polyclonal antibody | Signalway Antibody | 47141 |
| ITM2C Polyclonal antibody | Signalway Antibody | 39725 |
| LC3B-Specific Polyclonal antibody | Proteintech | 18725-1-AP |
| DYKDDDDK tag Monoclonal antibody | Proteintech | 66008-4-Ig |
| His-Tag Monoclonal antibody | Proteintech | 66005-1-Ig |
| HA-Tag Monoclonal antibody | Proteintech | 66006-2-Ig |
| GST-Tag Monoclonal antibody | Proteintech | 66001-2-Ig |
| TNF-a Monoclonal antibody | Abcam | ab1793 |
| IL-1b Monoclonal antibody | Abcam | ab156791 |
| Collagen Type II Polyclonal antibody | Proteintech | 28459-1-AP |
| MMP3 Polyclonal antibody | Proteintech | 17873-1-AP |
| **Oligonucleotides and other sequence-based reagents** | | |
| shRNA | This study | Table EV1 |
| qRT-PCR primers | This study | Table EV2 |
| **Chemicals, enzymes, and other reagents** | | |
| MG132 | MedChemExpress | HY-13259 |
| Bafilomycin A1 | MedChemExpress | HY-100558 |
| 3-Methyladenine | MedChemExpress | HY-19312 |
| ML282 | MedChemExpress | HY-13452 |
| DMEM medium | Gibco | 11965092 |
| Fetal Bovine Serum | Cytiva | SH30088.03HI |
| Lysosome Isolation Kit | Invent | LY-034 |
| NHS-sulfo-biotin | APEXBIO | A8001 |
| Protein A/G Magnetic Beads | MedChemExpress | HY-K0202 |
| IPTG | MedChemExpress | HY-15921 |
| GST-pull down Kit | Beyotime | P2262 |

| Reagent/resource | Reference or source | Identifier or catalog number |
|---|---|---|
| Proximity Ligation Assay Kit | Naveni | 60025 |
| Lipo2000 | Thermo Fisher | 11668030 |
| TRIzol kit | Thermo Fisher | 15596018CN |
| PKH26 | Sigma | PKH26GL |
| Software | | |
| ImageJ | National Institutes of Health | |
| GraphPad Prism 9 | GraphPad | |

## Ethical approval

The acquisition of human specimens used in this study was approved by the Ethics Committee of Tongji Medical College, Huazhong University of Science and Technology (2021 IEC 075). The volunteers were fully informed of the experimental purposes and provided signed informed consent. Animal experiments involving SD rats and Boer goats were approved by the Ethics Committee of the Huazhong University of Science and Technology (2022 IACUC 4035).

## Cell culture, transfection

Nucleus pulposus samples were obtained from patients who were about to undergo scoliosis correction surgery with informed consent and no harm to the patient's interests. Tissues were digested using 0.4% collagenase type II (Sigma, USA) for 6 h at 37 °C. The digested cells were separated through centrifugation at $300 \times g$ for 5 min, washed with PBS buffer (Gibco, USA), and cultured in F12 complete medium (Gibco) supplemented with 10% fetal bovine serum (Gibco) and 1% penicillin/streptomycin (Gibco). NPCs from passages 2 to 5 were used for further experiments. The extraction process and culture conditions for goat NPCs were the same as those of human NPCs.

Bone marrow blood specimens were obtained from patients who were about to undergo scoliosis correction surgery with informed consent and no harm to the patient's interests. The BMSC isolation kit (TBDscience, China) was used to isolate BMSC from bone marrow blood specimens according to the manufacturer's instructions. After isolation, the BMSCs were cultured in DMEM (Gibco, USA) containing 10% FBS (Gibco, USA) and 1% penicillin–streptomycin (Gibco, USA). Osteogenic, chondrogenic, and adipogenic differentiation assays were performed to determine the differentiation potential of BMSCS. BMSCS from passages 2 to 5 were used for further experiments.

HeLa and HEK293T cells were obtained from ATCC. Both cell lines were cultured in DMEM (Gibco) containing 10% FBS (Gibco) and 1% penicillin–streptomycin (Gibco). All cells were maintained at 37 °C under 5% $CO_2$ in a $CO_2$ incubator (Thermo Fisher, USA).

For plasmid transfection, cells were seeded on day 1, transfected with Lipo2000 (Thermo Fisher, USA) on day 2 according to the manufacturer's instructions, and tested for transfection efficiency for further experiments on day 4. The amino acid sequences used for EV engineering were presented in Table EV4.

## Drug treatment

MG132: MG132 has been reported as a proteasome inhibitor that can specifically inhibit proteasome-dependent protein degradation at a concentration of 20 μM (Lee and Goldberg, 1998). We stimulated the NPCs with 20 μM MG132 for 12 h and verified the inhibitory effect of MG132 on proteasome-mediated protein degradation by detecting the total protein ubiquitination level. The results showed that 20 μM MG132 effectively inhibited the protein degradation mediated by the ubiquitin-proteasome pathway (Appendix Fig. S1a).

Bafilomycin A1 (BafA1): BafA1 has been reported to inhibit lysosomal acidification and degradation by selectively inhibiting vacuolar $H^+$-ATPase at a concentration of 50 nM (Gagliardi et al, 1999). We stimulated NPCs with 50 nM BafA1 for 12 h and used the autophagy reporter probe GFP-RFP-LC3 to assess lysosomal function. When lysosomal acidification was impaired, GFP fluorescence increased. Stimulation with 50 nM BafA1 significantly inhibited the lysosomal degradation (Appendix Fig. S1b).

3-Methyladenine (3-MA): 3-MA specifically inhibits the autophagy/lysosome degradation pathway at an effective concentration of 5 mM (Seglen and Gordon, 1982). We stimulated NPCs with 5 mM 3-MA for 12 h and used the autophagy reporter probe GFP-RFP-LC3 to indicate autophagy activity. The results showed that 3-MA effectively inhibited autophagy in the NPCs (Appendix Fig. S1c).

ML282 is a highly effective small-molecule pan-inhibitor targeting the Ras superfamily of GTPases and has a significant inhibitory effect on Rab7 (Hong et al, 2010). We referred to previous reports and treated the cells with 20 μm ML282 for 12 h (Zhang et al, 2024). Using Rab7a GTPase-RILP, we detected the proportion of active Rab7a and assessed the effectiveness of the stimulation. The results showed that ML282 effectively inhibits Rab7a activity (Appendix Fig. S1d).

## Western blotting assay

Samples were lysed using lysis buffer (20 mM Tris, 150 mM NaCl, 1% Triton X-100, and 1 mM EDTA (pH 7.2)) containing 1% PMSF (Boster, USA) for 30 min on ice and sonicated using an ultrasonic cell splicer five times (8 s each). Next, the samples were centrifuged at $12{,}000 \times g$ at 4 °C for 15 min. The protein supernatants were collected to detect protein concentrations using a BCA Protein Quantification Kit (Boster, USA). Loading buffer (Boster, USA) was added to protein supernatants for denaturing at 95 °C for 10 min before loading. In total, 20 μg of protein was loaded into each sample well on 6–12% SDS-PAGE gels. Electrophoresis was performed at 80 V for 30 min, followed by 120 V for 50 min, and transmembrane at 300 mA for 2 h on ice. The PVDF membranes (Millipore, USA) were blocked in TBS/0.1% Tween (TBS/T) with 5% skim milk at room temperature for 2 h, followed by incubation with the corresponding primary antibody at 4 °C overnight. After elution with TBS/T, the secondary antibody was incubated at room temperature for 2 h and eluted again with TBS/T before visualization using a fully automated imaging system (Bio-Rad, Hercules, CA, USA) with an ECL Kit (Affinity Biosciences). Signal intensities were quantified using ImageJ Fiji software (National Institutes of Health, USA).

## Immunofluorescence

Cells grown on glass coverslips were fixed and permeabilized in methanol at $-20\,°C$ for 15 min. Then, the cells were washed thrice with PBS for 15 min. The slides were blocked with QuickBlock Buffer (Beyotime, China) for 20 min at room temperature. After blocking, the primary antibodies diluted with PBS/0.2% BSA solution were added to cover the sample areas and incubated at $37\,°C$ for 60 min. The slides were washed thrice with TBS/T for 15 min. Secondary antibodies diluted with PBS/0.2% BSA solution were added to cover the sample areas and incubated at $37\,°C$ for 60 min. The slides were washed thrice with TBS/T for 15 min. The nuclei were stained with DAPI (Beyotime, China) at $37\,°C$ for 5 min. The slides were washed thrice with TBS/T and sealed with an anti-fluorescence quenching agent (Beyotime, China). Finally, the slides were imaged using a Confocal Laser Scanning Microscope (CLSM) FV3000 (Olympus, Japan).

## Extracellular vesicles isolation and characterization

Cells prepared for EV extraction were cultured in DMEM (Gibco, USA) with 10% exosome-depleted fetal bovine serum (Gibco, USA), and cell supernatants were collected for EV extraction 48 h after transfection. For the production of EVs, we used an automatic EV extraction system (EXODUS H600) that automatically isolated EVs from a large quantity of cell supernatant at a maximum speed of 200 mL/h, which met the requirements for large-scale preparation of engineered EVs (Chen et al, 2021). Transmission electron microscopy (TEM), nanoparticle tracking analysis (NTA), western blotting, and nanoflow (Flow NanoAnalyzer, NanoFCM) were used to characterize the EVs. The particle number of EVs was determined using Nanoflow for subsequent experiments. We used fluorescence-activated sorting technology (CytoFLEX SRT) to isolate the engineered receptor-positive EVs.

## Extracellular vesicle degradation assay

EGFP-EVs were added to recipient cells at a concentration of $5 \times 10^3$ particles/cell and incubated for 12 h for uptake. Supernatants containing free EGFP-EVs were removed, and the cells were washed twice with PBS, followed by lysing to measure cellular uptake levels after the proteins were extracted. However, the degradation groups were incubated in fresh complete medium for another 12 h before cell lysis and protein extraction. EGFP levels were examined using western blotting and quantified using ImageJ (National Institutes of Health, USA). The resident GFP levels were calculated using the following formula:

$$Resident\ GFP\ level = GFP\ level_{degradation} \div GFP\ level_{uptake}$$

## Extracellular vesicle lysosomal sorting assay

EGFP-EVs were added to recipient cells at a concentration of $5 \times 10^3$ particles/cell and incubated for 24 h with Bafilomycin A1 (MCE, USA) at a concentration of 50 nM. After incubation, lysosomes were isolated from the whole-cell lysates for protein extraction using a Lysosome Isolation Kit (Invitrogen, USA). The cells in the same group were lysed for protein extraction as whole-cell controls. EGFP levels were examined using western blotting and quantified using the ImageJ software (NIH, USA). The GFP lysosomal enrichment was calculated using the following formula:

$$GFP\ lysosomal\ enrichment\ level = GFP\ level_{lysosome} \div GFP\ level_{cell\ lysate}$$

## Extracellular proteomic detection

### Sample preparation

EVs were labeled in vitro with NHS-sulfo-biotin and incubated with NPCs for 24 h. At the end of the incubation period, the cells were lysed with NP40 lysis buffer (50 mM Tris-HCl, 150 mM NaCl, 1 mM EDTA, and 1% NP-40 [pH 7.4]) on ice for 30 min and centrifuged at $12,000 \times g$ for 15 min to extract the proteins. The biotinylated proteins were enriched in the protein solution using streptavidin magnetic beads to obtain EV-derived proteins. EV proteomics was performed using label-free quantification technology. Briefly, reaction solution (1% SDC, 100 mM Tris-HCl [pH = 8.5], 10 mM TCEP, and 40 mM CAA) was added to the sample and incubated at $95\,°C$ for protein denaturation, reduction, and alkylation. Trypsin was added at a ratio of 1:50 (enzyme: protein, w/w) for digestion overnight at $37\,°C$. The next day, TFA was added to bring the pH down to 6.0 to terminate the digestion, and the supernatant was centrifuged at $12,000 \times g$ for 15 min. The peptide was purified using a custom-made SDB desalting column. The eluate was vacuum dried and stored at $-20\,°C$ for later use.

### LC-MS/MS detection

All samples were analyzed using a TimsTOF Pro (Bruker Daltonics, USA). An Ultimate 3000 RSLC Nano system (Thermo Scientific, USA) was coupled to a timsTOF Pro with a CaptiveSpray nanoion source (Bruker Daltonics, USA). Peptide samples were trapped on a C18 Trap column (75 μm × 2 cm, 3 μm particle size, 100 Å pore size, Thermo) and separated in a reversed-phase C18 analytical column (75 μm × 15 cm, 1.7 μm particle size, 100 Å pore size, IonOpticks). Mobile phases A (0.1% formic acid in water) and B (0.1% formic acid in ACN) were used to establish a separation gradient at a flow rate of 300 nL/min. MS data acquisition was performed in PASEF mode. The capillary voltage was set at 1500 V. MS and MS/MS spectra were acquired at 100–1700 $m/z$. The ion mobility was scanned from 0.75 of 1.4 Vs/cm². The accumulation and ramp times were set to 100 ms. The acquisition cycle of 1.16 s comprised one full MS scan and 10 PASEF MS/MS scans. Singly charged precursors were filtered through ion mobility, and only precursor signals over an intensity threshold of 1000 were selected for fragmentation. The target intensity was set to 10,000. The precursors were dynamically excluded for 0.4 min. The quadrupole isolation width was set as 2.0 Da at $m/z$ 700, 3.0 Da at $m/z$ 800. The collision energy was ramped linearly as a function of the mobility from 59 eV at 1/K0 = 1.6 Vs/cm² to 20 eV at 1/K0 = 0.6 Vs/cm².

### Data analysis

Raw MS data were analyzed with MaxQuant (version 2.0.1.0) using the Andromeda database search algorithm. Spectra files were searched against the Human protein sequence database downloaded from Uniprot (20230619, 20423 entries) using the following parameters: variable modifications were set as Oxidation (M)-

15.994915; Acetyl (Protein N-term)-42.010565, fixed modification was set as Carbamidomethyl (C)-57.021464; Enzyme digestion specificity was set to Trypsin/P with maximum two missed cleavages; Peptide mass tolerance in first search and main search were set to 20 ppm and 10 ppm; Fragment match tolerance was set to 40 ppm. Proteins that could not be distinguished based on unique peptides were merged into a single protein group using MaxQuant. Label-free protein quantification was performed using the MaxLFQ algorithm, and "match-between-runs" was enabled. The search results were filtered at 1% FDR at both the peptide and protein levels.

## Coimmunoprecipitation (Co-IP)

After incubation of the engineered EVs with cells in the presence of BafA1 (MCE, USA) for 24 h, the cellular proteins were lysed with NP40 lysis buffer (50 mM Tris-HCl [pH 7.4], 150 mM NaCl, 1 mM EDTA, and 1% NP-40) containing 1% PMSF (Boster, USA) on ice for 30 min and centrifuged at 12,000×g for 15 min to extract the proteins. Then, the protein solution was incubated with a specific anti-bait antibody/Protein A/G Magnetic Beads (MCE, USA) (1: 10 v/v) system (MedChemExpress, USA) at 4 °C overnight to enrich the bait protein and the interacting proteins. The magnetic beads were eluted with NP40 lysis buffer three times to remove nonspecifically bound proteins. Proteins were denatured using SDS−PAGE loading buffer (Boster, USA) at 95 °C for 10 min and examined using western blotting or mass spectrometry.

## GST pulldown

Rosetta *Escherichia coli* with GST or GST-LC3B plasmids was incubated at 37 °C and 220 rpm until the OD reached 0.6–1.0. IPTG (MedChemExpress, USA) was added at a final concentration of 1 mM and incubated for 4 h to induce protein expression. The engineered *E. coli* cells were lysed using a GST-pull down Kit (Beyotime, China) to extract the proteins, according to the manufacturer's instructions. The protein solution was incubated with GST purification resin for 30 min and washed thrice with PBS. After washing, GST-labeled proteins from the input groups were eluted with GSH elution buffer for the expression and purification of GST-labeled proteins. The isolated EVs were incubated with GST or GST-LC3B protein and conjugated with GST resin at 4 °C overnight. After washing thrice with PBS, the EVs that were bound to GST-LC3B were eluted with GSH elution buffer for further detection.

## Proximity ligation assay (PLA)

A PLA Kit (Naveni, Sweden) was used according to the manufacturer's instructions. Cells were fixed and permeabilized according to an immunofluorescence protocol. The blocking solution from the PLA Kit was used on the entire sample area at 37 °C for 60 min. Then, the blocking solution was discarded. Primary antibody (Flag, Proteintech, 1:50; His, Proteintech, 1:50) diluted with Diluent buffer from the PLA Kit was added to cover the sample area, followed by incubation for 120 min at 37 °C. The slides were washed thrice for 5 min each time with TBS/T. Then, Navanibody M1 and Navanibody R2 were diluted at 1:40 in Diluent buffer and added to cover the sample area, followed by incubation for 60 min at 37 °C. The slides were washed thrice for 5 min each

time with TBS/T. Enzyme 1 (1:40) and Buffer 1 (1:5) were diluted in water and added to cover the sample area for reaction 1 at 37 °C for 30 min. The slides were washed thrice for 5 min each time with TBS/T. Next, Enzyme 2 (1:40) and Buffer 2 (1:5) were diluted in water and added to cover the sample area for reaction 2 at 37 °C for 90 min. The slides were washed thrice for 5 min each time with TBS. DAPI was added for nuclear staining (at room temperature), and the slides were washed thrice with TBS before mounting. Finally, the images were obtained using a Confocal Laser Scanning Microscope (CLSM) FV3000 (Olympus, Japan).

## CRISPR-Cas9-mediated knockout in HeLa cells

ATG7, SQSTM1, MAP1LC3B, and MAP1LC3B2 KO HeLa cells were generated using the CRISPR-Cas9 gene-editing technology. Target sequences were selected using the CRISPR design available at the Broad Institute (https://portals.broadinstitute.org/gpp/public/analysis-tools/sgrna-design). Guide DNA sequences were inserted into the lentiCRISPRv2 plasmid (#52961; Addgene). The resulting plasmids were transfected using Lipo2000 (Thermo Fisher Scientific). After 36 h, the cells were treated with 10 µg/mL of puromycin for 48 h to screen the cells. Following antibiotic selection, 100 cells were diluted and inoculated into a 96-well plate. Single clones were selected after 2 weeks, and the selected KO clones were validated using Sanger sequencing (Appendix Fig. S7). The sgRNAs for the CRISPR design are shown in Table EV3.

## Luminex assay

First, 50 µL of beads diluted with an assay buffer and 50 µL of standard, sample, and blank control were added to a 96-well plate. The plate was affixed with a sealing membrane and incubated in the dark at room temperature for 30 min at 850 rpm on a plate oscillator. Then, the samples were discarded and washed thrice, and 25 µL of the diluted detection antibody was added. The plate was affixed to a sealed membrane and incubated in the dark at room temperature for 30 min at 850 rpm on a plate oscillator. After incubation, the detection antibody was discarded, and the cells were washed three times. Then, 50 µL of streptavidin-PE was added to each well. The plate was affixed with a sealing membrane and incubated in the dark at room temperature for 10 min at 850 rpm on a plate oscillator. The plates were washed thrice, and 125 µL of assay buffer was added to each well. A sealing membrane was affixed onto the plate, and the samples were incubated in the dark at room temperature for 2 min at 850 rpm on a plate oscillator. After incubation, the 96-well plate was fed into a corrected Bio-Plex plate to obtain the results.

## RNA-sequencing

RNA was extracted from the samples using a TRIzol kit (Thermo Fisher Scientific, USA), and mRNA was isolated from total RNA using magnetic beads containing oligo-dT. The captured mRNA was fragmented and reverse transcribed to synthesize cDNA, which was subsequently subjected to end repair via the addition of an A base at the 3' end. Subsequently, the cDNA fragments were ligated to the sequencing adapter, and incomplete ligation products and adapter self-ligation products were removed. A sequencing library was obtained using PCR amplification with primers that were

complementary to the adapter sequence, followed by purification with magnetic beads. A Qubit was used to determine library concentration, whereas an Agilent fragment analyzer was used to determine library fragment length to ensure library quality. Libraries were sequenced using PE150 (150 bp paired-end) on an Illumina NovaSeq 6000 sequencing platform.

## Immunoprecipitation mass spectrometry (IP-MS)

### Coimmunoprecipitation

Flag-SQSTM1 EVs were incubated with recipient cells in the presence of BafA1 for 24 h. After incubation, we removed the cell supernatants and washed with PBS twice. Then, cells were lysed using NP40 lysis buffer on ice for 30 min, and centrifuged at $12,000 \times g$ for 15 min to extract proteins. Then, the resulting protein solution was incubated with Anti-Flag antibody (Proteintech, China)/Protein A/G Magnetic Beads (MCE, USA) (1: 10 v/v) system at 4 °C overnight to enrich the Flag-SQSTM1 as well as its interacting proteins. After incubation, wash beads with NP40 lysis buffer three times. Flag-enriched proteins and their abundance were eluted for further detection by mass spectrometry.

### Sample preparation

Samples were incubated in the reaction buffer (2.5% SDS, 100 mM Tris-HCl pH 8.0) at 95 °C for 10 min for protein denaturation. After centrifugation, proteins in the supernatant were precipitated with TCA. The precipitant was then diluted with reaction buffer (1% SDC, 100 mM Tris-HCl, pH = 8.5, 10 mM TCEP, 40 mM CAA) and incubated for 30 min at 60 °C for reduction and alkylation. The solution was diluted with an equal volume of $H_2O$ and subjected to trypsin digestion by adding 1 μg trypsin for overnight digestion at 37 °C. The next day, TFA was used to bring the pH down to 6.0 to end the digestion. After centrifugation at $12,000 \times g$ for 15 min). The peptide was purified using SDB desalting columns. The eluate was vacuum dried and stored at −20 °C for later use.

### LC-MS/MS detection

All samples were analyzed on an UltiMate 3000 RSLCnano system coupled on-line with Q Exactive HF mass spectrometer through a Nanospray Flex ion source (Thermo). Peptide samples were injected into a C18 Trap column (75 μm*2 cm, 3 μm particle size, 100 Å pore size, Thermo), and separated in a reversed-phase C18 analytical column packed in-house with ReproSil-Pur C18-AQ resin (75 μm*25 cm, 1.9 μm particle size, 100 Å pore size). Mobile phase A (0.1% formic acid/3% DMSO/97% $H_2O$) and mobile phase B (0.1% formic acid/3% DMSO/97% ACN) were used to establish the separation gradient at a flow rate of 300 nL/min. The MS was operated in DDA top20 mode with a full scan range of 350–1500 $m/z$. AGC Target value for the full MS scan was 3E6 charges with a maximum injection time of 30 ms and a resolution of 60,000 at $m/z$ 200. Precursor ion selection window was kept at 1.4 $m/z$, and fragmentation was achieved by higher-energy collisional dissociation (HCD) with a normalized collision energy of 28. Fragment ion scans were recorded at a resolution of 15,000, an AGC of 1E5 and a maximum fill time of 50 ms. Dynamic exclusion was enabled and set to 30 s.

### Database search

MS raw data were analyzed with MaxQuant using the Andromeda database search algorithm. Spectra files were searched against

Human database (2023-06-19,20423 entries) using the following parameters: LFQ mode was checked for quantification; Variable modifications, Oxidation (M), Acetyl (Protein N-term) & Deamidation (NQ); Fixed modifications, Carbamidomethyl (C); Digestion, Trypsin/P; The MS1 match tolerance was set as 20 ppm for the first search and 4.5 ppm for the main search; the MS2 tolerance was set as 20 ppm. Search results were filtered with 1% FDR at both protein and peptide levels. Proteins denoted as decoy hits, contaminants, or only identified by sites were removed; the remaining identifications were used for further quantification analysis.

## Transmission electron microscope (TEM)

The extracellular vesicles were isolated and diluted to a final concentration of $1 \times 10^{10}$ for transmission electron microscopy. In all, 10 μL of extracellular vesicles were added on a copper mesh to settle for 1 min, and the floating fluid was aspirated off with filter paper. In total, 10 μL uranyl acetate was dropped onto the copper net to precipitate for 1 min, and the floating fluid was sucked off with filter paper. Let dry for a few minutes at room temperature. Extracellular vesicle samples were observed using a transmission electron microscope HT-7700 (Hitachi, Japan) at 100kv.

## Nanoparticle tracking analysis (NTA)

Extracellular vesicles were first isolated from the cell supernatant. The isolated extracellular vesicles were placed on ice. Extracellular vesicle samples were diluted 1000-fold using PBS, and the diluted samples were dropped onto a nanoparticle tracking analyzer ZetaVIEW (Particle Matrix, Germany). Particle size detection and analysis were completed as directed by the program.

## Immune electron microscopy (IEM)

The extracellular vesicles were isolated and diluted to a final concentration of $1 \times 10^{10}$ for transmission electron microscopy. 10 μL of extracellular vesicles were added on a copper mesh to settle for 1 min, and the floating fluid was aspirated off with filter paper. Samples were blocked by 1% BSA/TBS for 30 min at room temperature. Primary antibody (SQSTM1, Proteintech) was diluted with 1% BSA/TBS (1:200) and incubated with samples for 20 min at room temperature. Then, TBS was used to wash samples three times for 15 min. Secondary antibody (10 nm gold conjugated anti-mouse antibody, Sigma) was diluted with 1% BSA/TBS (1:200). Secondary antibody was incubated first at room temperature for 20 min, next at 37 °C for 1 h, finally at room temperature for 30 min. TBS was used to wash samples three times for 15 min followed by $ddH_2O$ wash five times for 25 min. In all, 10 μL uranyl acetate was dropped onto the copper net to precipitate for 1 min, and the floating fluid was sucked off with filter paper. Then, wash samples with 70% ethanol, followed by $ddH_2O$ three times each. Extracellular vesicle samples were observed using a transmission electron microscope HT-7700 (Hitachi, Japan) at 100kv.

## NanoFCM Flow NanoAnalyzer

Extracellular vesicles were first isolated from the cell supernatant. The isolated extracellular vesicles were placed on ice. Extracellular

vesicle samples were diluted 1000-fold using PBS. NanoFCM Flow NanoAnalyzer (NanoFCM, China) was calibrated by standard exosome samples. In total, 100 μL of the extracellular vesicles sample was collected at a pressure of 1.0 kPa. The collected data were analyzed by NF Profession2.0 (NanoFCM, China).

## Real-time PCR

The sample tissue should be ground in Trizol lysate on ice until there are no visible tissue blocks, while cells were lysed in Trizol lysate on ice for 10 min. After lysis, the samples were centrifuged at 12,000 rpm, 4 °C for 10 min, and the supernatant was removed. In all, 100 μL of chloroform was added to the supernatant, vortexed for 15 s, and allowed to stand for 20 min on ice. The samples were then centrifuged at 12,000 rpm at 4 °C for 30 min. The supernatant was removed by aspiration and washed twice with 75% ethanol. The resulting RNA was precipitated by centrifugation at 12,000 rpm, 4 °C for 5 min. After drying, RNA was dissolved in 20 μL of ddH$_2$O, and the concentration and purity of RNA were detected by Nanodrop2000 (Thermo Fisher, USA). The Reverse Transcription was performed using the HiScript III RT SuperMix for qPCR (Vazyme, China) according to the manufacturer's instructions. The resulting cDNA was mixed with specific primers (shown in Table EV2) and Universal SYBR qPCR Master Mix (Vazyme, China) according to the manufacturer's instructions. The resulting mixture was examined for Real-time PCR in CFX Connect (Bio-Rad, USA).

## shRNA-mediated knockdown

Targeting shRNAs were designed and constructed in the pLKO.1 vector. The targeting sequences were shown in Table EV1. HEK293T cells or NPCs were seeded at day 1. shRNAs were transfected into cells with Lipo2000 at a ratio of 1:2 (w/v) at day 2 (HEK293T) or day 3 (NPCs). Experiments were carried out 24 h after transfection. At day 4 (HEK293T) or day 5 (NPCs), the cells or extracellular vesicles were collected for further experiments.

## ELISA assay

The supernatant of the cells to be tested was centrifuged at 1000× *g* for 20 min at 4 °C after collection. The resulting supernatant was used for subsequent detection. Tissue samples were first rinsed with PBS to remove residual blood. After weighing, the tissues were cut up and dissolved in PBS containing 1% PMSF (1:9, w/v). Next, the tissue samples were fully ground at 4 °C in a glass paddle homogenizer. The resulting homogenate was centrifuged at 5000× *g*, 4 °C for 10 min, and the resulting supernatant was used for subsequent assays. human-IL1b (Elabscience, China), human-TNFa (Elabscience, China), rat-IL1b (Elabscience, China), rat-TNFa (Elabscience, China), goat-IL1b (Finebio, China), goat-TNFa (Finebio, China) ELISA kits were used following the manufacturer's instructions to detect these cytokines in the supernatant or intervertebral disc tissues.

## In vivo experiments

### Injection of PKH26-labeled EVs into IVDs of SD rats
Six-week-old male SD rats were randomly assigned to each experimental group. After anesthesia with sodium pentobarbital,

10 μL of PKH26-labeled EVs at a concentration of $1 \times 10^{10}$ was injected into the Co7/8 IVD of the SD rats using 33-gauge needles. PKH26 fluorescence signals were observed at the indicated time points using small animal imaging.

### Needle puncture IVDD model in SD rats
Six-week-old male SD rats were randomly assigned to each experimental group. After anesthesia with sodium pentobarbital, a 26-gauge needle was used to puncture the Co7/8 IVD of the SD rats. In the sham group, the needle only penetrated the skin. Two weeks after needle puncture, 10 μL of blank or engineered EVs at a concentration of $1 \times 10^{10}$ was injected into the Co7/8 IVD of SD rats using 33-gauge needles after anesthesia with sodium pentobarbital.

### Micro-CT, MRI, and X-ray evaluation
After 2 weeks of treatment, micro-CT, MRI, and X-ray were performed to evaluate the IVDD of Co7/8. The SD rats were anesthetized and kept in the supine position for tail evaluation. Pfirrmann grades were evaluated using T2-weighted MR images of the Co7/8 (Che et al, 2019). The disc height index (DHI) was calculated based on X-ray images (Chen et al, 2022).

### Surgical IVDD model in Boer goats
Eight 4-year-old male goats were randomly assigned to two groups. After anesthesia, the L3/4 discs of the goats were exposed, a lacet was inserted into the center of the disc to a depth of 5 mm, and 100 μL of chondroitinase ABC at a concentration of 5 U/mL was injected. In the sham group, only L3/4 disc exposure was performed. After modeling, the muscle, subcutaneous tissue, and skin were sutured layer-by-layer. After 14 days, we anaesthetized the goats, injected 100 μL of EVs or DTI-EVs at a concentration of $1 \times 10^{10}$ into the center of the L3/4 disc, and sutured the wound.

## HE staining

Tissue sections were immersed in xylene for 10 min twice. Gently shake off excess liquid between each step. Tissue sections were immersed in absolute ethanol for 5 min, 95% ethanol for 5 min, 85% ethanol for 5 min, 75% ethanol for 5 min to rehydrate the tissue. Rinsing with ddH$_2$O for 1 min. Sections were stained with hematoxylin solution for 4 min and then washed with ddH$_2$O for 2 min until no excess dye came out of the sections. Sections were differentiated with 0.8% hydrochloric acid alcohol for 2 s and washed with ddH$_2$O. Sections can also be treated with lithium carbonate solution to be bluer and washed with tap water for 2 min. Then, sections were stained with eosin solution (alcohol soluble) for 20 s without water washing. Sections were treated with 95% ethanol for 5 min and dehydrated with absolute ethanol I and II for 2 min. The tissue sections were transparent with xylene and then mounted with neutral balsam. Examination with microscope CX-31 (Olympus, Japan).

## Safranin-O-Fast Green staining

Tissue sections were immersed in xylene for 10 min twice. Gently shake off excess liquid between each step. Tissue sections were immersed in absolute ethanol for 5 min, 95% ethanol for 5 min, 85% ethanol for 5 min, 75% ethanol for 5 min to rehydrate the tissue. Rinsing with

**The paper explained**

**Problem**
The development of targeted protein degradation techniques still remains hindered by the following three major obstacles: First, the application of these extracellular targeting protein degradation (eTPD) techniques relies on the expression of specific transmembrane proteins such as CI-M6PR and RNF43. Second, TPD techniques are difficult to degrade multiple targets. Third, since most of the existing TPD technologies are small molecules, their biocompatibility and clearance in the human body are difficult to predict.

**Results**
The autophagy pathway plays a crucial role in the degradation of EVs. Mechanistically, SQSTM1 bound to EVs interacts with MAP1LC3B within the cell through its LIR motifs, thereby entering the autophagy pathway for degradation. Leveraging this mechanism, we have developed an EV-based targeted protein degradation platform. This platform does not rely on membrane receptors and has the ability to degrade multiple targets simultaneously.

**Impact**
Our research has discovered a previously unrecognized mechanism that mediates the degradation of EVs. In addition, we have innovatively developed a highly modular EV-based targeted degradation platform. This platform has the ability to target multiple targets and does not rely on cell membrane receptors, thereby expanding the application scope of targeted protein degradation.

ddH$_2$O for 1 min. Sections were stained in fast green for 5 min followed by a wash with ddH$_2$O gently to remove free fast green dye. Sections were washed for 10 s using a weak acid solution to further remove the fast green dye. The sections were stained in safranin solution for 5 min. Absolute ethanol was then used to differentiate for 8 min until the bone tissue was shown green. Immerse the sections in absolute ethanol for 10 s. The tissue sections were transparent with xylene and then mounted with neutral balsam. Examination with microscope CX-31 (Olympus, Japan).

## Masson staining

Tissue sections were immersed in xylene for 10 min twice. Gently shake off excess liquid between each step. Tissue sections were immersed in absolute ethanol for 5 min, 95% ethanol for 5 min, 85% ethanol for 5 min, 75% ethanol for 5 min to rehydrate the tissue. Rinsing with ddH$_2$O for 1 min. Immerse the dehydrated sections in Bouin's solution of Zenker's solution overnight followed by ddH$_2$O washing. Sections were stained with hematoxylin solution for 10 min and slightly washed with ddH$_2$O. 0.8%-1% hydrochloric acid alcohol was used for section differentiation, followed by ddH$_2$O washing for several min. Lithium carbonate solution was also used to make the sections bluer. Then, sections were stained with ponceau acid fuchsin solution for 10 min and washed with ddH$_2$O. Sections were treated with phosphomolybdic acid solution for about 5 min and then stained with aniline blue solution for 5 min without washing. 1% glacial acetic acid was used to treat sections, followed by dehydration with 95% alcohol and absolute for 5 min. The tissue sections were transparent with xylene and then mounted with neutral balsam. Examination with microscope CX-31 (Olympus, Japan).

## Histological evaluation

The IVD tissues were embedded in paraffin and sectioned. After deparaffinization and rehydration, the IVD sections were subjected to H&E, S-O, and Masson staining. The histological scores of the IVDs were evaluated based on the IVD staining images and a modified histological grading scale (Han et al, 2008).

## Statistical analysis

The experiments in this study were independently repeated at least three times unless otherwise stated in the figure legends. The sample size ($n$) of the biological replicates is shown in the corresponding figure legend. All data are presented as mean ± SD. Differences between the groups were analyzed using an unpaired two-tailed $t$ test or one-way analysis of variance (ANOVA). Significance was set at $P < 0.05$. Data analysis was performed using GraphPad Prism 9.0 (GraphPad Software, USA).

## Graphics

Figures 1a, 1g, 2a, 3a, 4h, 4p, 5c, 5g, 6a and synopsis were created with BioRender.com.

## Data availability

The datasets produced in this study are available in the following databases: RNA-Seq data: Gene ExpressionOmnibus GSE312760; Protein interaction AP-MS data: ProteomeXchange Consortium (https://proteomecentral.proteomexchange.org) with the dataset identifier PXD071687.

The source data of this paper are collected in the following database record: biostudies:S-SCDT-10_1038-S44321-025-00371-8.

## Peer review information

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

## Acknowledgements

This work was supported by the National Science Foundation of China (Nos. 82502966, 82130072, 82072505, and 82472511), ShenZhen Science and Technology Program (SGDX20230116093544006, JCYJ20240813153421028, JCYJ20250604190927038), the Natural Science Foundation of Hubei Province (No. 2025AFB089), and the China Postdoctoral Science Foundation (No. 2024M761065). We also thank the Huazhong University of Science and Technology Analytical & Testing Center, Medical sub-center, and Wuhan Center for Magnetic Resonance, Innovation Academy for Precision Measurement Science and Technology, Chinese Academy of Sciences, for technical support. The animal experiments in this work were supported by the Huazhong University of Science and Technology laboratory animal center.

## Author contributions

**Bide Tong**: Conceptualization; Data curation; Methodology. **Xiaoguang Zhang**: Resources; Investigation; Methodology. **Dingchao Zhu**: Resources; Validation; Investigation; Methodology. **Yulei Wang**: Data curation; Software; Validation; Investigation. **Junyu Wei**: Validation; Investigation; Visualization. **Zixuan Ou**:

Software; Validation. **Huaizhen Liang**: Conceptualization; Software; Visualization. **Hanpeng Xu**: Conceptualization; Software; Validation; Investigation. **Zhengdong Zhang**: Conceptualization; Validation; Investigation; Visualization. **Jie Lei**: Resources; Software; Validation; Visualization. **Xingyu Zhou**: Validation; Visualization. **Di Wu**: Resources; Software; Validation; Visualization; Writing—original draft; Project administration. **Yu Song**: Resources; Validation; Visualization; Writing—original draft; Project administration. **Kun Wang**: Conceptualization; Resources; Funding acquisition; Validation; Visualization; Writing—original draft; Project administration; Writing—review and editing. **Xiaobo Feng**: Resources; Validation; Visualization; Writing—original draft; Project administration. **Lei Tan**: Conceptualization; Resources; Funding acquisition; Validation; Visualization; Writing—original draft; Project administration; Writing—review and editing. **Zhiwei Liao**: Conceptualization; Resources; Validation; Investigation; Visualization; Writing—original draft; Project administration. **Cao Yang**: Conceptualization; Resources; Funding acquisition; Investigation; Project administration; Writing—review and editing.

Source data underlying figure panels in this paper may have individual authorship assigned. Where available, figure panel/source data authorship is listed in the following database record: biostudies:S-SCDT-10_1038-S44321-025-00371-8.

## Disclosure and competing interests statement

The authors declare no competing interests.

# Expanded View Figures

**Figure EV1.   The degradation of extracellular vesicles occurs through an autophagy-dependent lysosomal pathway.**

(A) Representative TEM image of HEK293T-derived EGFP-EVs (scale bar: 100 nm). (B) Nanoparticle tracking analysis (NTA) of EGFP-EVs. (C, D) Immunoblot assay analysis of EVs markers and EGFP expression in EGFP-EVs and cell lysates. (E) EGFP-EVs were added to NPCs for an incubation of 12 h. Then, free EGFP-EVs were washed out and replaced by fresh medium. Degradation of EGFP-EV in NPCs at 0, 3, 6, 12, and 24 h were analyzed by western blot ($n = 3$). (F–H) Analysis of EGFP level in DMSO, MG132 and BafA1 groups NPCs co-incubated with EGFP-EVs derived from HEK293T (F), HeLa (G) and MSC (H) cells for 12 h (uptake) and wash out for 12 h (degradation). (I–L) Analysis of EGFP level in DMSO, MG132 and BafA1 groups HeLa cells co-incubated with EGFP-EVs derived from HEK293T (I), HeLa (J) and MSC (K) cells for 12 h (uptake) and wash out for 12 h (degradation) ($n = 3$). (M–O) Analysis of EGFP level of EGFP in DMSO, 3-MA and ML282 groups NPCs co-incubated with EGFP-EVs derived from HEK293T (M), HeLa (N) and MSC (O) cells for 12 h (uptake) and wash out for 12 h (degradation). (P–S) Analysis of EGFP level in DMSO, 3-MA and ML282 groups HeLa cells co-incubated with EGFP-EVs derived from HEK293T (P), HeLa (Q) and MSC (R) cells for 12 h (uptake) and wash out for 12 h (degradation) ($n = 3$); (T) NPCs were treated with EGFP-EVs for 12 h. After incubation, free EGFP-EVs were washed out and replaced by fresh medium with 50 nM BafA1 for 12 h. Co-localization of EGFP and LAMPII was analyzed by immunofluorescence (scale bar: 10 μm) ($n = 6$). (U) NPCs were treated with EGFP-EVs for 12 h. After incubation, free EGFP-EVs were washed out and replaced by fresh medium with 50 nM BafA1 for 12 h. Lysosome enrichment level of EGFP in NPCs in control, 3-MA and ML282 groups were analyzed by western blot ($n = 4$). (V) Analysis of lysosome enrichment level of EGFP in shScramble and shATG7-treated NPCs ($n = 3$). Data were analyzed by unpaired two-tailed $t$ tests (L, S–V). Data were shown as mean ± SD. Each $n$ in (L, S–V) is biological independent samples. The $P$ values are labeled in the figure.

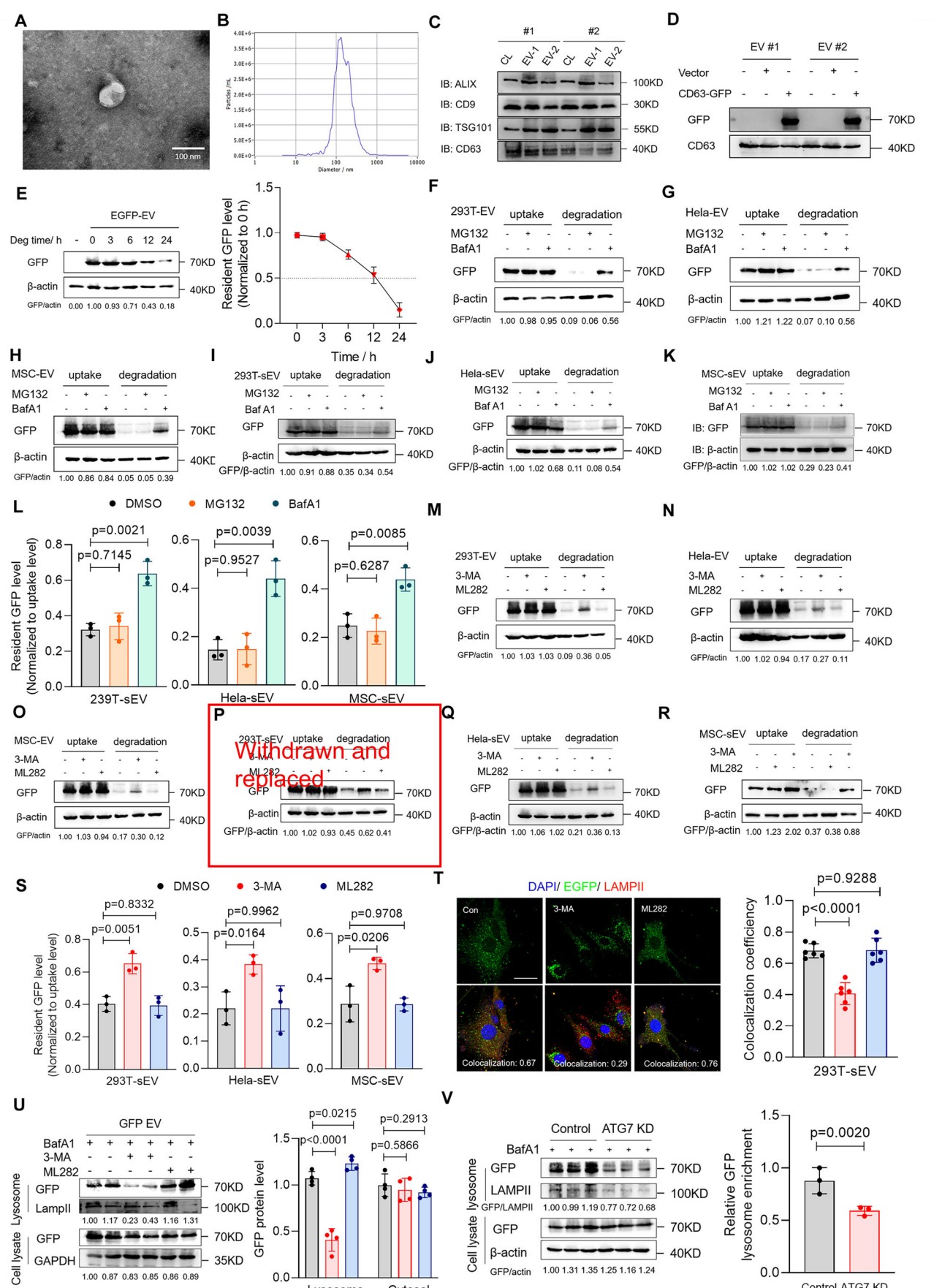

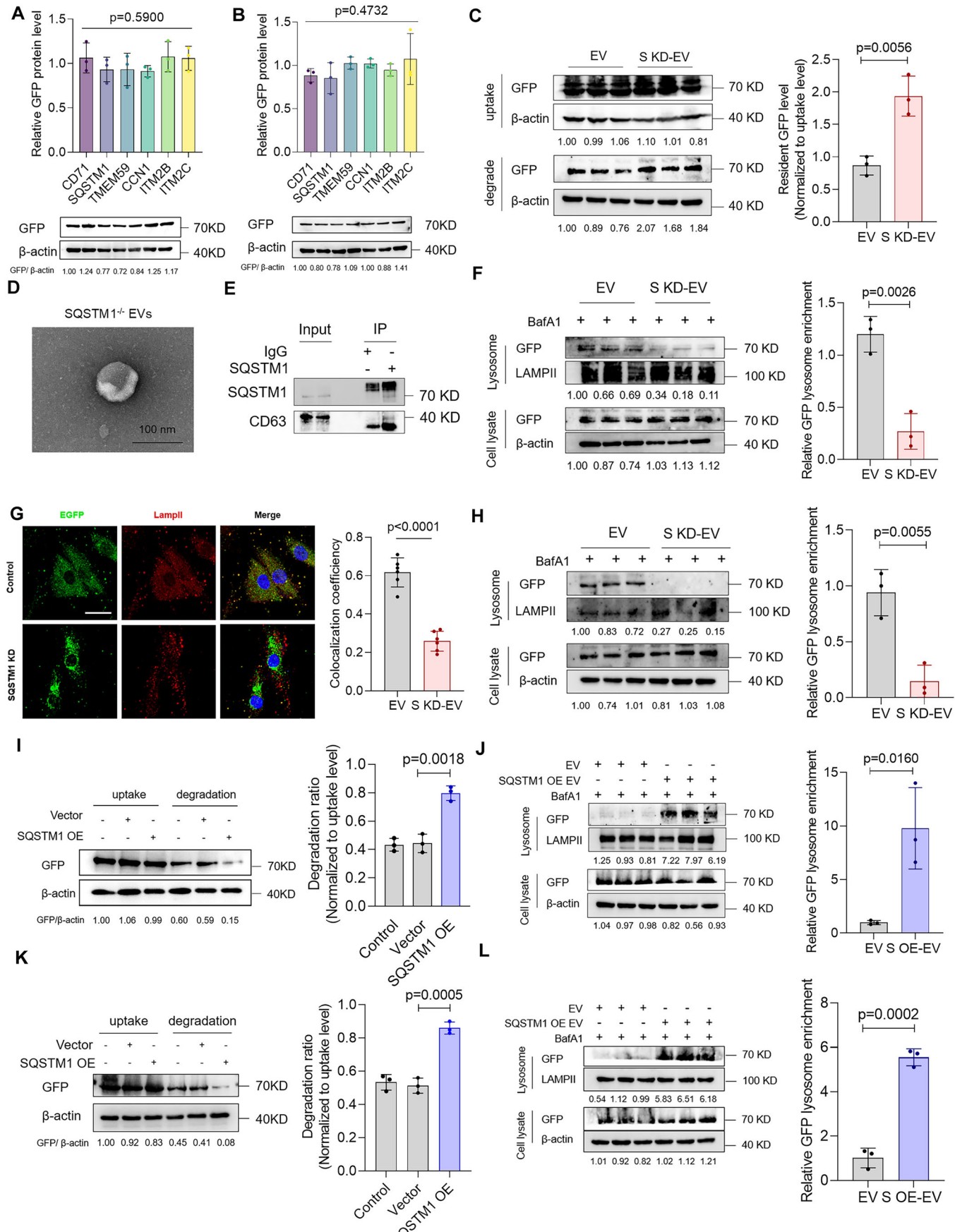

◀ **Figure EV2. SQSTM1 mediates extracellular vesicles entry into lysosome-dependent degradation pathway.**

(A) The uptake levels of specific antibody-blocking EVs (IgG, CD71, SQSTM1, TMEM59, CCN1, ITM2B, ITM2C) were detected after incubation for 12 h. IgG control was used to normalize the grayscale values of each group. Data were analyzed by one-way ANOVA ($n = 3$). (B) The uptake levels of specific gene-knockdown EVs (Vector, CD71, SQSTM1, TMEM59, CCN1, ITM2B, ITM2C) were detected after incubation for 12 h. Vector control was used to normalize the grayscale values of each group. Data were analyzed by one-way ANOVA ($n = 3$). (C) Control EVs and SQSTM1 KD EVs were incubated with NPCs for 12 h. Uptake level was analyzed at this time while degradation level was detected following wash out and another 12 h incubation ($n = 3$). (D) IEM assay to detect the expression of SQSTM1 on HEK-293T derived EVs (scale bar: 100 nm). (E) Anti-SQSTM1 antibody was used to pull down HEK-293T derived EVs. IgG was used as negative control. SQSTM1 as well as EV marker-CD63 were detected by western blot. (F) Control or SQSTM1 KD EVs were added to NPCs and co-incubated for 12 h. Then, free EVs were wash out and replaced by free mediun containing 50 nM BafA1 for 12 h. After incubation, lysosome enrichment method was used to detect the sorting efficiency of control and SQSTM1 knockdown EVs into lysosomes in NPCs ($n = 3$). (G) Control or SQSTM1 KD EVs were added to NPCs and co-incubated for 12 h. Then, free EVs were washed out and replaced by free mediun containing 50 nM BafA1 for 12 h. After incubation, immunofluorescence assay was used to detect the co-localization of EGFP with LampII (scale bar: 10 μm) ($n = 6$); (H) Control or SQSTM1 KD EVs were added to HeLa cells and co-incubated for 12 h. Then, free EVs were washed out and replaced by free mediun containing 50 nM BafA1 for 12 h. After incubation, lysosome enrichment method was used to detect the sorting efficiency of control and SQSTM1 knockdown EVs into lysosomes in HeLa cells ($n = 3$); (I, J) Analysis of degradation rate (I) and lysosomal sorting efficiency (J) of control and SQSTM1 overexpression EVs in NPCs ($n = 3$); (K, L) Analysis of degradation rate (K) and lysosomal sorting efficiency (l) of control and SQSTM1 overexpression EVs in HeLa cells ($n = 3$). Data were analyzed by one-way ANOVA (A, B) unpaired two-tailed $t$ tests (C, F, G, H, I, J, K, L). Data were shown as mean ± SD. Each $n$ in (A, B, C, F, G, H, I, J, K, L) is biological independent samples. The $P$ values are labeled in the figure.

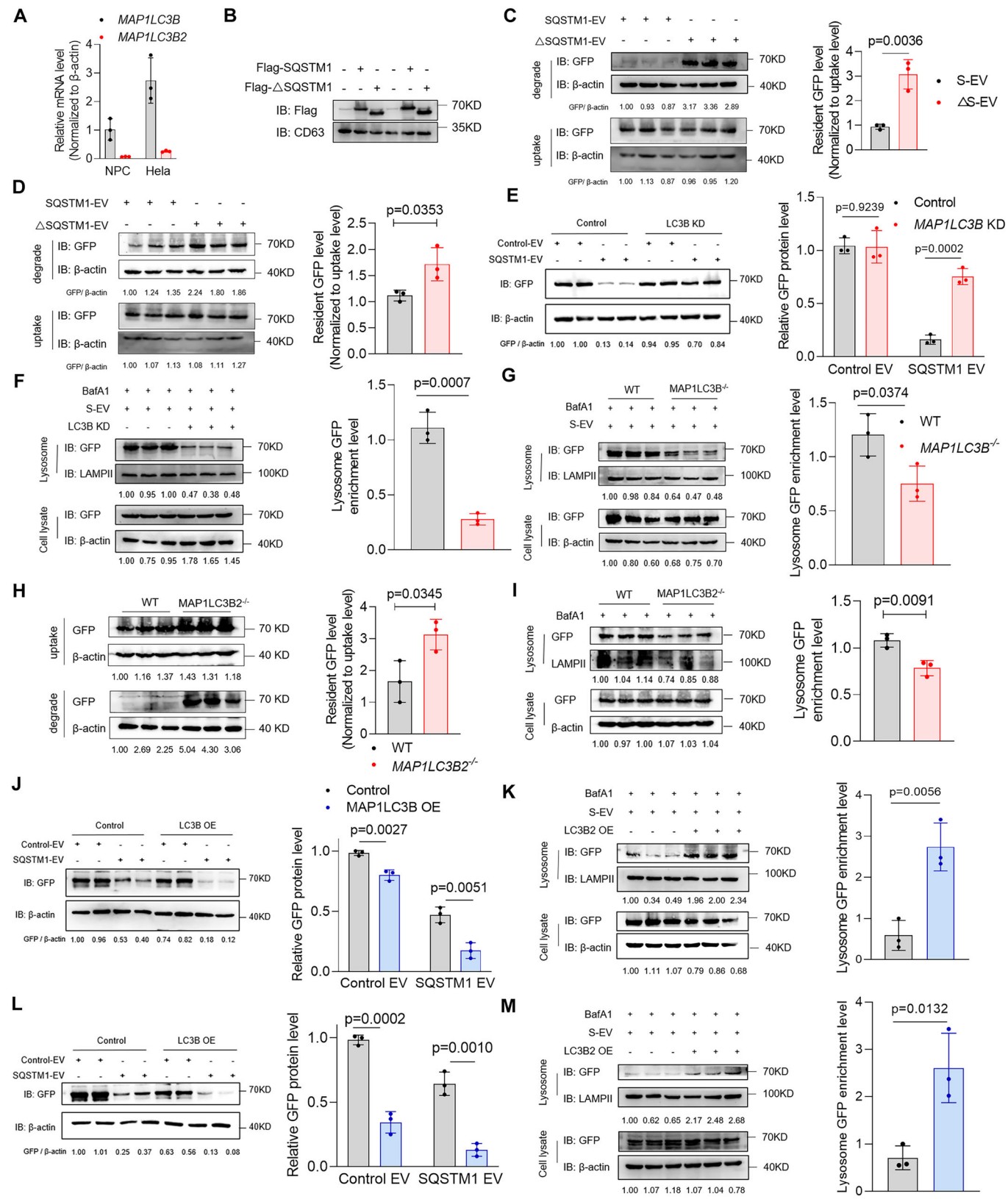

◀ **Figure EV3. EVs engineered with SQSTM1 degradation signal can be efficiently sorted into lysosome and degraded via interacting with MAP1LC3B.**

(A) The mRNA level of MAP1LC3B and MAP1LC3B2 in NPCs and HeLa cells were analyzed by Real-time PCR ($n = 3$). The expression of MAP1LC3B in NPCs was normalized as control. (B) The expression of Flag tagged SQSTM1 and Flag tagged truncated SQSTM1 in engineering EVs were detected by western blot. (C) SQSTM1 and its truncated form loaded EVs were incubated with NPCs for 12 h. Uptake level was analyzed at this time while degradation level was detected following wash out and another 12 h incubation ($n = 3$). (D) SQSTM1 and its truncated form loaded EVs were incubated with HeLa cells for 12 h. Uptake level was analyzed at this time while degradation level was detected following wash out and another 12 h incubation ($n = 3$). (E) Degradation level of control and SQSTM1-EVs in control or LC3B KD NPCs were detected after 12 h incubation following with washed out and another 12 h incubation in fresh medium ($n = 3$). (F) SQSTM1-EVs were added to control or LC3B KD NPCs for incubation of 12 h followed with wash out and incubation in fresh medium containing 50 nM BafA1 for 12 h. Lysosomal enrichment level were analyzed by lysosomal isolation and western blot ($n = 3$). (G) Lysosomal enrichment level of SQSTM1-EVs in wild-type and MAP1LC3B KO HeLa cells were analyzed by lysosomal isolation and western blot ($n = 3$). (H) The uptake and degradation level of SQSTM1 EVs in wild-type and MAP1LC3B2 KO HeLa cells were analyzed by western blot ($n = 3$). (I) Detection of lysosomal enrichment efficiency of SQSTM1-EV in wild-type and MAP1LC3B2 knockout HeLa cell lines ($n = 3$). (J, K) Detection of degradation level (J) and lysosomal enrichment efficiency (K) of SQSTM1-EV in control and MAP1LC3B overexpression NPC ($n = 3$). (L, M) Detection of degradation level (L) and lysosomal enrichment efficiency (m) of SQSTM1-EV in control and MAP1LC3B overexpression HeLa cells ($n = 3$). Data were analyzed by unpaired two-tailed $t$ tests (A, C–M). Data were shown as mean ± SD. Each $n$ in (A, C–M) is biological independent samples. The $P$ values are labeled in the figure.

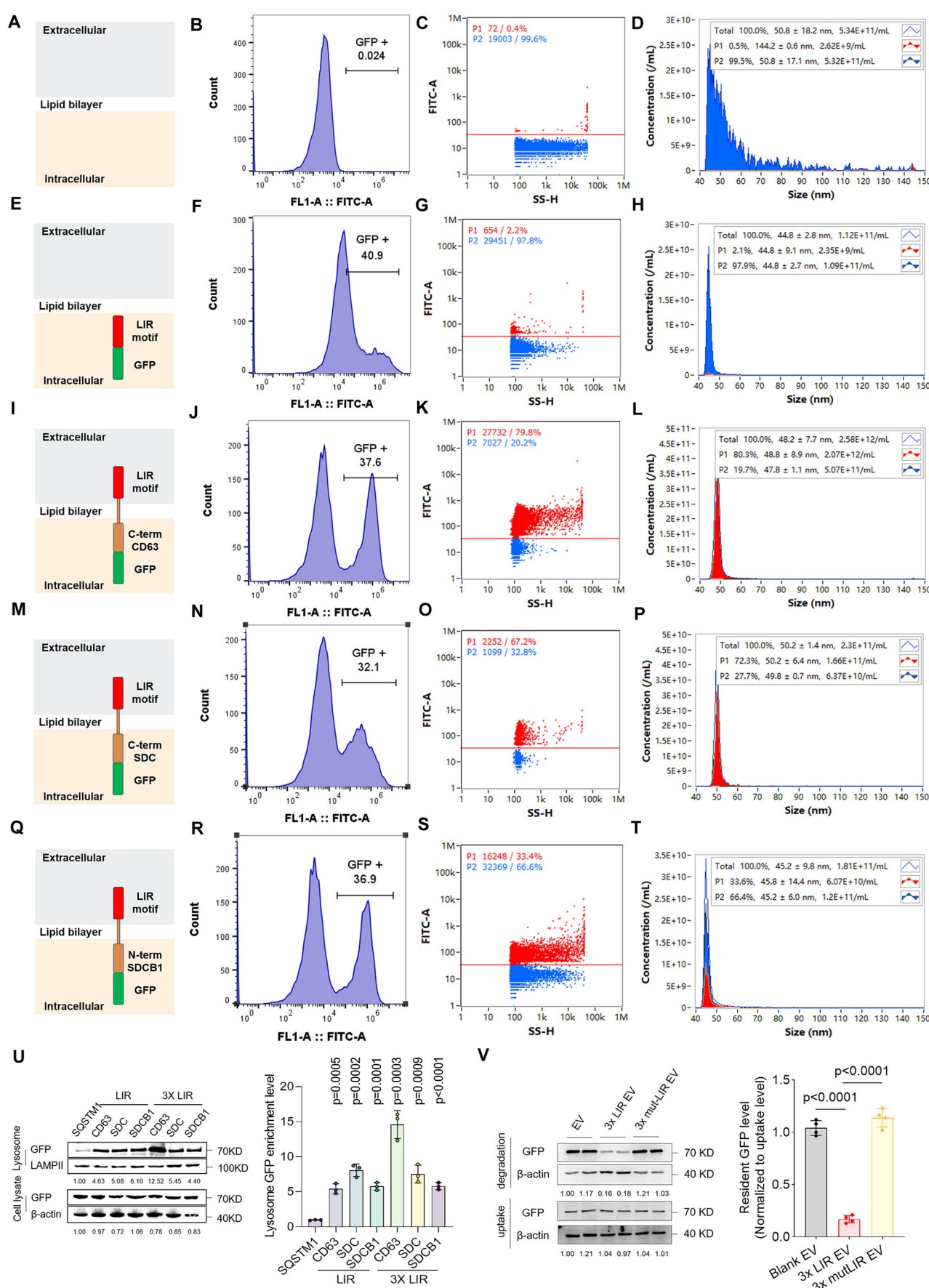

◀ **Figure EV4. The loading efficiency of EV expression scaffolds.**

(A–D) Control HEK293T (**A**) was analyzed by cytometry for GFP expression (**B**). HEK293T-derived EVs were analyzed by Nanoflow for GFP expression (**C**) as well as size distribution (**D**). (**E–H**) GFP-ILR motif transfected HEK293T (**E**) was analyzed by cytometry for GFP expression (**F**). Transfected HEK293T-derived EVs were analyzed by Nanoflow for GFP expression (**G**) as well as size distribution (**H**). (**I–L**) GFP-C term CD63-ILR motif transfected HEK293T (**I**) was analyzed by cytometry for GFP expression (**J**). Transfected HEK293T-derived EVs were analyzed by Nanoflow for GFP expression (**K**) as well as size distribution (**L**). (**M–P**) GFP-C term SDC-ILR motif transfected HEK293T (**M**) was analyzed by cytometry for GFP expression (**N**). Transfected HEK293T-derived EVs were analyzed by Nanoflow for GFP expression (**O**) as well as size distribution (**P**). (**Q–T**) GFP-N term SDCB1-ILR motif transfected HEK293T (**Q**) was analyzed by cytometry for GFP expression (**R**). Transfected HEK293T-derived EVs were analyzed by Nanoflow for GFP expression (**S**) as well as size distribution (**T**). (**U**) Different engineering EVs were added to NPCs for incubation of 12 h followed with wash out and incubation in fresh medium containing 50 nM BafA1 for 12 h. Lysosomal sorting efficiency of different EVs were detected by lysosome isolation and western blot ($n = 3$). (**V**) Representative immunoblot images and quantitative analysis of the resident GFP levels of blank, 3x LIR motifs and 3x L341V-mutated LIR motifs engineering EVs in HeLa cells ($n = 4$). Data were analyzed by unpaired two-tailed $t$ tests (**U**, **V**). Data were shown as mean ± SD. Each $n$ in (**U**, **V**) is biological independent samples. The $P$ values are labeled in the figure.

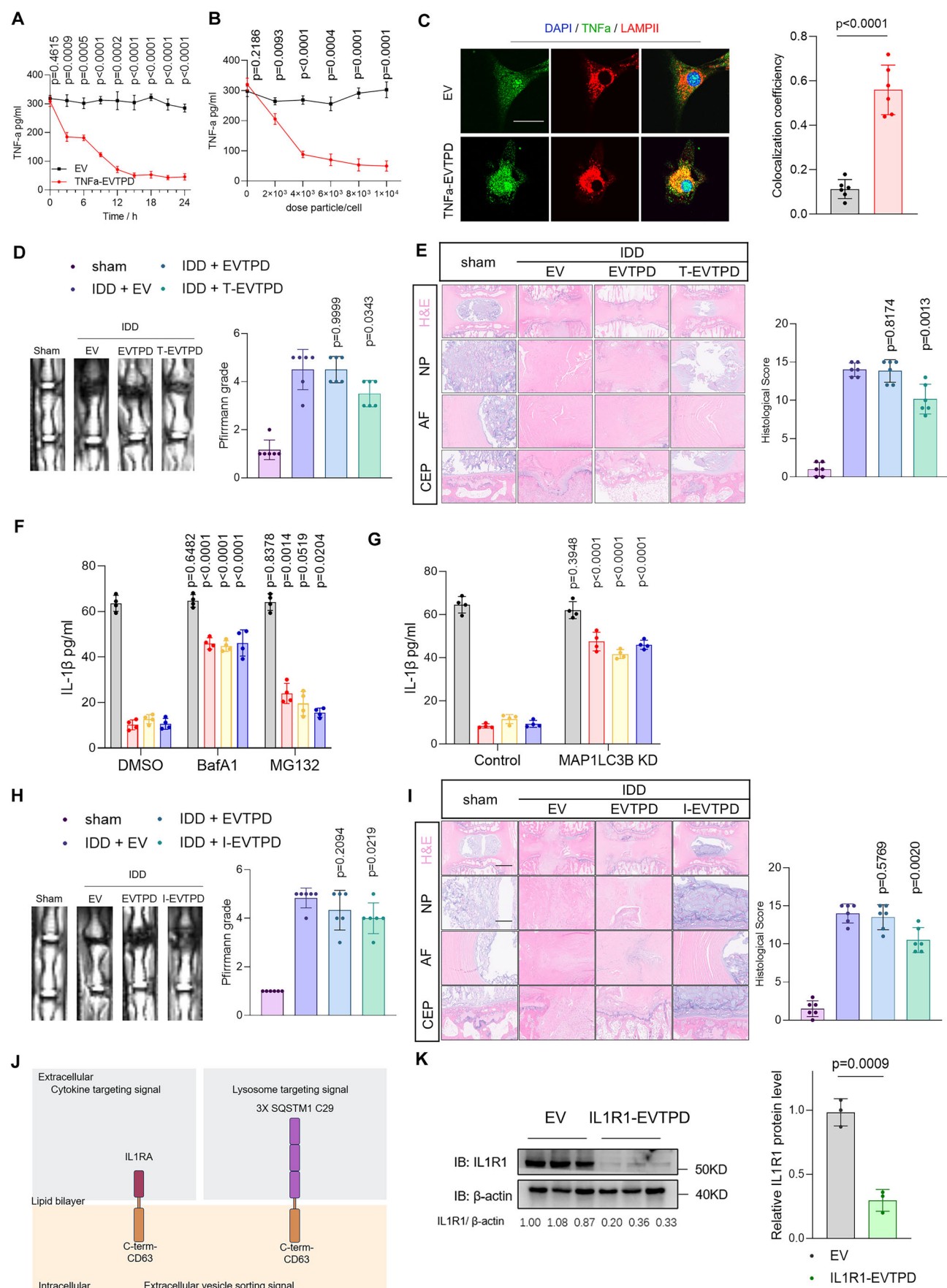

◄ **Figure EV5. EVs engineered with both degradation signal and cytokine-binding domains efficiently degrade TNF-α and IL-1β.**

(**A**) The degradation level of TNF-α in the supernatant of NPCs by blank EVs or TNFα-EVTPD at 0,3,6,9,12,15,18,21,24 h were detected by EVTPD assays ($n = 3$). (**B**) The degradation rate of TNF-α in the supernatant of NPCs by blank EVs and TNFα- EVTPD at concentration of 0, 2, 4, 6, 8 and $10 \times 10^3$ per cell were detected by ELISA assays ($n = 3$); (**C**) Fluorescence co-localization analysis of TNF-α and LAMPII in NPCs after EV, TNFA- EVTPD treatment for 24 h with 50 nM BafA1 ($n = 6$). (**D**) MRI images of co7/8 and co8/9 as control. The Pfirrmann grades of sham, IDD + EV, IDD + EVTPD and IDD + rTNFα- EVTPD were analyzed based on the MRI images ($n = 6$). (**E**) H&E staining images including intire intervertebral disc, nucleus pulposus (NP), annulus fibrosus (AF), and cartilage endplates (CEP) and histological scores of intervertebral disc sham, IDD + EV, IDD + EVTPD and IDD + rTNFα-EVTPD groups (Scale bar: 50 μm and 5 μm). (**F**) ELISA was used to detect the degradation rate of IL1β by IL1β- targeting EVs loaded with or without degradation signal in the DMSO, BafA1 and MG132 pretreated groups ($n = 4$). (**G**) ELISA was used to detect the effect of MAP1LC3B knockdown on the IL1β-degradation ability of engineered EVs ($n = 4$). (**H**) MRI images of co7/8 and co8/9 as control. The Pfirrmann grades of sham, IDD + EV, IDD + EVTPD and IDD + rIL1β- EVTPD were analyzed based on the MRI images ($n = 6$). (**I**) H&E staining images including entire intervertebral disc, nucleus pulposus (NP), annulus fibrosus (AF), and cartilage endplates (CEP) and histological scores of intervertebral disc tissue of rats in sham, IDD + EV, IDD + EVTPD and IDD + rIL1β-EVTPD groups (scale bar: 50 μm and 5 μm). (**J**) Schematic diagram of EVs loaded with degradation signal and IL1R1 binding domain. (**K**) Protein level analysis of IL1R1 levels after blank EV or IL1R1-EVTPD treatment for 24 h ($n = 3$). Data were analyzed by unpaired two-tailed *t* tests (**A–I, K**). Data were shown as mean ± SD. Each *n* in (**D, E, H, I**) is an individual rat, each *n* in (**A–C, F, G, K**) is biological independent samples. The *P* values are labeled in the figure.

