## [Peer Review File · EMBO Molecular Medicine]

Extracellular vesicle-based targeted protein degradation platform for multiple extracellular proteins

Bide Tong, Xiaoguang Zhang, Dingchao Zhu, Yulei Wang, Junyu Wei, Zixuan Ou, Huaizhen Liang, Hanpeng Xu, Zhengdong Zhang, Jie Lei, Xingyu Zhou, Di Wu, Yu Song, Kun Wang, Xiaobo Feng, Lei Tan, Zhiwei Liao, and Cao Yang

Corresponding authors: Cao Yang (caoyangunion@hust.edu.cn), Lei Tan (tanlei_union@hust.edu.cn), Zhiwei Liao (lzwhust@hust.edu.cn)

Review Timeline:

Submission Date:	14th Nov 24
Editorial Decision:	8th Jan 25
Revision Received:	14th Oct 25
Editorial Decision:	3rd Dec 25
Revision Received:	6th Dec 25
Accepted:	11th Dec 25

Editor: Jingyi Hou

Transaction Report:

8th Jan 2025

Dear Prof. Yang,

Thank you for submitting your work to EMBO Molecular Medicine. I would like to apologise for the slow process, which was due to the late arrival of reviewers' reports and holiday season. We have now heard back from the three reviewers who agreed to evaluate your manuscript. You will see from the comments below that the reviewers find the manuscript to be of interest. They raise, however, several important points, which should be convincingly addressed in a revision of this work.

I think that the recommendations of the reviewers are rather clear so there is no need to repeat the points listed below. Overall, the technology appears interesting, but the reviewers raised many concerns regarding technical quality and data presentation. Referee #3 highlighted issues with the controls used, while Referee #2 requested benchmarking against existing related methods, which need to be carefully addressed. During our pre-decision cross-commenting process (in which the reviewers are given a chance to make additional comments, including on each other's reports), Reviewer #3 made further comments, " Agree with benchmarking with existing methods on two levels: (i) comparing AATEC with existing targeted degradation approaches for extracellular proteins; and (ii) comparing T/I-AATECs with EVs presenting only with T & I cytokine receptors (i.e. no 3LIR) in disease models (is there added benefit of targeted degradation vs. neutralization?). I would still ask for the suggested control experiments and more details."

As you may already know, our editorial policy allows in principle a single round of major revision, so it is essential to provide responses to the reviewers' comments that are as complete as possible. Please feel free to contact me in case you would like to discuss in further detail any of the issues raised by the reviewers.

EMBO Molecular Medicine has a "scooping protection" policy, whereby similar findings that are published by others during review or revision are not a criterion for rejection. Should you decide to submit a revised version, I do ask that you get in touch after six months if you have not completed it, to update us on the status.

Please also contact us as soon as possible if similar work is published elsewhere. If other work is published, we may not be able to extend the revision period beyond six months.

I look forward to receiving your revised manuscript.

Sincerely,
Jingyi

Jingyi Hou
Editor
EMBO Molecular Medicine

We require:

- 1) A .docx formatted version of the manuscript text (including legends for main figures, EV figures and tables). Please make sure that the changes are highlighted to be clearly visible.
- 2) Individual production quality figure files as .eps, .tif, .jpg (one file per figure). For guidance, download the 'Figure Guide PDF': (<https://www.embopress.org/page/journal/17574684/authorguide#figureformat>).
- 3) A .docx formatted letter INCLUDING the reviewers' reports and your detailed point-by-point responses to their comments. As part of the EMBO Press transparent editorial process, the point-by-point response is part of the Review Process File (RPF), which will be published alongside your paper.

4) A complete author checklist, which you can download from our author guidelines (<https://www.embopress.org/page/journal/17574684/authorguide#submissionofrevisions>). Please insert information in the checklist that is also reflected in the manuscript. The completed author checklist will also be part of the RPF.

6) It is mandatory to include a 'Data Availability' section after the Materials and Methods. Before submitting your revision, primary datasets produced in this study need to be deposited in an appropriate public database, and the accession numbers and database listed under 'Data Availability'. Please remember to provide a reviewer password if the datasets are not yet public (see <https://www.embopress.org/page/journal/17574684/authorguide#dataavailability>).

.

12) Author contributions: You will be asked to provide CRediT (Contributor Role Taxonomy) terms in the submission system. These replace a narrative author contribution section in the manuscript.

13) A Conflict of Interest statement should be provided in the main text.

14) Every published paper now includes a 'Synopsis' to further enhance discoverability. Synopses are displayed on the journal webpage and are freely accessible to all readers. They include a short stand first (maximum of 300 characters, including space) as well as 2-5 one-sentence bullet points that summarize the paper. Please write the bullet points to summarize the key NEW findings. They should be designed to be complementary to the abstract - i.e. not repeat the same text. We encourage inclusion of key acronyms and quantitative information (maximum of 30 words / bullet point). Please use the passive voice. Please attach these in a separate file or send them by email, we will incorporate them accordingly.

15) All Materials and Methods need to be described in the main text using our 'Structured Methods' format. According to this format, the Methods section includes a Reagents and Tools Table (listing key reagents, experimental models, software and relevant equipment and including their sources and relevant identifiers) followed by a Methods and Protocols section describing the methods, ideally using a step-by-step protocol format. The aim is to facilitate adoption of the methodologies across labs.

Please download and fill our Reagents and Tools Table template (.docx), which you can find in our author guidelines: <https://www.embopress.org/page/journal/17574684/authorguide#structuredmethods>

An example of a Method paper with Structured Methods can be found here: <https://www.embopress.org/doi/10.15252/msb.20178071>.

***** Reviewer's comments *****

Referee #1 (Comments on Novelty/Model System for Author):

The animal models used in this manuscript is sophisticated and represent important validation of the developed technology, and provide sufficient support to its future applications.

Referee #1 (Remarks for Author):

In this manuscript, the authors engineered extracellular vesicles for the targeted degradation of soluble proteins by hijacking the autophagy-mediated lysosome degradation pathway. The developed modular autophagic adaptor targeting EV chimeras (AATEC) platform has been successfully applied to degrade TNF- α and IL-1 β , exhibiting potent anti-inflammatory effects in a rat and goat model of intervertebral disc degeneration. This paper is well-organized with comprehensive results of validation. I recommend accepting this paper after the following minor concerns have been addressed.

Scale bar is missed for Fig. 2j, Fig. 3h-l and Fig. S2g.

As shown in Fig. 4b-d, the proportion of dual-positive EVs is low (less than 10%). In the subsequent experiments, have the authors further isolated the dual-positive EVs or direct use the engineered EVs? Was there any comparison between these EVs?

How the particle numbers of EVs were determined for Fig. S5a-b and Fig. 4o?

Page 10, Fig. S4l, should the IL1R2-AATEC be changed to IL1R1-AATEC?

Referee #2 (Remarks for Author):

In this manuscript, the authors developed autophagic adaptor-targeting EV chimeras (AATECs) for targeted protein degradation. They identified the LIR motif as the essential signal to engage autophagosomes and to induce EV degradation in lysosomes. By incorporating triple LIR motifs and POI-targeting peptides into engineered EVs, they showed degradation of TNF- α and IL-1 β . The experiments are technically robust, supported by both in vitro and in vivo data, including large animal models. Overall, this method is of general interest, however I think for publication in a journal such as EMBO Molecular Medicine, a comparison to conventional lysosomal targeting agents is required. Benchmark this approach to the existing agents is required. In addition to the need for a comparison to lysosomal targeting agents, a number of specific points should be addressed before the paper is

suitable for publication.

1. This paper is so poorly organized. There is even no paragraphing under each subsection in the Result part. A lot of grammatical mistakes and typos are in the paper.
2. I'm wondering whether the amount of EVs produced in lab could meet the requirement for the goat experiment? The procedure for large-scale EV production should be described.
3. The EV degradation is not very convincing. First, GFP is unstable in low pH, so using it as a readout for degradation is not very accurate, as it would lose fluorescence in endosomes as well. In addition, some EVs can exocytose upon internalization - can the authors show that most of the EVs do not exocytose?
4. As shown in Fig 4q, some IL-1b were not sorted into lysosomes after AATEC treatment, as also indicated by the colocalization coefficient. Please clarify the localization of these IL-1b signals.
5. How long would these EVs be eliminated in vivo? Is it necessary to do repeated injections?

Referee #3 (Comments on Novelty/Model System for Author):

The model systems used are good but the controls used lack robustness, which makes interpreting the data difficult. Each single experiment is only as strong as the controls that are included and in this manuscript there are many controls that are missing, making interpretations difficult.

Referee #3 (Remarks for Author):

The paper describes a novel way of engineering extracellular vesicles as missiles for targeted degradation of extracellular proteins, such as cytokines. This is potentially very interesting approach for targeted degradation of extracellular proteins. However, although quite elaborate, this manuscript has conclusions that are not supported by experimental evidence and has many controls missing. If the authors address all of the experimental concerns, then the technology itself could be an excellent approach for targeting the degradation of many extracellular proteins, both for research and possibly therapeutically.

Specific concerns are as follows:

1. Scheme 1: The name AATEC sounds too similar to ATTEC and might lead to confusion. Please adopt an alternative name (EV-mediated Targeted Protein Degradation; EV-TPD perhaps?)
2. Please use positive controls to demonstrate that MG-132, BafA1, 3-MA (full form?) and ML282 are effective at the concentrations used and cite appropriate references for appropriate concentrations and specificity.
3. How were ATG7-KO cells validated for KO and functional consequences? These data are essential for any conclusions made here. How can you be certain these are not a result of clonal variation between WT and KO cells? Rescues with WT ATG7?
4. Why is the EV degradation of GFP in vivo so much longer than in cultured cells? Does this have any implications for potential therapeutic applications?
5. For biotinylation experiments, what were the target cells for EVs? The mass-spectrometry experiment might have been cleaner if they were done in a mouse target cell line (to only look for changes in human proteins sourced from HEK293 cells). Why are there so many protein changes, not just degraded but also stabilised? Was a Streptavidin IP also done in non-biotin EV treated cells or biotin treated EVs only to rule out false positives? One would expect all EV-associated proteins to be degraded.
6. How were experiments in 2e normalised? Any positive controls to control these? Perhaps blotting for IgGs might imply internalisation of antibodies bound to EVs. Would you expect the antibodies in the cytosol reducing environment to still bind to the respective EV targets? Did you test this?
7. Where are the knockdown controls for 2f? Did you verify the levels of SQSTM1 in EV surface?
8. 2g should include a negative control using EVs produced in SQSTM1^{-/-} (2h) or knockdown (2f) cells. Were any controls in intra-vesicular proteins blotted to show trypsin did not act inside EVs?
9. 2h - controls for SQSTM1^{-/-} cells missing. The best negative control for degradation of GFP is EVs produced from WT and SQSTM1^{-/-} cells applied to a multitude of target cells.
10. 2i - why were HEK293 EVs compared with HeLa? Perhaps both HeLa would be a better comparison. The differences are not huge considering SQSTM1 is proposed to be key for degradation.
11. 2j - IF negative controls for anti-LAMP2 antibody?
12. 2k - please comment on the long kinetics of degradation in rats in vivo and very small changes in the kinetics between WT and SQM controls.
13. 3b - why were so many protein interactors identified? Were any other proteins identified by MS tested for validation, as done for MAP1LC3?
14. S3c - the uptake and degrade blots look confusing. Why is uptake affected and not degradation as stated in the text?
15. 3e - relative to the levels of Flag-SQSTM1 or delta-SQSTM1, both seem to be pulled down by GST-LC3B.
16. No control blots/sequencing data for MAP1LC3B^{-/-} cells nor any functional validation of these cells.

17. 3h - validation of GFP-EEA1, Rab7a and calreticulin at the correct location first? No co-expression controls? Negative controls for PLA assays needed - given these often lead to false positive signals.
18. 3i - Validation of ATG7^{-/-}? No co-expression controls?
19. 3j - in the design for 3xLIR, an ideal negative control would be a mutant 3xLIR sequence that can no longer bind to LC3.
20. 3k - would have been better to include control EVs as well to see how degradation compared. Did you test the kinetics of degradation - i.e. one would expect more degradation even at the uptake timepoint of 12h. When you compare Figs. S1f, g, h to this, the degradation at the same time point is more robust without the need for additional SQSTM1. Additionally, here is an ideal place to test the requirements for ATG7 and MAP1LC3B^{-/-} cells as well, since the mechanism have been suggested just prior to this. Do the engineered degraders work with the same mechanisms that led to these degraders in the first place?
21. 4e - This is a key figure, but it does not clarify exactly what EVs were used here. What are non-target controls? Were nanoflow sorted EVs containing both target and 3LIR used or was it a mix pool of EVs, given ~5% of EVs contain both GFP and RFP signals? Similarly, what EVs were used for Fig. 4f and g? For 4f, g & h, what are the controls for BafA1, MG132 and MAP1LC3B KD?
22. Pg 9 - Fig. S4 in many places should be S5.
23. 3i-j: Precisely what are EV and AATEC controls? For this experiment, control EVs or those only presenting 3LIR or only presenting rTNFaR are ideal negative controls.
24. Fig. S5e: Did rTNFa-AATEC significantly improve IVD compared to EVs containing rTNFa alone?
25. The same comments that applied to EVs containing rTNFa alone TNFaR-AATEC from 21-24 apply to IL1R2-AATECs.
26. S5k,l: Exactly what was used here and what cell lines/models? IL1R1 binding domain or IL1R1 itself? If all the source of IL1R1 was the EVs, then degradation of EVs would result in reduction of IL1R1 anyway. So how is this targeting the degradation of receptors? This needs a lot more expansion and more controls to demonstrate targeted degradation of receptors proteins.
27. Fig. 5a: How was the quality of T/I-AATEC assessed relative to I- or T-AATECs? This is because, if you mixed I-AATEC EVs with T-AATECs (instead of having both in the same EVs), I suspect the outcome would be the same. Given that the percentage of EVs expressing two components in Fig. 4 (TNFR1 & 3LIR) was so low, one would expect even lower (negligible) EVs with all 3 components in one. It is not fully explained how these were purified, separated and applied. The authors need to demonstrate that EVs that have dual targeting are the ones that target both extracellular proteins and are better than individual or mixed pools of individual-targeting EVs. The same applies to all in vivo experiments where they should apply a 50% mixture of individual-targeting EVs (I- and T-AATEC) as controls - I suspect these would perform as well as I/T-AATEC EVs.
28. For Fig. 6 in vivo IVD degeneration assays, I am unable to comment on biology of IVD. However, the key question for me would be whether targeted degradation of TNF-a and IL-1b would be more efficacious than neutralisation of TNF-a and IL-1b. Therefore, key comparisons for me would have to be whether T/I-AATECs (degraders) perform better than those EVs that present TNF-aR and IL-1bR without the 3LIR (neutralisers). Here the T/I-AATECs are compared with unloaded EVs, making any conclusions difficult as the clinical efficacy could have been just as much due to neutralisation of the cytokines as their degradation. The dual degradation part is still underdeveloped and needs a lot more controls. The in vivo time-lines are a lot longer than the in-cellulo timelines for degradation - comments?

The manuscript also has a lot of editorial issues and would benefit from an editorial proof-reading. Furthermore, there are insufficient experimental details in many places making it hard to understand the experiment in question. This needs to be addressed thoroughly.

Response to Referees

Referee #1 (Comments on Novelty/Model System for Author):

The animal models used in this manuscript is sophisticated and represent important validation of the developed technology, and provide sufficient support to its future applications.

Reply: We sincerely appreciate your recognition of our work.

Referee #1 (Remarks for Author):

In this manuscript, the authors engineered extracellular vesicles for the targeted degradation of soluble proteins by hijacking the autophagy-mediated lysosome degradation pathway. The developed modular autophagic adaptor targeting EV chimeras (AATEC) platform has been successfully applied to degrade TNF- α and IL-1 β , exhibiting potent anti-inflammatory effects in a rat and goat model of intervertebral disc degeneration. This paper is well-organized with comprehensive results of validation. I recommend accepting this paper after the following minor concerns have been addressed.

Reply: Thank you very much for your recognition of our work. The detailed point-to-point response to your comments are provided in the following section.

1. Scale bar is missed for Fig. 2j, Fig. 3h-I and Fig. S2g.

Reply: We apologized for the missing of scale bar. We have added scale bars as thin white lines to the corresponding images to indicate the spatial dimensions accurately.

Fig. 2j Immunofluorescence images and colocalization analysis of EV-EGFP and intracellular Lamp II (scale bar: 10 μ m) (n=6).

Fig. 3h PLA assay were used to detect the interaction sites (Red) of EV-Flag-SQSTM1 and intracellular His-MAP1LC3B in HeLa cells. Meanwhile, HeLa cells were transfected with GFP-EEA1, GFP-Rab7a or GFP-Calreticulin to visualize subcellular location of interaction sites. Colocalization of PLA signals and GFP were analyzed by imageJ ($n=6$). Scale bar: 10 μm . **i**. The interaction sites (Red) of EV-Flag-SQSTM1 and intracellular His-MAP1LC3B in wild type and ATG7 KO HeLa cells were detected by PLA assay. Colocalization coefficient of PLA/GFP-Rab7a and relative fluorescence intensity of PLA signals were analyzed by imageJ ($n=6$). (Scale bar: 10 μm).

Fig. S2g Control or SQSTM1 KD EVs were added to NPCs and co-incubated for 12 h. Then, free EVs were washed out and replaced by free medium containing 50 nM BafA1 for 12h. After incubation, Immunofluorescence assay was used to detect the colocalization of EGFP with LampII (scale bar: 10 μm) ($n=6$).

2. As shown in Fig. 4b-d, the proportion of dual-positive EVs is low (less than 10%). In the subsequent experiments, have the authors further isolated the dual-positive EVs or directly use the engineered EVs? Was there any comparison between these EVs?

Reply: Thank you very much for your insightful question and suggestion. For the isolation of engineered EVs, we first collected and separated a mixed pool containing double-negative (blank), single-positive, and double-positive EVs. By fusing RFP at the C-terminal of the cytokine-targeting signal and GFP at the C-terminal of the lysosome-targeting signal, we detected the proportion of each subtype of EV in the mixed pool by nanoFCM. Subsequently, we used fluorescence-activated sorting technology (CytoFLEX SRT) to isolate double-positive EVs for subsequent experiments and applied nano-flow cytometry to detect the proportion of each subtype of EVs within the post-sorting EVs. The results showed that before sorting, the proportion of double-positive EVs was less than 10%, while after sorting, the proportion was close to 100%

(Appendix Fig. S3e), which confirmed the effectiveness of the EV engineering and sorting techniques.

Appendix Fig. S3e The loading efficiency of degradation signals (GFP) and targeting signals (RFP) on engineering EVs before and after fluorescence-activated sorting were detected using NanoFCM.

We described the sorting process of double-positive EVs in the Methods section as follows in **Page 19 Line 22-24**:

“For the isolation of engineered receptor-positive EVs, we used fluorescence-activated sorting technology (CytoFLEX SRT) to isolate the positive EVs”

3. How the particle numbers of EVs were determined for Fig. S5a-b and Fig. 4o?

Reply: Thank you very much for your question. We applied nano-flow cytometry to detect the particle number of extracted EVs and diluted them to the target concentrations for subsequent *in vitro* experiments.

We have added corresponding descriptions in Materials and Methods part in **Page 19 Line 21-22** to facilitate readers' understanding as follows:

“The particle number of EVs was determined by Nanoflow for subsequent experiments.”

Page10, Fig. S4l, should the IL1R2-AATEC be changed to IL1R1-AATEC?

Reply: We sincerely apologize for the writing errors and have corrected the description of our results in **Page 12 Line 10-11** as follows:

“The results showed that IL1R1-AATEC efficiently degraded IL1R1 (Fig. EV5 k), which confirmed the ability of AATEC to degrade cell surface targets.”

Fig. EV5k Protein level analysis of IL1R1 levels after EV or IL1R1-AATEC treatment for 24 h ($n=3$).

Referee #2 (Remarks for Author):

In this manuscript, the authors developed autophagic adaptor-targeting EV chimeras (AATECs) for targeted protein degradation. They identified the LIR motif as the essential signal to engage autophagosomes and to induce EV degradation in lysosomes. By incorporating triple LIR motifs and POI-targeting peptides into engineered EVs, they showed degradation of TNF- α and IL-1 β . The experiments are technically robust, supported by both in vitro and in vivo data, including large animal models. Overall, this method is of general interest, however I think for publication in a journal such as EMBO Molecular Medicine, a comparison to conventional lysosomal targeting agents is required. Benchmark this approach to the existing agents is required. In addition to the need for a comparison to lysosomal targeting agents, a number of specific points should be addressed before the paper is suitable for publication.

Reply: Thank you very much for your insightful question and suggestion.

- We chose the widely used Tri-GalNAc-based LYTAC technology as a control to compare the characteristics of AATEC in terms of targeted protein degradation ability and cell receptor dependence¹. As far as we know, currently there are no LYTAC reagents targeting TNF- α and IL-1 β . To prepare LYTAC reagents for targeted degradation of TNF- α and IL-1 β , we respectively used Adalimumab (a monoclonal antibody for TNF- α) and Canakinumab (a monoclonal antibody for IL-1 β) as the basis to prepare LYTAC reagents for targeted degradation of TNF- α and IL-1 β . Specifically, we first used the Azide protein labeling kit (Byotime) to label the azide groups for Adalimumab and Canakinumab, and then used the desalting column to purify the azidized monoclonal antibodies. Subsequently, we co-incubated the azide monoclonal antibodies with Tri-GalNAc-DBCO (MedChemExpress) in the click reaction solution to complete the click chemical reaction, allowing the Tri-GalNAc groups to be respectively coupled to Adalimumab and Canakinumab, thereby completing the preparation of Tri-GalNAc-Adalimumab (LYTAC reagent for TNF- α targeting) and Tri-GalNAc-Canakinumab (LYTAC reagent for IL-1 β targeting). Meanwhile, we used the same method to label human IgG (Byotime) with Tri-GalNAc to prepare Tri-GalNAc-IgG as the negative control.
- Next, we compared the targeting degradation capabilities of TNF α -AATEC and TNF α -LYTAC for TNF α in HepG2 cells, and used non-target AATEC without TNFR1 and IgG-LYTAC as negative controls. The results below showed that both TNF α -AATEC and TNF α -LYTAC could effectively degrade extracellular TNF- α , and their degradation capabilities were

comparable (Fig. 4 e). Meanwhile, the non-target AATEC and IgG-LYTAC did not have the ability to target and degrade TNF- α , which is consistent with the previously reported highly efficient liver-specific targeting degradation ability of LYTAC¹.

Fig. 4e ELISAs were used to detect the degradation rates of TNF- α in HepG2 cells by non-target AATEC, TNF α -AATEC, IgG-LYTAC and TNF α -LYTAC ($n=3$).

- c. As the targeted degradation ability of LYTAC relies on the recognition of ASGPR on the cell surface, we employed shRNA to knockout ASGPR in HepG2 cells and examined its influence on the targeted degradation capabilities of TNF α -AATEC and TNF α -LYTAC towards TNF α . Western blotting analysis confirmed that the ASGPR-targeted shRNA could significantly decrease the expression of ASGPR in HepG2 cells (Appendix Fig. S1 h). Compared with the group using non-specific shRNA, the knockdown of ASGPR led to a substantial reduction in the degradation ability of TNF α -LYTAC for TNF α , whereas the degradation ability of TNF α -AATEC remained unaffected (Fig. 4 f). This finding verified the dependence of Tri-GalNAc-based LYTAC degradation reagents on ASGPR. On the other hand, the degradation ability of AATEC was not influenced by the expression level of ASGPR.

Appendix Fig. S1h The ASGPR levels in HepG2 cells after shASGPR treatment were detected by Western blotting, and shScramble was used as a negative control.

Fig. 4f ELISAs were used to detect the degradation rates of TNF- α in HepG2 cells by TNF α -AATEC and TNF α -LYTAC in shNC and shASGPR groups ($n=3$).

- d. Meanwhile, we also compared the performance of IL1 β -LYTAC and IL1 β -AATEC in targeting and degrading IL-1 β in HepG2 cells. The results showed that the targeting degradation ability of IL1 β -AATEC was stronger than that of IL1 β -LYTAC, which was different from the result of TNF- α (Fig. 4 k). This suggests that for the AATEC platform, it has different degradation capabilities for different target proteins to be degraded, which may be due to the different binding strengths of the binding domains³.

Fig. 4k ELISAs were used to detect the degradation rates of IL-1 β in HepG2 cells by non-target AATEC, IL1 β -AATEC, IgG-LYTAC and IL1 β -LYTAC ($n=3$).

Reference:

1. Ahn, G., Banik, S. M., Miller, C. L., Riley, N. M., Cochran, J. R., & Bertozzi, C. R. (2021). LYTACs that engage the asialoglycoprotein receptor for targeted protein degradation. *Nature chemical biology*, 17(9), 937–946. <https://doi.org/10.1038/s41589-021-00770-1>
2. D'Souza, A. A., & Devarajan, P. V. (2015). Asialoglycoprotein receptor mediated hepatocyte targeting - strategies and applications. *Journal of controlled release : official journal of the Controlled Release Society*, 203, 126–139. <https://doi.org/10.1016/j.jconrel.2015.02.022>
3. Wells, J. A., & Kumru, K. (2024). Extracellular targeted protein degradation: an emerging modality for drug discovery. *Nature reviews. Drug discovery*, 23(2), 126–140. <https://doi.org/10.1038/s41573-023-00833-z>

We have added corresponding descriptions in Results part in **Page 10 Line 13-31** as follows: “Additionally, we chose the widely used Tri-GalNAc-based LYTAC technology as a control to compare the characteristics of AATEC in terms of targeted protein degradation ability and cell receptor dependence. To prepare LYTAC reagents for targeted TNF- α degradation, we used Adalimumab (monoclonal antibody for TNF- α) as the basis to prepare TNF- α -LYTAC. IgG-LYTAC was used as the negative control. Next, we compared the targeted protein degradation capacity of TNF- α -AATEC and TNF- α -LYTAC for TNF- α in HepG2 cells. Non-target AATEC without TNFR1 or IgG-LYTAC were used as negative controls. The results showed that both TNF- α -AATEC and TNF- α -LYTAC effectively degraded extracellular TNF- α with comparable degradation capacities (Fig. 4e). Meanwhile, non-specific AATEC and IgG-LYTAC did not exhibit the ability to target and degrade TNF- α . As the targeted degradation ability of LYTAC relies on the recognition of ASGPR, we used shRNA to knockdown ASGPR in HepG2 cells and examined its influence on the targeted degradation capabilities of TNF- α -AATEC and TNF- α -LYTAC towards TNF- α . Western blotting analysis confirmed that ASGPR-targeted shRNA significantly decreased ASGPR expression in HepG2 cells (Appendix Fig. S1h). ASGPR knockdown led to a substantial reduction in the degradation ability of TNF- α -AATEC for TNF- α compared with that in the non-specific shRNA group, whereas the degradation ability of TNF- α -AATEC remained unaffected (Fig. 4f). This finding verified the dependence of Tri-GalNAc-based LYTAC on ASGPR. However, the degradation ability of AATEC was not affected by ASGPR expression.”

1. This paper is so poorly organized. There is even no paragraphing under each subsection in the Result part. A lot of grammatical mistakes and typos are in the paper.

Reply: We apologize for the poor structure of the article. To facilitate understanding, we have divided the various sections of the results part of the article. At the same time, we have carefully checked and corrected the grammatical errors and spelling mistakes in the article.

2. I'm wondering whether the amount of EVs produced in lab could meet the requirement for the goat experiment? The procedure for large-scale EV production should be described.

Reply: Thank you very much for your insightful question and suggestion.

- a. For the production of EVs, we use an automatic EV extraction system (EXODUS H600), which can automatically isolate EV from a large amount of cell supernatant at a maximum speed of 200 mL/h, meeting the requirements for large-scale preparation of engineered EVs.
- b. Additionally, for the treatment of the goat intervertebral disc degeneration model, we adopt a local administration approach. Due to the relatively isolated and avascular state of the intervertebral disc, the drug clearance rate is slow and the drug absorption efficiency is high. The dosage used is 100 μ L per animal at a concentration of 1×10^{10} . The EXODUS H600 can meet the experimental requirements.

For clarity and ease of understanding, we have provided a detailed description of the method for EV production in the Methods section in **Page 19 Line 17-20** as follows:

“For the production of EVs, we use an automatic EV extraction system (EXODUS H600), which can automatically isolate EV from a large amount of cell supernatant at a maximum speed of 200 mL/h, meeting the requirements for large-scale preparation of engineered EVs.”

3. The EV degradation is not very convincing. First, GFP is unstable in low pH, so using it as a readout for degradation is not very accurate, as it would lose fluorescence in endosomes as well. In addition, some EVs can exocytose upon internalization - can the authors show that most of the EVs do not exocytose?

Reply: Thank you very much for your insightful question and suggestion.

- a. First, regarding the detection of GFP signals, due to its pH sensitivity, fluorescence is prone to quenching under conditions where the pH is lower than its own pKa (6)¹. The principle of this quenching is that the conformation of the GFP protein undergoes reversible changes under acidic conditions, but this does not lead to a change in the protein level of GFP. Therefore, although the GFP fluorescence signal is unstable under acidic conditions, the protein level of GFP is not directly affected by pH conditions. On the contrary, the change in GFP protein levels mainly depends on the initiation of the lysosomal protein degradation system. Therefore, we choose the protein level of GFP as the detection signal rather than its fluorescence signal. Specifically, we mainly use the western blot assay to detect its protein levels to avoid the influence of pH reduction on the GFP fluorescence signal; in addition, when conducting immunofluorescence experiments, we use a primary antibody against GFP to recognize the GFP signal and use a fluorescently labeled secondary antibody to indicate the location and relative intensity of GFP within the cell, thereby reducing signal loss caused by GFP fluorescence quenching.
- b. For EV exocytosis, previous literature has confirmed that after uptake, a portion of EVs will be exocytosed back to the extracellular space through the endosomal recycling process^{2,3}.

Therefore, the attenuation of intracellular EV signals may not only be due to EV degradation but also to their exocytosis. To detect the level of EV exocytosis and whether it is affected by the degradation system, we labeled EVs with CFSE and incubated them with cells for 12 hours to allow EV uptake. Then, we washed out the remaining uninternalized EVs in the extracellular fluid and detected the intracellular EV signal as the initial overall level. After 12 hours of incubation, we collected the supernatant and detected the CFSE signal to calculate the level of exocytosis of CFSE-EVs within 12 hours. By calculating the proportion of the CFSE signal from exocytosis to the initial total CFSE signal, we inferred the proportion of EVs that underwent exocytosis to the total internalized EVs. The results showed that the signal from exocytosed EVs accounted for 3-5% of the initial signal, and this proportion was not affected by the inhibition of the lysosomal degradation system by Bafilomycin A1⁴. This might be due to the endocytic sorting process of EVs before they entered the autophagic-lysosomal degradation system.

Appendix Fig. S1f The exocytosis levels of CFSE-labeled EVs were measured in both the control group (DMSO-treated) and the BafA1 pretreatment group, with the uptake level at 12 hours used to standardize the CFSE-EV signal within each group ($n=3$).

Reference:

- Mares, R. E., Meléndez-López, S. G., & Ramos, M. A. (2011). Acid-denatured Green Fluorescent Protein (GFP) as model substrate to study the chaperone activity of protein disulfide isomerase. *International journal of molecular sciences*, *12*(7), 4625–4636. <https://doi.org/10.3390/ijms12074625>
- Morad, G., Carman, C. V., Hagedorn, E. J., Perlin, J. R., Zon, L. I., Mustafaoglu, N., Park, T. E., Ingber, D. E., Daisy, C. C., & Moses, M. A. (2019). Tumor-Derived Extracellular Vesicles Breach the Intact Blood-Brain Barrier *via* Transcytosis. *ACS nano*, *13*(12), 13853–13865. <https://doi.org/10.1021/acsnano.9b04397>
- Tong, B., Liao, Z., Liu, H., Ke, W., Lei, C., Zhang, W., Liang, H., Wang, H., He, Y., Lei, J., Yang, K., Zhang, X., Li, G., Ma, L., Song, Y., Hua, W., Feng, X., Wang, K., Wu, X., Tan, L., ... Yang, C. (2023). Augmenting Intracellular Cargo Delivery of Extracellular Vesicles in Hypoxic Tissues through Inhibiting Hypoxia-Induced Endocytic Recycling. *ACS nano*, *17*(3), 2537–2553. <https://doi.org/10.1021/acsnano.2c10351>
- Yoshimori, T., Yamamoto, A., Moriyama, Y., Futai, M., & Tashiro, Y. (1991). Bafilomycin A1, a specific inhibitor of vacuolar-type H(+)-ATPase, inhibits acidification and protein degradation in lysosomes of cultured cells. *The Journal of biological chemistry*, *266*(26), 17707–17712.

We have added the corresponding results on **Page 4 Line 25-38** as follows:

“After EV uptake, a portion of EVs are exocytosed back to the extracellular space through the endosomal recycling process[30, 31]. Therefore, the attenuation of intracellular EV signals may not only be attributed to EV degradation but also to their exocytosis. To detect the level of EV exocytosis and whether it was affected by the degradation system, we labelled the EVs with CFSE and incubated them with cells for 12 h to allow EV uptake. Then, we washed the remaining uninternalized EVs from the extracellular fluid and detected the intracellular EV signal as the initial overall level. After an additional 12 h of incubation, we collected the supernatant and detected the CFSE signal to calculate the level of exocytosis of CFSE-EVs in 12 h. By calculating the proportion of the CFSE signal attributed to exocytosis to that of the initial total CFSE signal, we inferred the proportion of EVs that underwent exocytosis to that of the total internalized EVs. The results showed that the signal from exocytosed EVs accounted for 3–5% of the initial signal, and this proportion was not affected by the inhibition of lysosomal degradation by BafA1 (Appendix Fig. S3f). This result may be attributed to the endocytic sorting process of EVs before they enter the lysosomal degradation system.”

4. As shown in Fig 4q, some IL-1 β were not sorted into lysosomes after AATEC treatment, as also indicated by the colocalization coefficient. Please clarify the localization of these IL-1 β signals.

Reply: Thank you very much for your insightful question and suggestion.

We mainly observed the co-localization of IL-1 β with lysosomes after AATEC treatment to characterize the ability of AATEC to target and degrade IL-1 β . Interestingly, even after AATEC treatment, some IL-1 β did not co-localize with lysosomes. We speculate that there might be two reasons for this. The first is newly synthesized IL-1 β within the cell, which is basically not affected by AATEC and should mainly exist in the Golgi apparatus. The second is IL-1 β that has entered the endosome but has not yet entered the lysosome, which should exist in the endosomal structure. Therefore, we observed the co-localization of IL-1 β with the Golgi apparatus and endosomes to verify our hypothesis. The results showed that IL-1 β co-localized with the Golgi apparatus, and this co-localization was not affected by AATEC, indicating that AATEC mainly affects the degradation of IL-1 β rather than its generation. In contrast, IL-1 β co-localization with endosomes was rarely observed before AATEC treatment, but significantly increased after AATEC treatment, indicating that endosomes, as the pre-station for lysosomal sorting, play a role in the intermediate degradation of IL-1 β .

Appendix Fig. S2g Fluorescence colocalization analysis of IL1 β and Golga2 in HeLa cells after EV or IL1 β -

AATEC treatment for 24 h (Scale bar: 10 μ m) ($n=5$).

Appendix Fig. S2h Fluorescence colocalization analysis of IL1 β and EEA1 in HeLa cells after EV or IL1 β -AATEC treatment for 24 h (Scale bar: 10 μ m) ($n=5$).

We have added the corresponding results on **Page 11 Line 35-Page 12 Line 2** as follows:
“Notably, some IL-1 β did not co-localize with lysosomes even after IL1 β -AATEC treatment. This may be because of two reasons: (1) the newly synthesized IL-1 β in the cell is unaffected by AATEC and should primarily exist in the Golgi apparatus or (2) the IL-1 β that has entered the endosome is yet to enter the lysosome, which should exist in the endosomal structure. Therefore, we observed the co-localization of IL-1 β with the Golgi apparatus and endosomes, which verifies our hypothesis. These results showed that some IL-1 β co-localized with the Golgi apparatus and that this co-localization was not affected by IL1 β -AATEC (Appendix Fig. S2g). This indicates that IL1 β -AATEC primarily affects IL-1 β degradation rather than its generation. In contrast, IL-1 β co-localization with endosomes was rarely observed without IL1 β -AATEC but significantly increased after IL1 β -AATEC treatment (Appendix Fig. S2h). This indicates that endosomes play a role in the intermediate degradation of IL-1 β .”

5. How long would these EVs be eliminated *in vivo*? Is it necessary to do repeated injections?

Reply: Thank you very much for your insightful question and suggestion.

We have confirmed through *in vivo* experiments that the natural degradation of EVs in the local intervertebral disc takes 3 weeks, which is much longer than the 24-hour degradation process *in vitro*. The reason for this might be that the amount of EVs used in the *in vivo* experiment (1×10^{10} /mL) is much higher than that in the *in vitro* experiment (1×10^8 /mL). Additionally, the internal micro-environment of the intervertebral disc is relatively independent and avascular, with low metabolic activity and a small number of nucleus pulposus cells¹. These factors might contribute to the slower degradation rate of EVs within the intervertebral disc *in vivo*. Based on the unique slow degradation characteristics of EVs in the intervertebral disc, we did not conduct repeated dosing but rather single-dose administration and evaluated the effect after the treatment ended.

We have added the corresponding discussion on **Page 16 Line 15-21** as follows:

“Additionally, the *in vivo* experiments helped confirm that the duration for the natural degradation of EVs in the local IVD is 3 weeks, which is much longer than the 24 h observed for degradation *in vitro*. The reason may be attributed to the number of EVs used in the *in vivo* experiment ($1 \times$

10¹⁰ particles/mL), which is much higher than that used in the in vitro experiment (1 × 10⁸ particles/ mL). Additionally, the internal microenvironment of the IVD is relatively independent and avascular with low metabolic activity and a small number of NPCs[72]. These factors may have contributed to the slower in vivo EV degradation rate in the IVDs.”

Reference:

1. Risbud, M. V., & Shapiro, I. M. (2014). Role of cytokines in intervertebral disc degeneration: pain and disc content. *Nature reviews. Rheumatology*, 10(1), 44–56.
<https://doi.org/10.1038/nrrheum.2013.160>

Referee #3 (Comments on Novelty/Model System for Author):

The model systems used are good but the controls used lack robustness, which makes interpreting the data difficult. Each single experiment is only as strong as the controls that are included and in this manuscript there are many controls that are missing, making interpretations difficult.

Reply: Thank you very much for your insightful questions and valuable suggestions. In the corresponding experiments, we incorporated the necessary control groups in accordance with your recommendations to strengthen the validity of our conclusions, and we have provided detailed, point-by-point responses to each of your comments in the following sections.

Referee #3 (Remarks for Author):

The paper describes a novel way of engineering extracellular vesicles as missiles for targeted degradation of extracellular proteins, such as cytokines. This is potentially very interesting approach for targeted degradation of extracellular proteins. However, although quite elaborate, this manuscript has conclusions that are not supported by experimental evidence and has many controls missing. If the authors address all of the experimental concerns, then the technology itself could be an excellent approach for targeting the degradation of many extracellular proteins, both for reaserach and possibly therapeutically.

Reply: Thank you very much for your recognition and suggestions regarding our work. We have added the necessary control groups as per your suggestions to strengthen the corresponding conclusions.

Specific concerns are as follows:

1. Scheme 1: The name AATEC sounds too similar to ATTEC and might lead to confusion. Please adopt a alternative name (EV-mediated Targeted Protein Degradation; EV-TPD perhaps?)

Reply: Thank you very much for your insightful suggestion. We have changed all occurrences of AATEC in the full text to EV-mediated targeted protein degradation (EVTPD) to avoid confusion with ATTEC. Meanwhile, to facilitate understanding, in the remaining part of this reply letter, we will still continue to use the name “AATEC” to refer to the revised “EVTPD” to avoid confusion.

2. Please use positive controls to demonstrate that MG-132, BafA1, 3-MA (full form?) and ML282 are effective at the concentrations used and cite appropriate references for appropriate concentrations and specificity.

Reply: Thank you very much for your insightful suggestions. We have verified the usage concentrations and effects of these inhibitors as follows:

- a. MG132: MG132 has been reported as a proteasome inhibitor that can specifically inhibit proteasome-dependent protein degradation at the concentration of $20 \mu\text{M}$ ¹. We stimulated the nucleus pulposus cells with $20 \mu\text{M}$ MG132 for 12 hours and verified the inhibitory effect of MG132 on proteasome-mediated protein degradation by detecting the total protein ubiquitination level. The results showed that $20 \mu\text{M}$ MG132 could effectively inhibit the protein degradation mediated by the ubiquitin-proteasome pathway.

Appendix Fig. S1a HeLa cells were treated with 0, 10, and $20 \mu\text{M}$ MG132 for 12 hours, and the levels of protein ubiquitination were assessed using Western blot analysis.

- b. Bafilomycin A1 (BafA1): BafA1 has been reported to be able to inhibit the acidification process and degradation ability of lysosomes by selectively inhibiting vacuolar H^+ -ATPase at the concentration of 50nM ^{2,3}. We stimulated the nucleus pulposus cells with 50nM concentration of BafA1 for 12 hours, and used the autophagy reporter probe GFP-RFP-LC3 to indicate the lysosomal function. When the lysosomal acidification function was impaired, the degree of GFP fluorescence quenching dependent on acidity will decrease, resulting in an decreasing of RFP+ GFP- subset population. The results showed that the stimulation with 50nM concentration of BafA1 significantly inhibited the lysosomal degradation ability.

Appendix Fig. S1b HeLa cells expressing GFP-RFP-LC3 were treated with DMSO or 50nM BafA1 for 12 hours, and the intracellular GFP and RFP signals were detected using flow cytometry ($n=3$).

- c. 3-Methyladenine (3-MA): 3-MA has been reported to specifically inhibit the

autophagy/lysosome degradation pathway, and its effective concentration is 5 mM⁴. We stimulated the nucleus pulposus cells with 5 mM 3-MA for 12 hours, and used the autophagy reporter probe GFP-RFP-LC3 to indicate the autophagy activity. The results showed that 3-MA could effectively inhibit the autophagy level of the nucleus pulposus cells.

Appendix Fig. S1c HeLa cells expressing GFP-RFP-LC3 were treated with DMSO or 5 mM 3-Methyladenine for 12 hours, and the intracellular GFP and RFP signals were detected using flow cytometry ($n=3$).

- d. ML282 has been reported as a highly effective small molecule pan-inhibitor targeting the Ras superfamily GTPases, and it shows a significant inhibitory effect on Rab7⁵. We referred to previous reports and treated the cells with 20 μ M ML282 for 12 hours⁶. We then used Rab7a-pull down to detect the proportion of active Rab7a-GTP to assess the effectiveness of the stimulation. The results showed that ML282 could effectively inhibit the activity of Rab7a.

Appendix Fig. S1d HeLa cells were treated with 20 μ M ML282 for 12 hours. The levels of total Rab7 and active Rab7 (Rab7-GTP) were evaluated using Western blot with Rab7 antibody and Rab7-GTP-specific antibody, respectively ($n=3$).

Reference:

1. Lee, D. H., & Goldberg, A. L. (1998). Proteasome inhibitors: valuable new tools for cell biologists. *Trends in cell biology*, 8(10), 397–403. [https://doi.org/10.1016/s0962-8924\(98\)01346-4](https://doi.org/10.1016/s0962-8924(98)01346-4)
2. Gagliardi, S., Rees, M., & Farina, C. (1999). Chemistry and structure activity relationships of bafilomycin A1, a potent and selective inhibitor of the vacuolar H⁺-ATPase. *Current medicinal chemistry*, 6(12), 1197–1212.
3. Wang, R., Wang, J., Hassan, A., Lee, C. H., Xie, X. S., & Li, X. (2021). Molecular basis of V-ATPase inhibition by bafilomycin A1. *Nature communications*, 12(1), 1782. <https://doi.org/10.1038/s41467-021-22111-5>
4. Seglen, P. O., & Gordon, P. B. (1982). 3-Methyladenine: specific inhibitor of autophagic/lysosomal protein degradation in isolated rat hepatocytes. *Proceedings of the National Academy of Sciences of the United States of America*, 79(6), 1889–1892. <https://doi.org/10.1073/pnas.79.6.1889>
5. Hong, L., Simons, P., Waller, A., Strouse, J., Surviladze, Z., Ursu, O., Bologna, C., Gouveia,

- K., Agola, J. O., BasuRay, S., Wandinger-Ness, A., Sklar, L., Simpson, D. S., Schroeder, C. E., Golden, J. E., & Aubé, J. (2011). A small molecule pan-inhibitor of Ras-superfamily GTPases with high efficacy towards Rab7. In *Probe Reports from the NIH Molecular Libraries Program*. National Center for Biotechnology Information (US).
6. Zhang, T., Linghu, K. G., Tan, J., Wang, M., Chen, D., Shen, Y., Wu, J., Shi, M., Zhou, Y., Tang, L., Liu, L., Qin, Z. H., & Guo, B. (2024). TIGAR exacerbates obesity by triggering LRRK2-mediated defects in macroautophagy and chaperone-mediated autophagy in adipocytes. *Autophagy*, 20(8), 1741–1761. <https://doi.org/10.1080/15548627.2024.2338576>

We have added the corresponding description on Method section **Page 18 line 6-27** as follows:

Drug treatment

MG132: MG132 has been reported as a proteasome inhibitor that can specifically inhibit proteasome-dependent protein degradation at the concentration of 20 μ M[76]. We stimulated the NPCs with 20 μ M MG132 for 12 h and verified the inhibitory effect of MG132 on proteasome-mediated protein degradation by detecting the total protein ubiquitination level. The results showed that 20 μ M MG132 effectively inhibited the protein degradation mediated by the ubiquitin-proteasome pathway (Appendix Fig. S1a).

Bafilomycin A1 (BafA1): BafA1 has been reported to inhibit lysosomal acidification and degradation by selectively inhibiting vacuolar H⁺-ATPase at a concentration of 50 nM[77]. We stimulated NPCs with 50 nM BafA1 for 12 h and used the autophagy reporter probe GFP-RFP-LC3 to assess lysosomal function. When lysosomal acidification was impaired, GFP fluorescence increased. Stimulation with 50 nM BafA1 significantly inhibited the lysosomal degradation (Appendix Fig. S1b).

3-Methyladenine (3-MA): 3-MA specifically inhibits the autophagy/lysosome degradation pathway at an effective concentration of 5 mM[78]. We stimulated NPCs with 5 mM 3-MA for 12 h and used the autophagy reporter probe GFP-RFP-LC3 to indicate autophagy activity. The results showed that 3-MA effectively inhibited autophagy in the NPCs (Appendix Fig. S1c).

ML282 is a highly effective small-molecule pan-inhibitor targeting the Ras superfamily of GTPases and has a significant inhibitory effect on Rab7[33]. We referred to previous reports and treated the cells with 20 μ M ML282 for 12 h[79]. Using Rab7a GTPase-RILP, we detected the proportion of active Rab7a and assessed the effectiveness of the stimulation. The results showed that ML282 effectively inhibit Rab7a activity (Appendix Fig. S1d).

3. How were ATG7-KO cells validated for KO and functional consequences? These data are essential for any conclusions made here. How can you be certain these are not a result of clonal variation between WT and KO cells? Rescues with WT ATG7?

Reply: Thank you very much for your insightful suggestions.

- a. Regarding the validation of the ATG7 KO cell line, we used Sanger sequencing and Western blot assay to verify the KO efficiency. The results showed that the gRNA pairs-targeted fragment was effectively deleted and the expression of ATG7 protein was absent in the ATG7 KO cell line.

Appendix Fig. S7a Sanger sequencing was employed to assess the knockout efficiency of the target region flanked by the gRNA pairs in the ATG7 KO cell line. For reference, the corresponding DNA sequence of wild type ATG7 gene is presented above.

Appendix Fig. S1g The expression levels of ATG7 protein in wild-type and ATG7 KO cell lines was detected by western blotting ($n=3$).

- b. Furthermore, we conducted an ATG7 rescue experiment to rule out the influence of clonal variation on ATG7 KO cells. The results showed that expression of ATG7 in ATG7 KO cells could effectively restore the degradation efficiency of EVs in the cells. This further confirmed the role of the ATG7-dependent autophagy pathway in the degradation of EVs.

Appendix Fig. S1e Representative immunoblot images and quantitative analysis of the resident GFP level of HEK293T-derived EVs in the wild type, ATG7^{-/-} and ATG7 rescue Hela cell lines ($n=3$).

We have added the corresponding statements in the results section **Page 5 Line 12-15** as follows: “Furthermore, we conducted an ATG7 rescue experiment to rule out the influence of clonal variation in the ATG7 KO cells. The results showed that ATG7 expression in the ATG7 KO cells effectively restored the EV degradation efficiency (Appendix Fig. S1e) and confirmed the role of the autophagy pathway in EV degradation.”

4. Why is the EV degradation of GFP in vivo so much longer than in cultured cells? Does this have any implications for potential therapeutic applications?

Reply: Thank you very much for your insightful questions.

- a. We have confirmed through *in vivo* experiments that the natural degradation of EVs in the local intervertebral disc takes 3 weeks, which is much longer than the 24-hour degradation process *in vitro*. The reason for this might be that the amount of EVs used in the *in vivo* experiment (1×10^{10} / mL) is much higher than that in the *in vitro* experiment (1×10^8 / mL). Additionally, the internal micro-environment of the intervertebral disc is relatively independent and avascular, with low metabolic activity and a small number of nucleus pulposus cells¹. These factors might contribute to the slower degradation rate of EVs within the intervertebral disc *in vivo*.
- b. Based on the unique slow degradation characteristics of EVs in the intervertebral disc, we did not conduct repeated dosing but rather single-dose administration and evaluated the effect after the treatment ended.

We have added the corresponding discussion on **Page 16 Line 15-21** as follows:

“Additionally, the *in vivo* experiments helped confirm that the duration for the natural degradation of EVs in the local IVD is 3 weeks, which is much longer than the 24 h observed for degradation *in vitro*. The reason may be attributed to the number of EVs used in the *in vivo* experiment (1×10^{10} particles/mL), which is much higher than that used in the *in vitro* experiment (1×10^8 particles/ mL). Additionally, the internal microenvironment of the IVD is relatively independent and avascular with low metabolic activity and a small number of NPCs[72]. These factors may have contributed to the slower *in vivo* EV degradation rate in the IVDs.”

Reference:

1. Risbud, M. V., & Shapiro, I. M. (2014). Role of cytokines in intervertebral disc degeneration: pain and disc content. *Nature reviews. Rheumatology*, *10*(1), 44–56.
<https://doi.org/10.1038/nrrheum.2013.160>

5. For biotinylation experiments, what were the target cells for EVs? The mass-spectrometry experiment might have been cleaner if they were done in a mouse target cell line (to only look for changes in human proteins sourced from HEK293 cells). Why are there so many protein changes, not just degraded but also stabilised? Was a Streptavidin IP also done in non-biotin EV treated cells or biotin treated EVs only to rule out false positives? One would expect all EV-associated proteins to be degraded.

Reply: Thank you very much for your insightful suggestion.

- a. The target cells we used are human nucleus pulposus cells.
- b. Since we hypothesize that the interaction between EVs and intracellular proteins mediates the intracellular sorting process of EVs, we chose to use EVs derived from the same species, HEK293T cells, to preserve their interaction with intracellular proteins in human nucleus pulposus cells.
- c. Based on our mass spectrometry results, we found that the degradation of many EV proteins (such as SQSTM1) was inhibited by Baf A1, which is in line with our expected hypothesis and indicates that the lysosome-dependent degradation pathway has the ability to degrade these specific subgroups of EVs. However, in fact, the degradation of some EV proteins was not inhibited by Baf A1. This might be due to the fact that the degradation of these specific subgroups of EVs does not rely on the Baf A1-dependent pathway or they have a resistance to

the lysosomal degradation system. Interestingly, the attenuation of a small number of EV proteins was enhanced after the inhibition of Baf A1. We speculate that this unusual activation might be caused by the activation of the endosomal recycling system after the inhibition of lysosomal degradation, leading to the exocytosis of this part of EVs¹. These reasons need to be further confirmed by future research.

- d. To eliminate false positive signals, we followed your suggestion and added a non-biotin-labeled EV group, where non-labeled EVs were directly added to the cells and underwent the same treatment steps as the experimental group. Based on the mass-spectrometry results, we removed non-specifically enriched signals in non-biotin-labeled EV group to rule out false positive signals in the final results.

Reference:

1. Banushi, B., Joseph, S. R., Lum, B., Lee, J. J., & Simpson, F. (2023). Endocytosis in cancer and cancer therapy. *Nature reviews. Cancer*, 23(7), 450–473. <https://doi.org/10.1038/s41568-023-00574-6>

We have added the main conclusions drawn from the above content as a supplement to the discussion section in **Page 16 Line 1-14** as follows:

“We hypothesized that the interaction between EVs and intracellular proteins mediates the intracellular EV sorting process. To validate this, we chose to use EVs derived from the same species (for example, HEK293T cells) to preserve their interaction with intracellular proteins in human NPCs. A non-biotin-labeled EV group was included to eliminate false-positive signals. The mass spectrometry results showed that the degradation of several EV proteins (such as SQSTM1) was inhibited by BafA1, which concurs with our hypothesis and indicates that the lysosome-dependent degradation pathway could degrade these specific EV subgroups. However, BafA1 did not inhibit the degradation of certain EV proteins, which may be attributed to the fact that the degradation of these specific EV subgroups does not rely on the BafA1-dependent pathway or that these EVs are resistant to the lysosomal degradation system. Notably, the attenuation of a small number of EV proteins was enhanced after BafA1 inhibition. We speculate that this unusual activation may be caused by the activation of the endosomal recycling system after the inhibition of lysosomal degradation, leading to the exocytosis of these EVs[72]. Future studies are warranted to confirm these inferences.”

6. How were experiments in 2e normalised? Any positive controls to control these? Perhaps blotting for IgGs might imply internalisation of antibodies bound to EVs. Would you expect the antibodies in the cytosol reducing environment to still bind to the respective EV targets? Did you test this?

Reply: Thank you very much for your insightful question and suggestion.

- a. The normalization methods for each group in Fig 2e are as follows: First, we calculated the remaining GFP levels based on the GFP signals of each group after uptake (as shown in Fig. S2 a) and degradation. Then, normalize the remaining GFP levels of the other groups using the remaining GFP level of the control group (i.e., the IgG group) as the reference. The normalized remaining GFP levels of each group shown in Fig 2e are obtained in this way.

- b. Our experiment mainly aims to screen EV surface proteins with degron characteristics through the specific blocking effect of antibodies. However, the degradation mechanism of EVs is still unclear at present. To our knowledge, no research has clearly pointed out which protein can act as a degron. Therefore, we are unable to add an appropriate positive control group.
- c. Antibodies have non-specific binding, which can lead to changes in EV uptake efficiency¹. To eliminate the bias caused by this effect, we used IgG as a negative control to exclude the impact of non-specific binding on EV uptake efficiency. However, as endosomes acidify and mature, the binding of antibodies to EVs is disrupted, so the blocking effect of antibodies on EVs is limited to before the maturation of endosomes². According to the results we obtained, the blocking of SQSTM1 antibodies can significantly inhibit the degradation efficiency of EVs. Therefore, we speculate that the timing of SQSTM1-mediated EV sorting should occur before the full acidification of endosomes.

Reference:

1. Wiklander, O. P. B., Mamand, D. R., Mohammad, D. K., ... El Andaloussi, S. (2024). Antibody-displaying extracellular vesicles for targeted cancer therapy. *Nature biomedical engineering*, 8(11), 1453–1468. <https://doi.org/10.1038/s41551-024-01214-6>
2. Igawa, T., Ishii, S., Tachibana, T., Maeda, A., ... Hattori, K. (2010). Antibody recycling by engineered pH-dependent antigen binding improves the duration of antigen neutralization. *Nature biotechnology*, 28(11), 1203–1207. <https://doi.org/10.1038/nbt.1691>

We have added the main conclusions drawn from the above content as a supplement to the result section in **Page 6 Line 35-38** as follows:

“The acidification and maturing of the endosomes disrupts the antibody–EV binding; therefore, the effect of EV blocking by the antibodies is limited to the period before endosomal maturation[34]. As SQSTM1 blocking significantly inhibits the EV degradation efficiency, we hypothesized that SQSTM1-mediated EV sorting occurs before the full acidification of the endosomes.”

7. Where are the knockdown controls for 2f? Did you verify the levels of SQSTM1 in EV surface?

Reply: Thank you very much for your insightful question and suggestion.

- a. We utilized PLKO.1-Scrambled¹ encoding non-targeting random sequence as a negative control (shNC in Fig. 2f) to minimize potential biases introduced by the transfection process, shRNA generation, and related experimental factors on the efficiency of EV uptake and sorting.
- b. To specifically detect the level of SQSTM1 on the surface of EVs, we stained EVs with anti-SQSTM1 antibody and measured the level of SQSTM1 on the surface of EVs by NanoFCM. The results confirmed that the use of shRNA targeting SQSTM1 significantly reduced the levels of SQSTM1 on the surface of EVs compared with shNC control group.

Appendix Fig. S1i The proportion of SQSTM1-positive EVs in the shNC and shSQSTM1 groups were quantitatively detected using NanoFCM ($n=3$).

Reference:

1. Fischer, J. W., Busa, V. F., Shao, Y., & Leung, A. K. L. (2020). Structure-Mediated RNA Decay by UPF1 and G3BP1. *Molecular cell*, 78(1), 70–84.e6. <https://doi.org/10.1016/j.molcel.2020.01.021>

We have added the main conclusions drawn from the above content as a supplement to the result section in **Page 6 Line 13-15** as follows:

“pLKO.1-Scrambled encodes a nontargeting random sequence, and this was used as the negative control. The knockdown efficacy was verified by western blotting (Appendix Fig. S8 a–f).”

8. 2g should include a negative control using EVs produced in SQSTM1^{-/-} (2h) or knockdown (2f) cells. Were any controls in intra-vesicular proteins blotted to show trypsin did not act inside EVs?

Reply: Thank you very much for your insightful question and suggestion.

- a. To verify the specificity of SQSTM1 staining, we collected EVs from SQSTM1^{-/-} cells and detected the expression of SQSTM1 using immunotransmission electron microscopy. The results showed that EVs produced by SQSTM1^{-/-} cells did not exhibit a positive signal, confirming the effectiveness of SQSTM1 knockout and the specificity of SQSTM1 staining.

Fig. 2g The expression of SQSTM1 in EVs was detected by immunotransmission electron microscopy (Scale bar: 100 nm). EVs derived from SQSTM1^{-/-} cells were used as negative control.

- b. We employed ALIX, a well-established intraluminal EV marker¹, as a representative indicator to assess changes in intraluminal protein levels, as shown in Fig. 2h. The results demonstrated that treatment with trypsin alone did not alter the levels of ALIX in EVs, thereby confirming that trypsin does not target intraluminal proteins within EVs in the absence of a permeabilizing agent.

Fig. 2h 0.25% Trypsin was added at a one-quarter volume ratio to digest HEK293T-derived EVs with or without 1% Triton-X 100 at 37 ° C for 2 hours. EV without trypsin degradation was used as a negative control. The expression of SQSTM1, CD63 and ALIX of the resulting EVs were examined by Western blot assays.

Reference:

1. Baietti, M. F., Zhang, Z., Mortier, E., Melchior, A., Degeest, G., Geeraerts, A., Ivarsson, Y., Depoortere, F., Coomans, C., Vermeiren, E., Zimmermann, P., & David, G. (2012). Syndecan-syntenin-ALIX regulates the biogenesis of exosomes. *Nature cell biology*, 14(7), 677–685. <https://doi.org/10.1038/ncb2502>

We have added the main conclusions drawn from the above content as a supplement to the Results section in Page 6 Line 28-30 and Page 6 Line 32-35 as follows:

“EVs produced by SQSTM1^{-/-} cells were used as negative controls to confirm the specificity of SQSTM1 staining (Fig. EV2 d).”

“ALIX is a well-established intraluminal EV marker that has been used as a representative indicator of changes in intraluminal protein levels[36]. Treatment with trypsin alone did not alter the ALIX levels in EVs, which confirmed that trypsin alone did not target intraluminal proteins within EVs (Fig. 2h).”

9. 2h - controls for SQSTM1^{-/-} cells missing. The best negative control for degradation of GFP is EVs produced from WT and SQSTM1^{-/-} cells applied to a multitude of target cells.

Reply: Thank you very much for your insightful suggestion.

We treated nucleus pulposus cells, Hela cells, and HEK293T cells with GFP-EV generated from WT and SQSTM1^{-/-} cells. The results showed that regardless of the type of cells used, the knockout of SQSTM1 effectively inhibited the degradation efficiency of EVs. This further confirmed the important role of SQSTM1 in the degradation process of EVs.

Appendix Fig. S1a-c Representative immunoblot images and quantitative analysis of the resident GFP level of wild type and SQSTM1^{-/-} EVs in the HeLa, HEK293T and nucleus pulposus cells (n=3).

We have added the main conclusions drawn from the above content as a supplement to the result section in **Page 7 Line 2-6** as follows:

“Moreover, we treated NPCs, HeLa cells, and HEK293T cells with GFP-EVs generated from WT and SQSTM1^{-/-} cells. These results showed that regardless of the type of cells used, SQSTM1 KO in the EVs effectively inhibited their degradation efficiency (Fig. 2i and Appendix Fig. S1 l-m).”

10. 2i - why were HEK293 EVs compared with HeLa? Perhaps both HeLa would be a better comparison. The differences are not huge considering SQSTM1 is proposed to be key for degradation.

Reply: Thank you very much for your insightful suggestion.

We conducted experiments using GFP-EVs derived from HeLa cells to verify our hypothesis. The results showed that knocking out SQSTM1 in the receptor HeLa cells did not affect the degradation process of EVs, which was consistent with the conclusion obtained in HEK293T cells.

Appendix Fig. S1m Representative immunoblot images and quantitative analysis of the resident GFP level of HeLa-derived EVs in the wild type and *SQSTM1*^{-/-} HeLa cell lines ($n=3$).

We have added the main conclusions drawn from the above content as a supplement to the result section in **Page 6 Line 41-43** and **Page 7 Line 1** as follows:

“Additionally, we used HeLa-derived GFP EVs to minimize potential bias from the difference between the cell lines. The results showed that *SQSTM1* knockout in recipient HeLa cells did not influence the degradation of HeLa-derived EVs, which is consistent with the results obtained for HEK293T-derived EVs (Appendix Fig. S1 m).”

11. 2j - IF negative controls for anti-LAMP2 antibody?

Reply: Thank you very much for your insightful suggestion.

We included a negative control group in which the anti-LAMP2 primary antibody was omitted, while the corresponding secondary antibody was applied. The results demonstrated that this control group displayed no detectable fluorescence signal, further supporting the specificity of LAMP2 staining.

Appendix Fig. S2a Representative fluorescence images show the localization of Lamp2 in HeLa cells. The negative control group used IgG antibody as the primary antibody (Scale bar: 10 μ m).

12. 2k - please comment on the long kinetics of degradation in rats *in vivo* and very small changes in the kinetics between WT and SQM controls.

Reply: Thank you very much for your insightful suggestion.

a. We have confirmed through *in vivo* experiments that the natural degradation of EVs in the local intervertebral disc takes 3 weeks (Fig. 2k), which is much longer than the 24-hour

degradation process *in vitro*. The reason for this might be that the amount of EVs used in the *in vivo* experiment (1×10^{10} /mL) is much higher than that in the *in vitro* experiment (1×10^8 /mL). Additionally, the internal micro-environment of the intervertebral disc is relatively independent and avascular, with low metabolic activity and a small number of nucleus pulposus cells¹. These factors might contribute to the slower degradation rate of EVs within the intervertebral disc *in vivo*. Based on the unique slow degradation characteristics of EVs in the intervertebral disc, we did not conduct repeated dosing but rather single-dose administration and evaluated the effect after the treatment ended.

We have added the corresponding discussion on **Page 16 Line 15-21** as follows:

“Additionally, the *in vivo* experiments helped confirm that the duration for the natural degradation of EVs in the local IVD is 3 weeks, which is much longer than the 24 h observed for degradation *in vitro*. The reason may be attributed to the number of EVs used in the *in vivo* experiment (1×10^{10} particles/mL), which is much higher than that used in the *in vitro* experiment (1×10^8 particles/ mL). Additionally, the internal microenvironment of the IVD is relatively independent and avascular with low metabolic activity and a small number of NPCs[72]. These factors may have contributed to the slower *in vivo* EV degradation rate in the IVDs.”

- b. Based on the results of *in vivo* experiments, the overexpression of SQSTM1 significantly promotes the degradation of EVs *in vivo* (Fig. 2k). This effect was evident as early as the 5th day and persisted until the 15 day when most of the EVs in both groups had been degraded, indicating that SQSTM1 played an important role in the degradation process of EVs *in vivo*. However, the overexpression of SQSTM1 did not significantly alter the degradation trend of EVs. This might be related to the limited metabolic activity of cells in the intervertebral disc and the limitation of autophagy flux¹. Our previous experimental results also confirmed that the degradation process of EV depends on the occurrence of autophagy (Fig. 1).

We have added the corresponding discussion on **Page 16 Line 21-26** as follows:

“The *in vivo* experiments show that SQSTM1 overexpression significantly promotes EV degradation *in vivo*. This effect was evident as early as day 5 and persisted until day 15 when most of EVs in both groups were degraded. This indicates that SQSTM1 plays an important role in EV degradation *in vivo*. However, SQSTM1 overexpression did not significantly alter the EV degradation trend. This may be related to the limited metabolic activity of cells in the IVD and the limitation of autophagic flux[72].”

Reference:

1. Risbud, M. V., & Shapiro, I. M. (2014). Role of cytokines in intervertebral disc degeneration: pain and disc content. *Nature reviews. Rheumatology*, *10*(1), 44–56.
<https://doi.org/10.1038/nrrheum.2013.160>

13. 3b - why were so many protein interactors identified? Were any other proteins identified by MS tested for validation, as done for MAP1LC3?

Reply: Thank you very much for your insightful questions and valuable suggestions.

- a. In the IP-MS experiment of Fig. 3b, we used IgG as a negative control to eliminate non-specific enrichment signals, and used anti-Flag antibody to specifically detect the protein

interaction profiles of EV-SQSTM1 in recipient cell. The results showed that various proteins, including those related to autophagosomes, endoplasmic reticulum, and mitochondria, were specifically enriched. This indicates that EV-SQSTM1 has a wide and complex protein interaction network within the cell, and the potential biological significance still needs to be further clarified in future studies.

- b. Our selection of MAP1LC3 was primarily based on two key considerations. First, according to the results of the IP-MS analysis, MAP1LC3 exhibited the highest level of enrichment among proteins other than SQSTM1, which was used as the bait protein. This suggests that MAP1LC3 is a major interacting partner of SQSTM1. Second, given our previous findings that autophagy plays a critical role in the degradation of EVs (Fig. 1), and considering that MAP1LC3 functions as an adaptor protein in autophagy by facilitating the delivery of cargo into the autophagic pathway¹, it is a plausible candidate for mediating the entry of EVs into the autophagy-dependent degradation pathway. However, we did not perform further experimental validation on whether other identified proteins are involved in the degradation of EVs, and their potential roles remain to be elucidated in future studies.

Reference:

1. Li, X., He, S., & Ma, B. (2020). Autophagy and autophagy-related proteins in cancer. *Molecular cancer*, 19(1), 12. <https://doi.org/10.1186/s12943-020-1138-4>

We added the above content as a part of the discussion to the Page 14 Line 21-30 as follows:

“We chose MAP1LC3 from the potential SQSTM1 interacting protein spectrum (identified via IP-MS) primarily because of two key considerations: (1) MAP1LC3 exhibited the highest level of enrichment among proteins in the IP-MS analysis only next to that of SQSTM1, which was used as the bait protein. This suggests that MAP1LC3 is a major interacting partner of SQSTM1; (2) as autophagy plays a critical role in EV degradation (Fig. 1) and as MAP1LC3 functions as an adaptor protein in autophagy by facilitating the delivery of cargo into the autophagic pathway[74], MAP1LC3 was considered a plausible candidate for mediating EV entry into the autophagy-dependent degradation pathway. However, we did not perform further experimental validation to determine whether other identified proteins are involved in EV degradation, and their potential roles need to be elucidated in future studies.”

14. S3c - the uptake and degrade blots look confusing. Why is uptake affected and not degradation as stated in the text?

Reply: Thank you very much for your questions. We sincerely apologize for the incorrect labeling. There was an error in the grouping labels of the uptake group and the degradation group in the original Fig. S3c. We have corrected the relevant content and presented the updated Fig. EV3c as follows:

Fig. EV3c SQSTM1 and its truncated form loaded EVs were incubated with NPCs for 12 h. Uptake level was analysed at this time while degradation level was detected following wash out and another 12 h incubation ($n=3$)

15. 3e - relative to the levels of Flag-SQSTM1 or delta-SQSTM1, both seem to be pulled down by GST-LC3B.

Reply: Thank you very much for your insightful question.

Fig. 3e demonstrates that overexpression of full-length SQSTM1 in EVs enhances their binding capacity to GST-LC3B. The results show that, compared to the control group, EVs overexpressing Flag-SQSTM1 exhibit significantly increased binding to GST-LC3B, as indicated by a marked elevation in the TGS101 signal, an EV marker. In contrast, EVs overexpressing the truncated SQSTM1 variant lacking the LIR domain display pull-down levels comparable to those of the control group, with no significant difference in TSG101 signal intensity observed between the two groups. The detection of the truncated SQSTM1 signal may be attributed to low-level background expression of SQSTM1 on the EV surface, resulting in the passive pull down of a limited number of truncated SQSTM1 loaded EVs by GST-LC3B. Collectively, these findings confirm that SQSTM1 overexpression promotes the interaction between EVs and LC3B, and this effect is dependent on the LIR domain of SQSTM1.

Fig. 3e GST or GST-LC3B were used to pull down blank EVs, SQSTM1-overexpressed EVs or truncated SQSTM1-overexpressed EVs.

We added the above content as a part of the discussion on **Page 8 Line 8-20** as follows:

“To further verify that EVs directly bind to MAP1LC3B via SQSTM1, we extracted SQSTM1-loaded EVs and GST-LC3B proteins and used GST-LC3B to pull down SQSTM1-EVs *in vitro*.”

Blank EVs were used in the control group. The results showed that EVs overexpressing Flag-SQSTM1 exhibited a significantly increased binding capacity for GST-LC3B, as evidenced by a pronounced increase in the TSG101 (EV marker) signal (Fig. 3e). In contrast, the pull-down levels of EVs overexpressing truncated SQSTM1 were comparable to those observed in the control group with no significant difference in TSG101 signal intensity. The detection of the truncated SQSTM1 signal may be attributed to the low-level background expression of SQSTM1 on the EV surface, resulting in the passive pull-down of a limited number of truncated SQSTM1-loaded EVs by GST-LC3B. However, SQSTM1 knockdown in EVs reduced EV binding to GST-LC3B (Fig. 3f). These findings confirm that SQSTM1 overexpression promotes EV-LC3B interaction and that this effect is dependent on the LIR domain of SQSTM1.”

16. No control blots/sequencing data for MAP1LC3B^{-/-} cells nor any functional validation of these cells.

Reply: Thank you very much for your insightful question.

Regarding the validation of the MAP1LC3B^{-/-} cell line, we used Sanger sequencing to verify the KO efficiency. The results showed that the gRNA pairs-targeted fragment was effectively deleted and the expression of MAP1LC3B protein was absent in the MAP1LC3B KO cell line (Appendix Fig. S7 b). Meanwhile, the construction and validation of the other KO cell lines (ATG7, SQSTM1, MAP1LC3B2) involved in this work are also presented in Appendix Fig. S7 as shown below:

Appendix Fig. S7a-d Sanger sequencing of the knockout fragment of ATG7 KO (a), SQSTM1 KO (b), MAP1LC3B (c), MAP1LC3B2 KO (d) cell lines. The corresponding wild-type DNA sequences were aligned.

We added the above content as a part of the method to the Page 21 Line 30-31 as follows:
“Selection of single clones were performed after 2 weeks while the selected KO clones were

validated by Sanger sequencing (Appendix Fig. S7 a-d)."

17. 3h - validation of GFP-EEA1, Rab7a and calreticulin at the correct location first? No co-expression controls? Negative controls for PLA assays needed - given these often lead to false positive signals.

Reply: Thank you very much for your insightful questions and valuable suggestions.

- a. To validate the subcellular localization specificity of GFP-EEA1¹ (an early endosome marker), GFP-Rab7a² (a late endosome marker), and GFP-Calreticulin³ (an endoplasmic reticulum marker), we co-expressed commercially available fluorescent labeling reagents, including Early Endosome-RFP (Invitrogen), Late Endosome-RFP (Invitrogen), and ER-RFP (Invitrogen), as reference markers to confirm the accurate localization of GFP-tagged protein expression. The results demonstrated that GFP-EEA1, GFP-Rab7a and GFP-Calreticulin effectively colocalized with Early Endosome-RFP, Late Endosome-RFP and ER-RFP respectively, thereby verifying the specificity and reliability of these fluorescently labeled proteins in marking early endosomes, late endosomes and the endoplasmic reticulum.

Appendix Fig. S2c Representative fluorescence image of HeLa cells co-expressing GFP-EEA1 and RFP-Early Endosome (Scale bar: 10 μ m).

Appendix Fig. S2d Representative fluorescence image of HeLa cells co-expressing GFP-Calreticulin and RFP-ER (Scale bar: 10 μ m).

Appendix Fig. S2e Representative fluorescence image of HeLa cells co-expressing GFP-Rab7 and RFP-Late Endosome (Scale bar: 10 μ m).

- b. To minimize the risk of false-positive signals in the PLA assay, a negative control group was established by omitting SQSTM1^{Flag} EVs, while following all other experimental procedures as per the standard PLA protocol. The absence of detectable PLA signals in the negative control group confirmed the specificity of the PLA assay.

Appendix Fig. S2b PLA assay were used to detect the interaction sites (Red) of EV-Flag-SQSTM1 and intracellular His-MAP1LC3B in HeLa cells. The group without SQSTM1^{Flag}-EVs addition was set as the negative control group (Scale bar: 10µm).

Reference:

1. Mu, F. T., Callaghan, J. M., Steele-Mortimer, O., Stenmark, H., Parton, R. G., Campbell, P. L., McCluskey, J., Yeo, J. P., Tock, E. P., & Toh, B. H. (1995). EEA1, an early endosome-associated protein. EEA1 is a conserved alpha-helical peripheral membrane protein flanked by cysteine "fingers" and contains a calmodulin-binding IQ motif. *The Journal of biological chemistry*, 270(22), 13503–13511. <https://doi.org/10.1074/jbc.270.22.13503>
2. Skjeldal, F. M., Haugen, L. H., Mateus, D., Frei, D. M., Rødseth, A. V., Hu, X., & Bakke, O. (2021). De novo formation of early endosomes during Rab5-to-Rab7a transition. *Journal of cell science*, 134(8), jcs254185. <https://doi.org/10.1242/jcs.254185>
3. Michalak M. (2024). Calreticulin: Endoplasmic reticulum Ca²⁺ gatekeeper. *Journal of cellular and molecular medicine*, 28(5), e17839. <https://doi.org/10.1111/jcmm.17839>

We added the above content as a part of the result in **Page 8 Line 39-43** and **Page 9 Line 1-2** as follows:

“The early endosome marker GFP-EEA, late endosome marker GFP-Rab7a, and endoplasmic reticulum marker GFP-Calreticulin were found to have been accurately distributed to their respective organelles (Appendix Fig. S2 c–e). We minimized the risk of false-positive signals in the PLA assay by establishing a negative control group by omitting the anti-Flag and anti-His primary antibodies and followed the rest of the standard PLA protocol. The absence of detectable PLA signals in the negative control group confirmed the specificity of the PLA assay (Appendix Fig. S2 b).”

18. 3i - Validation of ATG7-/-? No co-expression controls?

Reply: Thank you very much for your insightful questions and valuable suggestions.

- a. Regarding the validation of the ATG7^{-/-} cell line, we used Sanger sequencing and Western blot assay to verify the knockout efficiency. The results showed that the gRNA pairs-targeted fragment was effectively deleted and the expression of ATG7 protein was absent in the ATG7^{-/-} cell line.

Appendix Fig. S7a Sanger sequencing was employed to assess the knockout efficiency of the target region flanked by the gRNA pairs in the ATG7 KO cell line. For reference, the corresponding DNA sequence of wild type ATG7 gene is presented above.

Appendix Fig. S1g The expression levels of ATG7 protein in wild-type and ATG7 KO cell lines was detected by Western blotting.

- b. To verify the effectiveness of GFP-Rab7a labeling in ATG7^{-/-} cells, we co-expressed GFP-Rab7a and Late Endosome-RFP (a commercial late endosome marker, Invitrogen) in ATG7^{-/-} cells. The results showed that GFP-Rab7a and LE-RFP were still highly co-localized in ATG7 knockout cells, indicating that GFP-Rab7a can effectively label late endosomes in ATG7 knockout cells.

Appendix Fig. S2f Representative fluorescence images show the localization of GFP-Rab7, RFP-Late Endosome in wild type and ATG7^{-/-} HeLa cells (Scale bar: 10 μm).

19. 3j - in the design for 3xLIR, an ideal negative control would be a mutant 3xLIR sequence that

can no longer bind to LC3.

Reply: Thank you very much for your insightful suggestions.

We generated a fusion protein consisting of the L341V-mutated 3xLIR sequence (a previously reported LC3-binding-deficient mutation¹) and the C-terminus of CD63, and evaluated the effect of this genetic engineering on the degradation efficiency of EVs. Unmodified blank EVs were employed as negative controls, whereas EVs carrying the wild-type 3x LIR sequence served as positive controls. The results demonstrated that the incorporation of the mutant 3xLIR motif failed to significantly improve the degradation efficiency of EVs compared to the negative control, thereby confirming that the 3x LIR motifs can effectively facilitate EV degradation through binding to MAP1LC3B.

Fig. EV4v Representative immunoblot images and quantitative analysis of the resident GFP level of blank, 3x LIR motifs and 3x L341V-mutated LIR motifs engineering EVs in HeLa cells ($n=4$).

Reference:

1. Goode, A., Butler, K., Long, J., Cavey, J., Scott, D., Shaw, B., Sollenberger, J., Gell, C., Johansen, T., Oldham, N. J., Searle, M. S., & Layfield, R. (2016). Defective recognition of LC3B by mutant SQSTM1/p62 implicates impairment of autophagy as a pathogenic mechanism in ALS-FTLD. *Autophagy*, 12(7), 1094–1104. <https://doi.org/10.1080/15548627.2016.1170257>

We added the above content as a part of the result in Page 9 Line 1-7 as follows:

“Moreover, we generated a fusion protein comprising the L341V-mutated triple LIR sequence (previously reported LC3-binding-deficient mutation[47]) and C-terminal of CD63 and evaluated its effect on EV degradation efficiency. Unmodified blank EVs were used as negative controls, whereas EVs carrying the wild-type triple LIR motifs were used as positive controls. The results indicated that the incorporation of the mutant triple LIR motif did not significantly enhance EV degradation efficiency compared with those of the negative control (Fig. EV4 v). This finding confirmed that the triple LIR motifs promoted EV degradation by binding to LC3.”

20. 3k - would have been better to include control EVs as well to see how degradation compared. Did you test the kinetics of degradation - i.e. one would expect more degradation even at the uptake timepoint of 12h. When you compare Figs. S1f, g, h to this, the degradation at the same time point is more robust without the need for additional SQSTM1. Additionally, here is an ideal

place to test the requirements for ATG7 and MAP1LC3B^{-/-} cells as well, since the mechanism have been suggested just prior to this. Do the engineered degraders work with the same mechanisms that led to these degraders in the first place?

Reply: Thank you very much for your insightful questions and valuable suggestions.

a. To more intuitively demonstrate the enhancing effect of LIR motif loading on the degradation efficiency of EVs, we introduced unmodified blank EVs as a negative control and detected the GFP signal within the cells 12 hours after uptake to evaluate the impact of 3xLIR motif loading on the degradation efficiency of EVs. Meanwhile, we set up a control group where SQSTM1 was directly overexpressed in EVs. The results showed that the expression of SQSTM1 could significantly increase the degradation efficiency of EVs, which was consistent with our previous results (Fig. 2). Moreover, compared with direct expression of SQSTM1, the loading of 3x LIR motifs could further enhance the degradation efficiency of EVs. This phenomenon might be attributed to the limited sorting capacity of EVs for SQSTM1 protein, which is far lower than the expression level of its scaffold protein CD63. Therefore, the 3xLIR-CD63 fusion protein has an advantage in expression efficiency on EVs surfaces.

Appendix Fig. S3a Representative immunoblot images and quantitative analysis of the resident GFP level of blank, SQSTM1 overexpression and 3x LIR motifs engineering EVs in HeLa cells (n=3).

b. To verify whether the degradation mechanism of LIR motif-loaded EVs depends on ATG7 and MAP1LC3B, we examined the degradation efficiency of EVs in wild-type, ATG7^{-/-}, and MAP1LC3B^{-/-} cells. The results indicated that, in comparison with wild-type cells, both ATG7^{-/-} and MAP1LC3B^{-/-} significantly impeded the degradation efficiency of 3x LIR motif-loaded EVs (Appendix Fig. S3 b-c). This further verified that LIR motif-engineered EVs enter the degradation pathway in a manner that depends on autophagy and is reliant on MAP1LC3B recognition.

Appendix Fig. S3b-c Representative immunoblot images and quantitative analysis of the resident GFP level of 3x LIR motif-loaded EVs in the wild type, *ATG7*^{-/-} (b) and *MAP1LC3B1*^{-/-} (c) HeLa cell lines (n=3).

We added the above content as a part of the result in **Page 9 Line 25-31** as follows:

“We confirmed whether the degradation mechanism of LIR motif-loaded EV was dependent on *ATG7* and *MAP1LC3B* by evaluating the EV degradation efficiency in wild-type, *ATG7*^{-/-}, and *MAP1LC3B*^{-/-} cells. The results showed that both *ATG7* KO and *MAP1LC3B* KO significantly impeded the degradation efficiency of the triple LIR motif-loaded EVs (Appendix Fig. S3b-c). This confirmed that the triple LIR motif-loaded EVs entered the degradation process in an autophagy-dependent and *MAP1LC3B* recognition-dependent manner. Thus, LIR motif-loaded EVs are efficiently sorted into lysosomes for degradation via the interaction with *MAP1LC3B*.”

21. 4e - This is a key figure, but it does not clarify exactly what EVs were used here. What are non-target controls? Were nanoflow sorted EVs containing both target and 3LIR used or was it a mix pool of EVs, given ~5% of EVs contain both GFP and RFP signals? Similarly, what EVs were used for Fig. 4f and g? For 4f, g & h, what are the controls for BafA1, MG132 and *MAP1LC3B* KD?

Reply: Thank you very much for your insightful questions and valuable suggestions.

- For clarity, we provide a detailed description of the experimental grouping as follows: First, based on the presence or absence of the TNFR1 fusion protein (i.e., TNF α -targeting recognition signal), the EVs were categorized into two main groups — the non-targeted EV group (without TNFR1 fusion protein) and the TNF α -targeted EV group (with TNFR1 fusion protein). Subsequently, each of these groups was further classified based on the presence and type of degradation signal, resulting in four subgroups: no degradation signal (No LTS), 3LIR-CD63 fusion protein-loaded (3LIR-CD63), 3LIR-SDC fusion protein-loaded (3LIR-SDC), and 3LIR-SDCB1 fusion protein-loaded (3LIR-SDCB1). Consequently, a total of eight distinct experimental groups were established based on the combination of TNFR1 fusion protein presence and degradation signal type.
- For the isolation of dual-positive EVs, fluorescence-activated sorting (CytoFLEX SRT) was employed based on the RFP signal fused to the C-terminal of the targeting signal and the GFP

signal fused to the C-terminal of the degradation signal (as shown in Fig. 4a), to specifically enrich EVs carrying both targeting and degradation signals. The sorting efficiency was further validated using NanoFCM, demonstrating that the proportion of dual-positive EVs exceeded 98% across all samples as shown below:

Appendix Fig. S3e The loading efficiency of degradation signals (GFP) and targeting signals (RFP) on engineering EVs before and after fluorescence-activated sorting were detected using NanoFCM.

- c. For Fig. 4f, the control groups for the BafA1 and MG132 experimental groups were negative control groups treated with the same dose of solvent (DMSO). For Fig. 4g, the control group for MAP1LC3B was a negative control group treated with Scramble shRNA that lacks targeting ability¹. For clarity, we have explicitly annotated the corresponding positions in the figure with "DMSO" and "shScramble" in place of the general term "control" to prevent potential ambiguity and enhance conceptual precision.

Fig. 4c ELISAs were used to detect the degradation rate of TNF- α by engineered EVs in the DMSO, BafA1 and MG132 pretreatment groups ($n=4$);

Fig. 4d TNF- α degradation level of engineered EVs were detected by ELISA in shScramble and shMAP1LC3B pretreated NPCs ($n=4$);

Reference:

1. Fischer, J. W., Busa, V. F., Shao, Y., & Leung, A. K. L. (2020). Structure-Mediated RNA Decay by UPF1 and G3BP1. *Molecular cell*, 78(1), 70–84.e6. <https://doi.org/10.1016/j.molcel.2020.01.021>

We added the above content as a part of the result in Page 10 Line 11-22 as follows:

“Then, triple LIR motifs fused with the EV scaffolds were co-loaded to generate TNF α -AATEC (Fig. 4a). Nanoflow was used to assess the loading efficiency of both the signals at the single-EV level (Appendix Fig. S3e). To isolate dual-positive EVs, fluorescence-activated sorting was used based on the RFP signal fused to the C-terminal of the targeting signals and the GFP signal fused to the C-terminal of the degradation signals (as shown in Fig. 4a). The sorting efficiency was further validated using NanoFCM, which showed that the proportion of dual-positive EVs was > 98% in all samples (Appendix Fig. S3e). Next, ELISA was performed to assess the capacity of TNF α -AATEC to degrade TNF- α . The results showed that neither the degradation signals alone (3x LIR-CD63, 3x LIR-SDC, or 3x LIR-SDCB1) nor targeting signals alone (TNFR1–CD63 fusion protein) alone degraded TNF- α (Fig. 4b). Only EVs with both signals significantly reduced the TNF- α level in the supernatant. Additionally, among the three degradation signals, 3x LIR-CD63 and 3x LIR-SDCB1 showed better degradation ability against TNF- α (Fig. 4b).”

22. Pg 9 - Fig. S4 in many places should be S5.

Reply: We sincerely apologize for the error in the text. The corresponding expression has been corrected from "Fig S4" to "Fig S5".

23. 5i-j: Precisely what are EV and AATEC controls? For this experiment, control EVs or those only presenting 3LIR or only presenting rTNF α R are ideal negative controls.

Reply: Thank you very much for your insightful questions and valuable suggestions.

- a. The control groups we set up are introduced as follows: EV represents unmodified blank EVs, AATEC refers to EVs modified with degradation signals (i.e., 3LIR-CD63) but without the

targeting signal for TNF α , and rTNF α -AATEC indicates double-positive EVs that carry both degradation and TNF α -targeting signals. For better understanding, we have added corresponding descriptions at the relevant positions in the results Page 10 Line 41-42 as: “Unmodified blank EVs (EV) and EVs modified with degradation signals but without the targeting signal (AATEC) were used as control groups.”

- b. For the control group that only displayed the degradation signal (3xLIR motifs), we have set it up in Fig. 5i and labeled it as the "AATEC" group. The results showed that it did not have the ability to degrade TNF- α and could not significantly alleviate the progression of intervertebral disc degeneration. This indicates that the EV-based targeted degradation strategy against TNF- α has therapeutic effects on intervertebral disc degeneration.

Fig. 5h The ability of EVs, AATEC and rTNF α -AATEC to degrade TNF- α in rat intervertebral discs was detected by ELISA ($n=6$)

- c. Targeted recognition receptors designed against TNF- α and utilized as neutralizing agents represent an established and widely adopted approach for counteracting cytokines¹. Therefore, incorporating a therapeutic control group is essential for comparative analysis. To evaluate the therapeutic efficacy of cytokine neutralizing agents versus EV-based targeted degradation strategies, we conducted comparative experiments using both *in vitro* and *in vivo* intervertebral disc degeneration model (Appendix Fig. S5).

In this context, EVs co-loaded with TNFR1 and IL1R2 were employed as the neutralizing agent control group (T/I-EV), whereas EVs co-loaded with both TNFR1/IL1R2 targeting signals and the 3x LIR motifs constituted the targeted degradation group (T/I-AATEC). Additionally, we mixed EVs co-loaded with TNFR1 and IL1R2 with those carrying only the 3x LIR motifs at a 1:1 ratio (T/I-EV + AATEC) to assess whether the targeting and degradation signals function independently. The therapeutic effects of various treatment modalities were evaluated using both *in vitro* nucleus pulposus cell degeneration models and *in vivo* intervertebral disc degeneration models.

Our results demonstrated that the neutralizing agent group (T/I-EV), the mixture of neutralizing agents and non-targeted degradation EVs (T/I-AATEC), and the targeted degradation group (T/I-AATEC) all effectively inhibited the progression of intervertebral disc degeneration. Notably, the targeted degradation group exhibited significantly superior therapeutic effects compared to the group treated solely with neutralizing agents, indicating that targeted degradation of inflammatory factors offers enhanced therapeutic efficacy over simple cytokine neutralization. Furthermore, the targeted degradation group also outperformed the mixed group of neutralizing agents and non-targeted degradation EVs, suggesting that the therapeutic function of targeted degradation EVs relies on the synergistic action of both targeted recognition and intracellular degradation, rather than the independent

operation of these components.

Appendix Fig. S5 T/I-AATECs displayed stronger anti-inflammatory and regenerative repair ability compared with decoy EVs.

a The mRNA level of inflammatory cytokines (IL-1 β , IL6), matrix metalloproteinases (MMP3, MMP14) and extracellular matrix (collagen II, aggrecan) in LPS pre-conditioned NPCs treated with EV, T/I-EV, mixture of T/I-EV and AATEC, T/I-AATEC for 24 h were detected by Real-time PCR ($n=3$); **b** The level of I κ B α and phosphorylation of p65 in LPS pre-conditioned NPCs treated with EV, T/I-EV, mixture of T/I-EV and AATEC, T/I-AATEC for 24 h were detected by western blot; **c-f** Rat co/7/8 IVDs following needle puncture were treated with EVs, rI/T-EVs, mixture of rI/T-EVs and AATEC, rI/T-AATEC. MRI images and Pifrrmann grades of rat co/7/8

IVDs in different groups were shown in (c) while co8/9 as control ($n=6$); H&E (d), S-O (e), Masson (f) staining of rat IVDs were shown (Scale bar: 50 μ m; 5 μ m). Data were analysed by unpaired two-tailed t-tests (a, c). Data were shown as mean \pm SD. Each n in c is an individual rat, n in a is biological independent samples.

Reference:

1. Conceição, M., Forcina, L., Wiklander, O. P. B., Gupta, D., Nordin, J. Z., Vrellaku, B., McClorey, G., Mäger, I., Görgens, A., Lundin, P., Musarò, A., Wood, M. J. A., Andaloussi, S. E., & Roberts, T. C. (2021). Engineered extracellular vesicle decoy receptor-mediated modulation of the IL6 trans-signalling pathway in muscle. *Biomaterials*, 266, 120435. <https://doi.org/10.1016/j.biomaterials.2020.120435>

We added the above content as a part of the result in **Page 13 Line 35-43 and Page 14 Line 1-14** as follows:

“As the decoy EVs that are only equipped with the targeting domain elicit anti-inflammatory effects[46, 52], we compared the therapeutic efficacy of cytokine-neutralizing agents against those of EV-based targeted degradation strategies. We performed comparative experiments using both *in vitro* and *in vivo* IVDD models. EVs co-loaded with TNFR1 and IL1R2 were used as the neutralizing agent control group (T/I-EV), whereas EVs co-loaded with both TNFR1/IL1R2 targeting signals and triple LIR motifs constituted the targeted degradation group (T/I-AATEC). Additionally, we mixed EVs co-loaded with TNFR1 and IL1R2 with EVs carrying only the triple LIR motifs in a 1:1 ratio (T/I-EV + AATEC) to assess whether targeting and degradation signals function independently. The results showed that T/I-AATEC exerted potent anti-inflammatory effect, whereas T/I-EV exhibited a relatively mild regenerative effect by promoting ECM production and inhibiting MMP and inflammatory cytokine production (Appendix Fig. S5a). Furthermore, combining T/I-EV and AATEC did not significantly enhance the regenerative capacity of T/I-EVs (Appendix Fig. S5a), which shows that T/I-AATEC plays a more effective anti-inflammatory role than that of decoy EVs by simultaneously trapping and degrading targeted cytokines. Additionally, the T/I-AATEC group showed the lowest NF- κ B pathway activation level (Appendix Fig. S5b). Next, we applied rT/I-EV and a mixture of rT/I-EVs and AATEC to treat IVDD *in vivo*. The MRI images showed that both rT/I-EVs and the rT/I-EV–AATEC mixture exerted a mild alleviating effect on IVD water loss, whereas T/I-AATEC showed a relatively stronger therapeutic effect (Appendix Fig. S5c). Histological assays including haematoxylin–eosin (H&E), Safranin O-Fast Green (S-O), and Masson staining of the IVD showed that T/I-AATEC exhibited a higher regenerative capacity than those of both T/I-EVs and T/I-EVs–AATEC mixture. This confirms that T/I-AATEC exert a more potent anti-inflammatory effect than that of the decoy EVs. Overall, T/I-AATEC effectively degraded both TNF- α and IL-1 β , which resulted in potent anti-inflammatory effects.”

24. Fig. S5e: Did rTNF α -AATEC significantly improve IVD compared to EVs containing rTNF α alone?

Reply: Thank you very much for your insightful questions and valuable suggestions.

Regarding the comparison of therapeutic effects between targeted degraders and neutralizers, we

have provided the answer in part c of the previous question (No. 23). Here, we used EVs with both TNFR1/IL1R2 as neutralizers and compared their therapeutic effects with those of targeted degrading EVs for TNF α /IL1 β . The results showed that targeted degrading EVs (T/I-AATEC) were more effective than both decoy EVs (T/I-EVs) and the mixture of decoy EVs and non-targeting AATEC (T/I-EVs + AATEC) in treating intervertebral disc degeneration (Appendix Fig. S5).

Appendix Fig. S5 T/I-AATECs displayed stronger anti-inflammatory and regenerative repair ability compared with decoy EVs.

a The mRNA level of inflammatory cytokines (IL-1 β , IL6), matrix metalloproteinases (MMP3, MMP14) and extracellular matrix (collagen II, aggrecan) in LPS pre-conditioned NPCs treated with EV, T/I-EV, mixture of T/I-EV and AATEC, T/I-AATEC for 24 h were detected by Real-time PCR ($n=3$); **b** The level of I κ B α and phosphorylation of p65 in LPS pre-conditioned NPCs treated with EV, T/I-EV, mixture of T/I-EV and AATEC, T/I-AATEC for 24 h were detected by western blot; **c-f** Rat co7/8 IVDs following needle puncture were treated with EVs, rI/T-EVs, mixture of rI/T-EVs and AATEC, rI/T-AATEC. MRI images and Piffrrmann grades of rat co7/8 IVDs in different groups were shown in (c) while co8/9 as control ($n=6$); H&E (d), S-O (e), Masson (f) staining of rat IVDs were shown (Scale bar: 50 μ m; 5 μ m);. Data were analysed by unpaired two-tailed t-tests (**a**, **c**). Data were shown as mean \pm SD. Each n in **c** is an individual rat, n in **a** is biological independent samples.

25. The same comments that applied to EVs containing rTNF α alone TNF α R-AATEC from 21-24 apply to IL1R2-AATECs.

Reply: Thank you very much for your insightful questions and valuable suggestions.

- Regarding the explanation of the experimental groups and the issues with text marking, the IL1 β -targeting section is similar to the TNF- α -targeting section. Therefore, we made the same changes in the corresponding parts as we did for TNF- α to facilitate understanding.
- For the isolation of dual-positive EVs, fluorescence-activated sorting (CytoFLEX SRT) was employed based on the RFP signal fused to the C-terminal of the targeting signal and the GFP signal fused to the C-terminal of the degradation signal (as shown in Fig. 4a), to specifically enrich EVs carrying both targeting and degradation signals. The sorting efficiency was further validated using NanoFCM, demonstrating that the proportion of dual-positive EVs exceeded 97% across all samples.

Appendix Fig. S3f The loading efficiency of degradation signals (GFP) and targeting signals (RFP) on engineering EVs before and after fluorescence-activated sorting were detected using NanoFCM.

- For the neutralizing agent of IL1 β , we have provided the answer in part c of the previous question (No. 23). Based on the combined pro-inflammatory effect of TNF α and IL1 β in intervertebral disc degeneration, we used a TNFR/ILR2 combined loading strategy to uniformly verify the therapeutic effect of the neutralizing agent compared to the targeted degrading agent, without separately isolating IL-1 β and TNF- α . The results showed that

targeted degrading EVs (T/I-AATEC) were more effective than both neutralizing agent (T/I-EVs) and the mixture of decoy EVs and non-targeting AATEC (T/I-EVs + AATEC) in treating intervertebral disc degeneration (Appendix Fig. S5).

Appendix Fig. S5 T/I-AATECs displayed stronger anti-inflammatory and regenerative repair ability compared with decoy EVs.

a The mRNA level of inflammatory cytokines (IL-1 β , IL6), matrix metalloproteinases (MMP3, MMP14) and extracellular matrix (collagen II, aggrecan) in LPS pre-conditioned NPCs treated with EV, T/I-EV, mixture of T/I-EV and AATEC, T/I-AATEC for 24 h were detected by Real-time PCR ($n=3$); **b** The level of IkBa and

phosphorylation of p65 in LPS pre-conditioned NPCs treated with EV, T/I-EV, mixture of T/I-EV and AATEC, T/I-AATEC for 24 h were detected by western blot; **c-f** Rat co7/8 IVDs following needle puncture were treated with EVs, rI/T-EVs, mixture of rI/T-EVs and AATEC, rI/T-AATEC. MRI images and Piffmann grades of rat co7/8 IVDs in different groups were shown in (c) while co8/9 as control ($n=6$); H&E (d), S-O (e), Masson (f) staining of rat IVDs were shown (Scale bar: 50 μ m; 5 μ m); Data were analysed by unpaired two-tailed t-tests (a, c). Data were shown as mean \pm SD. Each n in c is an individual rat, n in a is biological independent samples.

26. S5k,l: Exactly what was used here and what cell lines/models? IL1R1 binding domain or IL1R1 itself? If all the source of IL1R1 was the EVs, then degradation of EVs would result in reduction of IL1R1 anyway. So how is this targeting the degradation of receptors? This needs a lot more expansion and more controls to demonstrate targeted degradation of receptors proteins.

Reply: Thank you very much for your insightful questions and valuable suggestions.

- In Fig S5 k, l: human nucleus pulposus cells were utilized, and the treatments included blank unmodified extracellular vesicles (EVs) as well as EVs co-loaded with IL1RA (IL1R1 recognition signal) and 3LIR (degradation signal), referred to as IL1R1-AATEC. The loading strategy is illustrated in Figure S5k.
- We used IL-1RA (which has been reported in the literature as an inhibitory binding ligand of IL1R1¹) as the recognition signal for AATEC to identify IL1R1, and the engineering EVs do not carry IL1R1. The source of IL1R1 is the endogenous expression in the nucleus pulposus cells, rather than an exogenous supplement from EVs. Therefore, the expression level of IL1R1 within the cells can effectively reflect the degradation effect of IL1R1-AATEC on the target receptor.
- For the targeted degradation of IL1R1, we added two control groups: one group contained only IL1Ra (the recognition signal for IL1R1, i.e., the inhibitor), and the other group contained only the degradation sequence (3xLIR motif) in EVs. The results confirmed that neither of these two groups could effectively degrade IL1R1 in nucleus pulposus cells. This further confirmed that the targeted degradation ability of AATEC depends on the combined effect of the targeting signal and the degradation signal.

Appendix Fig. S3d Protein level analysis of IL1R1 levels in HeLa cells pretreated with blank EVs, IL1Ra-loaded EVs, 3xLIR-loaded EVs and IL1R1-AATEC (loaded with both IL1Ra and 3xLIR motifs) for 24 h ($n=3$).

Reference:

- Garlanda, C., Dinarello, C. A., & Mantovani, A. (2013). The interleukin-1 family: back to the

future. *Immunity*, 39(6), 1003–1018. <https://doi.org/10.1016/j.immuni.2013.11.010>

We added the above content as a part of the result in **Page 12 Line 8-16** as follows:

“Furthermore, we verified the ability of EVs to degrade cell surface proteins by loading the IL1R1 binding domain (derived from IL1Ra) and triple LIR motifs into IL1R1-AATEC to degrade IL1R1 in human NPCs (Fig. EV5 j). The results showed that IL1R1-AATEC efficiently degraded IL1R1 (Fig. EV5 k), which confirmed the ability of AATEC to degrade cell-surface targets. Moreover, we created two control groups: one containing only IL1Ra (the recognition signal for IL1R1, i.e., the inhibitor) and the other containing only the triple LIR motifs in EVs. The results indicated that neither of these two groups showed effective degradation of IL1R1 in NPCs (Appendix Fig. S3d). This validated the fact that targeted degradation ability of AATEC depended on the combined effect of both targeting and degradation signals.”

27. Fig. 5a: How was the quality of T/I-AATEC assessed relative to I- or T-AATECs? This is because, if you mixed I-AATEC EVs with T-AATECs (instead of having both in the same EVs), I suspect the outcome would be the same. Given that the percentage of EVs expressing two components in Fig. 4 (TNFR1 & 3LIR) was so low, one would expect even lower (negligible) EVs with all 3 components in one. It is not fully explained how these were purified, separated and applied. The authors need to demonstrate that EVs that have dual targeting are the ones that target both extracellular proteins and are better than individual or mixed pools of individual-targeting EVs. The same applies to all in vivo experiments where they should apply a 50% mixture of individual-targeting EVs (I- and T-AATEC) as controls - I suspect these would perform as well as I/T-AATEC EVs.

Reply: Thank you very much for your insightful questions and valuable suggestions.

- a. For the purification of T/I-AATEC, we mainly used fluorescence-based sorting technology. We respectively fused BFP, GFP and RFP to the C-terminal of the degradation signal 3xLIR-CD63, the targeting signal TNFR1 and the targeting signal IL1R2, and sorted the triple positive EVs by fluorescence sorting technology (CytoFLEX SRT). The sorting efficiency of EVs was rechecked by NanoFCM and was almost 100%, as shown below:

Appendix Fig. S3g The loading efficiency of degradation signals (BFP) and targeting signals (RFP and GFP) on engineering EVs before and after fluorescence-activated sorting were detected using NanoFCM.

- b. Our aim in expressing TNFR1 and IL1R2 simultaneously on the same EVs is to endow each EV with more degrading functions, that is, to degrade both TNF α and IL1 β at the same time. Therefore, theoretically, the targeted degradation effect of T/I-AAATEC expressing TNFR1 and IL1R2 simultaneously should be equivalent to the sum of the targeted degradation effects of T-AAATEC and I-AAATEC. Conversely, if the same dose is applied, the targeted degradation effect of T/I-AAATEC should be twice that of the mixed pool of T-AAATEC and I-AAATEC. Take TNF- α as an example: under the same dosing conditions, only half of the mixed pool (T-AAATEC) is capable of targeting TNF- α , whereas the entire population of T/I-AAATEC, which are co-loaded with TNFR and IL1R2, can effectively mediate the targeted degradation of TNF- α .

We further verified this interesting hypothesis through *in vitro* experiments in NP cells. ELISA results showed that the the mixed pool of T-AAATEC and I-AAATEC has the ability to degrade both TNF α and IL1 β . However, the targeted degradation effect of T/I-AAATEC was significantly better than that of the mixed pool of T-AAATEC and I-AAATEC, which might be due to the difference in the loading amounts of TNFR1 and IL1R2 on each EV.

Appendix Fig. S3h ELISAs were used to detect the degradation of TNF- α and IL-1 β by EVs, TNF- α -

AATEC, IL1 β -AATEC, mix pool of TNF α -AATEC and IL1 β -AATEC, T/I-AATEC at the concentration of 10×10^3 per cell after cocubation with NPCs for 24 h ($n=4$).

We added the above content as a part of the result in **Page 12 Line 30-36** as follows:

“We hypothesized that as the simultaneous expression of TNFR1 and IL1R2 on the same EVs endows each EV with additional degrading capabilities such as the ability to degrade both TNF- α and IL-1 β simultaneously, the targeted degradation ability of T/I-AATEC is the sum of that of T-AATEC and I-AATEC at the same dose. We verified this hypothesis by performing an *in vitro* experiment. ELISA results showed that the targeted degradation effect of T/I-AATEC was significantly better than that of the mixed pool of T-AATEC and I-AATEC at the same dose (Appendix Fig. S3h), which may be attributed to the difference in the loading quantities of TNFR1 and IL1R2 on each EV.”

28. For Fig. 6 *in vivo* IVD degeneration assays, I am unable to comment on biology of IVD. However, the key question for me would be whether targeted degradation of TNF- α and IL-1 β would be more efficacious than neutralisation of TNF- α and IL-1 β . Therefore, key comparisons for me would have to be whether T/I-AATECs (degraders) perform better than those EVs that present TNF- α R and IL-1 β R without the 3LIR (neutralisers). Here the T/I-AATECs are compared with unloaded EVs, making any conclusions difficult as the clinical efficacy could have been just as much due to neutralisation of the cytokines as their degradation. The dual degradation part is still underdeveloped and needs a lot more controls. The *in vivo* time-lines are a lot longer than the *in-cellulo* timelines for degradation - comments?

Reply: Thank you very much for your insightful questions and valuable suggestions.

- a. Targeted recognition receptors designed against TNF- α and utilized as neutralizing agents represent an established and widely adopted approach for counteracting cytokines¹. Therefore, incorporating a therapeutic control group is essential for comparative analysis. To evaluate the therapeutic efficacy of cytokine neutralizing agents versus EV-based targeted degradation strategies, we conducted comparative experiments using both *in vitro* and *in vivo* intervertebral disc degeneration model (Appendix Fig. S5).

In this context, EVs co-loaded with TNFR1 and IL1R2 were employed as the neutralizing agent control group (T/I-EV), whereas EVs co-loaded with both TNFR1/IL1R2 targeting signals and the 3x LIR motifs constituted the targeted degradation group (T/I-AATEC). Additionally, we combined EVs co-loaded with TNFR1 and IL1R2 with EVs carrying only the 3x LIR motifs at a 1:1 ratio (T/I-EV + AATEC) to assess whether the targeting and degradation signals function independently. The therapeutic effects of various treatment modalities were evaluated using both *in vitro* nucleus pulposus cell degeneration models and *in vivo* intervertebral disc degeneration models. Our results demonstrated that the neutralizing agent group (T/I-EV), the mixture of neutralizing agents and non-targeted degradation EVs (T/I-AATEC), and the targeted degradation group (T/I-AATEC) all effectively inhibited the progression of intervertebral disc degeneration. Notably, the targeted degradation group exhibited significantly superior therapeutic effects compared to the group treated solely with neutralizing agents, indicating that targeted degradation of inflammatory factors offers

enhanced therapeutic efficacy over simple cytokine neutralization. Furthermore, the targeted degradation group also outperformed the mixed group of neutralizing agents and non-targeted degradation EVs, suggesting that the therapeutic function of targeted degradation EVs relies on the synergistic action of both targeted recognition and intracellular degradation, rather than the independent operation of these components.

Appendix Fig. S5 T/I-AATECs displayed stronger anti-inflammatory and regenerative repair ability compared with decoy EVs.

a The mRNA level of inflammatory cytokines (IL-1 β , IL6), matrix metalloproteinases (MMP3, MMP14) and extracellular matrix (collagen II, aggrecan) in LPS pre-conditioned NPCs treated with EV, T/I-EV, mixture of T/I-EV and AATEC, T/I-AATEC for 24 h were detected by Real-time PCR ($n=3$); **b** The level of I κ B α and

phosphorylation of p65 in LPS pre-conditioned NPCs treated with EV, T/I-EV, mixture of T/I-EV and AATEC, T/I-AATEC for 24 h were detected by western blot; **c-f** Rat co7/8 IVDs following needle puncture were treated with EVs, rI/T-EVs, mixture of rI/T-EVs and AATEC, rI/T-AATEC. MRI images and Pfirrmann grades of rat co7/8 IVDs in different groups were shown in (c) while co8/9 as control ($n=6$); H&E (d), S-O (e), Masson (f) staining of rat IVDs were shown (Scale bar: 50 μ m; 5 μ m);. Data were analysed by unpaired two-tailed t-tests (a, c). Data were shown as mean \pm SD. Each n in c is an individual rat, n in a is biological independent samples.

- b. We have confirmed through *in vivo* experiments that the natural degradation of EVs in the local intervertebral disc takes 3 weeks, which is much longer than the 24-hour degradation process *in vitro* (Fig. 1). The reason for this might be that the amount of EVs used in the *in vivo* experiment (1×10^{10} /mL) is much higher than that in the *in vitro* experiment (1×10^8 /mL). Additionally, the internal micro-environment of the intervertebral disc is relatively independent and avascular, with low metabolic activity and a small number of nucleus pulposus cells¹. These factors might contribute to the slower degradation rate of EVs within the intervertebral disc *in vivo*. Based on the unique slow degradation characteristics of EVs in the intervertebral disc, we did not conduct repeated dosing but rather single-dose administration and evaluated the effect after the treatment ended.

Reference:

1. Conceição, M., Forcina, L., Wiklander, O. P. B., Gupta, D., Nordin, J. Z., Vrellaku, B., McClorey, G., Mäger, I., Görgens, A., Lundin, P., Musarò, A., Wood, M. J. A., Andaloussi, S. E., & Roberts, T. C. (2021). Engineered extracellular vesicle decoy receptor-mediated modulation of the IL6 trans-signalling pathway in muscle. *Biomaterials*, 266, 120435. <https://doi.org/10.1016/j.biomaterials.2020.120435>
2. Risbud, M. V., & Shapiro, I. M. (2014). Role of cytokines in intervertebral disc degeneration: pain and disc content. *Nature reviews. Rheumatology*, 10(1), 44–56. <https://doi.org/10.1038/nrrheum.2013.160>

We added the above content as a part of the result in **Page 13 Line 35-43 and Page 14 Line 1-14** as follows:

“As the decoy EVs that are only equipped with the targeting domain elicit anti-inflammatory effects[46, 52], we compared the therapeutic efficacy of cytokine-neutralizing agents against those of EV-based targeted degradation strategies. We performed comparative experiments using both *in vitro* and *in vivo* IVDD models. EVs co-loaded with TNFR1 and IL1R2 were used as the neutralizing agent control group (T/I-EV), whereas EVs co-loaded with both TNFR1/IL1R2 targeting signals and triple LIR motifs constituted the targeted degradation group (T/I-AATEC). Additionally, we mixed EVs co-loaded with TNFR1 and IL1R2 with EVs carrying only the triple LIR motifs in a 1:1 ratio (T/I-EV + AATEC) to assess whether targeting and degradation signals function independently. The results showed that T/I-AATEC exerted potent anti-inflammatory effect, whereas T/I-EV exhibited a relatively mild regenerative effect by promoting ECM production and inhibiting MMP and inflammatory cytokine production (Appendix Fig. S5a). Furthermore, combining T/I-EV and AATEC did not significantly enhance the regenerative capacity of T/I-EVs (Appendix Fig. S5a), which shows that T/I-AATEC plays a more effective anti-inflammatory role than that of decoy EVs by simultaneously trapping and degrading targeted

cytokines. Additionally, the T/I-AATEC group showed the lowest NF-kB pathway activation level (Appendix Fig. S5b). Next, we applied rT/I-EV and a mixture of rT/I-EVs and AATEC to treat IVDD *in vivo*. The MRI images showed that both rT/I-EVs and the rT/I-EV–AATEC mixture exerted a mild alleviating effect on IVD water loss, whereas T/I-AATEC showed a relatively stronger therapeutic effect (Appendix Fig. S5c). Histological assays including haematoxylin–eosin (H&E), Safranin O-Fast Green (S-O), and Masson staining of the IVD showed that T/I-AATEC exhibited a higher regenerative capacity than those of both T/I-EVs and T/I-EVs–AATEC mixture. This confirms that T/I-AATEC exert a more potent anti-inflammatory effect than that of the decoy EVs. Overall, T/I-AATEC effectively degraded both TNF- α and IL-1 β , which resulted in potent anti-inflammatory effects.”

The manuscript also has a lot of editorial issues and would benefit from an editorial proof-reading. Furthermore, there are insufficient experimental details in many places making it hard to understand the experiment in question. This needs to be addressed thoroughly.

Reply: Thank you very much for your insightful questions and valuable suggestions. We have supplemented the necessary experimental content and carried out comprehensive revisions and polishing of the language within the manuscript, which has significantly enhanced the scientificity and readability of our article.

3rd Dec 2025

Dear Prof. Yang,

Thank you for the submission of your revised manuscript to EMBO Molecular Medicine. We have now received the enclosed report from one of the two referees who agreed to re-assess it. As the original Reviewer #3 was unable to provide a timely review, we had asked Referee #2 to also evaluate your response to Reviewer #3's comments. As you will see, the referee is now supportive, and I am pleased to inform you that we will be able to accept your manuscript pending the following amendments:

The remaining minor issues raised by Ref #2.

On a more editorial level:

1. Please reduce the keyword number to five.
 2. Please remove the "Authors' contribution" section.
 3. The references need to be formatted according to the EMBO Molecular Medicine reference style. Please list up to 10 co-authors of a paper before adding et al. in the reference list. Citations should be listed in alphabetical order.
 4. "Conflict of Interest statement" should be renamed to "Disclosure and competing interests statement".
 5. Source data: in the nomenclature, please remove "(former figure x)".
 6. During our standard imaging check, our data integrity officer identified some potential cell/sample reuse:
 - between Figure 6G and H
 - cell reuse within Figure EV5: It appears to be a similar reuse of different parts of the same cell.
 - cell reuse (Actin) between Figure Appendix S1A and S3A
 - possible cell reuse within Appendix Figure S4 F & G
 - possible cell reuse within Appendix Figure S5 D & E
- Please carefully re-examine these image panels. If any reuse is intentional, it must be explicitly stated in the corresponding figure legends.
- Please upload the Appendix file at a higher resolution. Currently, the blot become pixelated under image analysis.
7. "The paper explained" requires some grammatical revision. Could you please upload an updated version with improved clarity and readability?
 8. BIORENDER: Please remove from figure legends and add a dedicated "graphics" section to the Methods, following this format:
Graphics:
(some of the... OR Figure #... OR synopsis) Graphics were created with BioRender.com.
 9. Appendix: remove the supplementary methods from the appendix and merge with the Methods section in the main manuscript file.
 10. Please upload the synopsis image as a separate file, in PNG format with the required dimensions of 550 px in width and 300-600 px in height.
 11. EV tables:
 - Please remove the legends for the EV tables from the manuscript text and add them to the top of each corresponding Excel file, above each table.
 - There is a citation for a Supplementary Table 4, please correct the nomenclature.
 12. Data availability: Before submitting your revision, primary datasets produced in this study (RNA-seq and MS data) need to be deposited in appropriate public databases, and the accession numbers and databases listed under 'Data Availability'. Please also provide specific URLs for the the datasets. See <https://www.embopress.org/page/journal/17574684/authorguide#dataavailability>).

13. There are numerous mismatches between the figure subpanels and their corresponding legends. Several subpanels are missing from the figures but are still referenced in the legends. Please carefully review all figures to ensure that every panel is accurately presented and correctly matched to its legend.

14. Please remove the "Correspondence" paragraph on p. 24 of the manuscript text.

15. Please address the following issues in the figure legends:

- Please note that the exact p values are not provided in the legends of figures 1B, F; 2J, K; 3H, K; 4B, C, D, E, F, H, J, K, L, M, O, P; 5B, I; 6F, I; EV 1 T, U; EV2 G, EV4 B, EV5 A, C, E, F, G.
- Please indicate the statistical test used for data analysis in the legends of figures 2C, D
- Please note that information related to n is missing in the legend of figure 2D

We look forward to receiving a new revised version of your manuscript as soon as possible.

Kind regards,
Jingyi

Jingyi Hou
Senior Editor
EMBO Molecular Medicine

*** Instructions to submit your revised manuscript ***

***** Reviewer's comments *****

Referee #2 (Remarks for Author):

I appreciate the authors' efforts. All my concerns have been addressed. This manuscript is ready for publication.

I appreciate the authors' efforts in addressing the reviewer #3's concerns. Many additional experiments were conducted and the quality of the manuscript has improved significantly. I am satisfied with most of the responses and new data. A minor issue remains regarding the name AATEC, which is still used in the caption for Figure EV5.

The authors addressed the remaining editorial issues.

11th Dec 2025

Dear Prof. Yang,

We are pleased to inform you that your manuscript is accepted for publication and is now being sent to our publisher to be included in the next available issue of EMBO Molecular Medicine.

You may qualify for financial assistance for your publication charges - either via a Springer Nature fully open access agreement or an EMBO initiative. Check your eligibility: <https://link.springer.com/journal/44321/how-to-publish-with-us>

Yours sincerely,
Jingyi

Jingyi Hou
Senior Editor
EMBO Molecular Medicine

>>> Please note that it is EMBO Molecular Medicine policy for the transcript of the editorial process (containing referee reports and your response letter) to be published as an online supplement to each paper. If you do NOT want this, you will need to inform the Editorial Office via email immediately. More information is available here: <https://link.springer.com/partners/embo-press/editorial-policies#Peer%20review>